# CROSS-ATTENTION HEAD POSITION PATTERNS CAN ALIGN WITH HUMAN VISUAL CONCEPTS IN TEXT-TO-IMAGE GENERATIVE MODELS

**Jungwon Park[1], Jungmin Ko[2], Dongnam Byun[1], Jangwon Suh[1], Wonjong Rhee[1,2*]**

[1]Department of Intelligence and Information, Seoul National University

[2]Interdisciplinary Program in Artificial Intelligence, Seoul National University

{quoded97, jungminko, east928, rxwe5607, wrhee}@snu.ac.kr

## ABSTRACT

Recent text-to-image diffusion models leverage cross-attention layers, which have been effectively utilized to enhance a range of visual generative tasks. However, our understanding of cross-attention layers remains somewhat limited. In this study, we introduce a mechanistic interpretability approach for diffusion models by constructing Head Relevance Vectors (HRVs) that align with human-specified visual concepts. An HRV for a given visual concept has a length equal to the total number of cross-attention heads, with each element representing the importance of the corresponding head for the given visual concept. To validate HRVs as interpretable features, we develop an ordered weakening analysis that demonstrates their effectiveness. Furthermore, we propose *concept strengthening* and *concept adjusting* methods and apply them to enhance three visual generative tasks. Our results show that HRVs can reduce misinterpretations of polysemous words in image generation, successfully modify five challenging attributes in image editing, and mitigate catastrophic neglect in multi-concept generation. Overall, our work provides an advancement in understanding cross-attention layers and introduces new approaches for fine-controlling these layers at the head level[1].

## 1 INTRODUCTION

Recent advancements in Text-to-Image (T2I) models have demonstrated an unprecedented ability to generate high-quality images with strong image-text alignment. These models often leverage powerful pre-trained text encoders; for instance, Stable Diffusion (Rombach et al., 2022) uses CLIP (Radford et al., 2021), while Imagen (Saharia et al., 2022) uses T5 (Raffel et al., 2020). Given that language is more expressive than previously used supervision signals (Gandelsman et al., 2023), text representations have empowered visual generative tasks, allowing for a high degree of control over the generation process. However, our understanding of the inner workings of T2I models remains limited, and even the latest models continue to struggle with certain failure cases.

Significant progress has been made in understanding the inner workings of deep neural networks. Olah et al. (2018) demonstrated numerous examples showing interpretable features at various levels in neural networks; Olah et al. (2020) expanded this analysis by exploring connections between units in the networks. Notably, Templeton et al. (2024) identified interpretable features in the cutting-edge large language model (LLM)–Claude 3 Sonnet–and used these features to guide the model's generation towards safer outcomes. Building on this line of work, we analyze T2I generative models with a focus on their cross-attention (CA) layers. We introduce a novel method to construct *head relevance vectors* (HRVs) that align with *human-specified* visual concepts. An HRV for a given visual concept is a vector whose length equals to the total number of CA heads, with each element representing the importance of the corresponding head for that concept. We demonstrate that these vectors reflect

---

[*]Corresponding Author.
[1]Our code is available at https://github.com/SNU-DRL/HRV

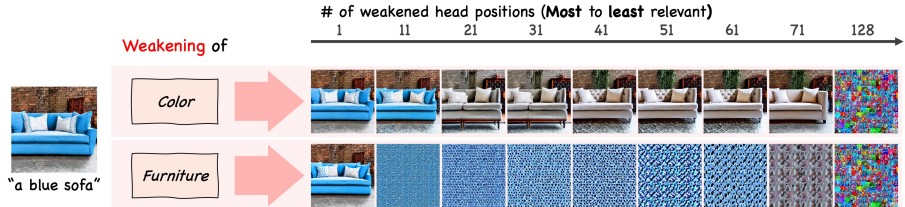

(a) Ordered weakening analysis of CA heads from high to low relevance for two visual concepts.

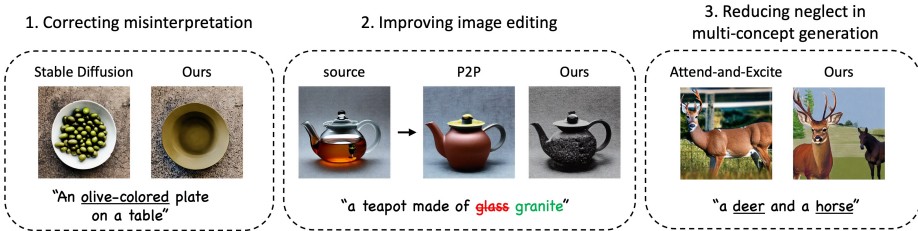

(b) Enhancing three visual generative tasks with our head relevance vectors.

Figure 1: We develop a method for constructing head relevance vectors (HRVs) that align with useful visual concepts. For a specified visual concept, an HRV assigns a relevance score to individual cross-attention heads, revealing their importance for the visual concept. Our analysis shows that the constructed HRVs can serve as interpretable features. We also demonstrate that HRV can be effectively integrated for improving three visual generative tasks.

human-interpretable features by using *ordered weakening analysis*, where we sequentially weaken the activation of CA heads and examine the resulting generated images (Figure 1a).

To demonstrate the utility of HRVs, we propose *concept strengthening* and *concept adjusting* methods to enhance three visual generative tasks (Figure 1b). In image generation, our approach significantly reduces the misinterpretation of polysemous words. When generating images from text prompts containing such words, our method decreased the misinterpretation rate from 63.0% to 15.9%. In image editing, our method enhances the widely used Prompt-to-Prompt (P2P) (Hertz et al., 2022) algorithm in modifying object attributes (color, material, geometric patterns) and image attributes (image style, weather conditions), achieving 2.32% to 11.79% higher image-text alignment scores compared to state-of-the-art methods. Additionally, in human evaluations, it received more than twice the preference scores compared to existing methods. In multi-concept generation, our method improves upon the state-of-the-art Attend-and-Excite (Chefer et al., 2023) algorithm. By reducing catastrophic neglect – the omission of objects or attributes in generated images – our approach enhances performance by 2.3% to 6.3% across two benchmark types.

We use Stable Diffusion v1.4 (Rombach et al., 2022) as the primary model for our analysis and experiments. To show that our findings are not limited to a single model, we also show that they are consistent in Stable Diffusion XL (Podell et al., 2024). Furthermore, we show that our approach allows for flexible adjustment of human-specified concepts. These results indicate that our method of constructing HRVs can be also useful for different model architectures and target concept.

## 2 RELATED WORK

**Early works on interpretable neurons and interpretable features:** Early works on visual generative models have trained variational autoencoders (VAEs) using specifically designed loss functions on datasets with distinct attributes (Kulkarni et al., 2015; Higgins et al., 2017; Chen et al., 2018; Klys et al., 2018). While these methods successfully controlled a few attributes present in the training dataset, they were limited by a lack of fine-grained control, strong dependence on training data, and possible need for manual supervision of attributes. Another approach is to identify meaningful features in intermediate layers of neural networks (e.g., generative adversarial networks (GANs)), and modify those features to control the generation outputs (Plumerault et al., 2020; Shen & Zhou, 2021). Also, a seminal work by Olah et al. (2018) demonstrated that interpretable features in neural networks can be identified at various levels: single neurons, spatial positions, channels, or groups of neurons across different positions and channels. They presented numerous examples showing how

these interpretable features emerge at various levels within a neural network's architecture. While these efforts successfully revealed meaningful features, they were limited by an inability to capture user-specified or human-specified concepts. Moreover, the identified features were not always directly usable for controlling attributes in generative tasks.

**Recent works with multi-modal and generative models:** Since the development of powerful multi-modal models that can map images and text to a joint embedding space, few studies have used text to interpret intermediate representations in vision models (Goh et al., 2021; Hernandez et al., 2022; Yuksekgonul et al., 2023b). Most recently, Gandelsman et al. (2023) examined self-attention heads from the last layers of CLIP-ViT (Dosovitskiy et al., 2021), identifying meaningful correlations with several visual concepts. While these studies primarily focused on non-generative models, our study shifts focus to the interpretable features within visual generative models. Regarding the recently proposed large language model, Claude 3 Sonnet, Templeton et al. (2024) demonstrated that identifying meaningful concepts from model activations provides valuable insights into understanding model behavior. Among the various features identified, their use of safety-related features to guide text generation towards safer outcomes is particularly relevant for large-scale text-generative models. Our study also explores large-scale generative models, showing that *human-specified* visual concepts can be captured and applied in three distinct visual generative tasks.

**Text-to-image diffusion models with cross-attention layers:** Building on the large-scale T2I diffusion models, researchers have developed methods to tackle various visual generative tasks, such as image editing (Hertz et al., 2022) and multi-concept generation (Chefer et al., 2023). Many of these studies utilize the publicly available Stable Diffusion (Rombach et al., 2022), which efficiently generates images through a diffusion denoising process in latent space. This model incorporates an autoencoder and a U-Net (Ronneberger et al., 2015) with multi-head cross-attention (CA) layers that integrate CLIP (Radford et al., 2021) text embeddings. Researchers have explored these CA layers to enhance control over text-conditioned image generation (Feng et al., 2022; Parmar et al., 2023; Chefer et al., 2023; Tumanyan et al., 2023; Wu et al., 2023). For example, P2P (Hertz et al., 2022) manipulates image layout and structure by swapping CA maps between source and target prompts. However, these methods typically update entire CA layers without offering fine-grained control over individual attention heads. We introduce head relevance vectors and develop two techniques for head-level control of CA layers, enabling precise steering of human-specified visual concepts.

## 3 METHOD FOR CONSTRUCTING HEAD RELEVANCE VECTORS

The core idea involves selecting a set of visual concepts of interest, using a large language model (LLM) to pre-select 10 associated words for each concept, and utilizing random image generations for updating head relevance vectors (HRVs). The key to a successful update lies in identifying the visual concept that best matches a particular head and increasing the corresponding element of the HRVs. Figure 2 illustrates the process of a single update. We begin by providing background on the cross-attention (CA) layer, followed by detailed descriptions of our methodology for constructing HRVs that correspond to a set of human-specified visual concepts.

**Cross-attention in T2I diffusion models:** Let $\mathbf{P}$ be a generation prompt, $\mathbf{Z}_t$ be a noisy image at generation timestep $t$, and $\psi(\cdot)$ be a CLIP text-encoder. In each CA head, the spatial features of the noisy image $\phi(\mathbf{Z}_t)$ are projected to a query matrix $\mathbf{Q} = l_Q(\phi(\mathbf{Z}_t))$, while the CLIP text embedding $\psi(\mathbf{P})$ is projected to a key matrix $\mathbf{K} = l_K(\psi(\mathbf{P}))$ and a value matrix $\mathbf{V} = l_V(\psi(\mathbf{P}))$, using learned linear layers $l_Q, l_K$, and $l_V$. Then, the CA map is calculated by measuring the correlations between $\mathbf{Q}$ and $\mathbf{K}$ as

$$\mathbf{M} = \text{softmax}\left(\frac{\mathbf{Q}\mathbf{K}^T}{\sqrt{d}}\right), \tag{1}$$

where $d$ is the projection dimension of the keys and queries. The CA output $\mathbf{M}\mathbf{V}$ is a weighted average of the value $\mathbf{V}$, with the weights determined by the CA map $\mathbf{M}$. This operation is performed in parallel across multi-heads in the CA layer, and their outputs are concatenated and linearly projected using a learned linear layer to produce the final CA output.

**Image generation prompts, visual concepts, and concept-words:** To generate 2,100 random images, we used 2,100 generation prompts. Of these, 1,000 prompts were constructed using 1,000 ImageNet classes (Deng et al., 2009), formatted as 'A photo of a {class name}.' The remaining 1,100

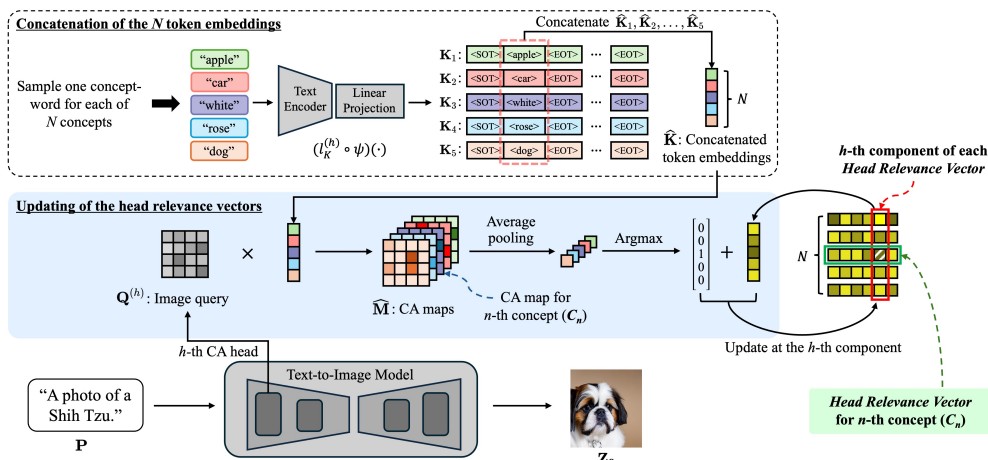

Figure 2: Overview of a single HRV update for a cross-attention (CA) head position $h$. While generating a random image, the most relevant visual concept is identified. Then the concept's head relevance vector (HRV) is updated to have an increased value in position $h$. For illustration purpose, we are showing only 5 visual concepts ($N = 5$) and 6 CA heads ($H = 6$). In our main experiments, we adopt $N = 34$ and $H = 128$. This update is repeated over all the head positions $h = 1, \ldots, H$ and all timesteps $t = 1, \ldots, T$ for a sufficiently large number of random image generations.

prompts were adopted from PromptHero (PromptHero) to enhance diversity. The visual concepts can be specified as an arbitrary set. In our study, we have interacted with GPT-4o (OpenAI, 2024) to list 34 commonly used visual concepts including object categories and image properties. While our study mainly focuses on the 34 visual concepts, we demonstrate in Appendix J that a new visual concept can be flexibly and easily added. For each of the selected visual concepts, we have generated 10 representative concept-words using GPT-4o. The full list of 34 visual concepts, along with 10 corresponding concept-words for each, can be found in Table 3 of Appendix A.

**T2I models and head relevance vectors:** We focus on Stable Diffusion v1, which contains $H = 128$ CA heads. Let $C_1, \ldots, C_N$ represent the specified visual concepts, where $N = 34$ in our main experiments. For each visual concept $C_n$, we define a *head relevance vector (HRV)* as an $H$-dimensional vector that expresses how each CA head is activated by the corresponding visual concept. Initially, all $N$ head relevance vectors are set to zero and are iteratively updated following the process illustrated in Figure 2. The full iteration involved 2,100 generated images, with each image used to iterate over all $H = 128$ heads and $T = 50$ timesteps. Extension to a larger diffusion model, Stable Diffusion XL, is discussed in Section 6.1.

**Concatenation of the $N$ token embeddings:** In each HRV update shown in Figure 2, we generate a concatenation of token embeddings for the $N$ visual concepts and use it to identify the best matching visual concept for the $h$-th head. To enhance diversity in the process, we randomly sample one concept-word for each visual concept from the list provided in Table 3 of Appendix A. The sampled $N$ concept-words are individually embedded using the CLIP text encoder $\psi(\cdot)$, followed by the learned linear key projection layer $l_K^{(h)}$ at the $h$-th CA head position. This produces $N$ key matrices $\mathbf{K}_1, \mathbf{K}_2, \ldots, \mathbf{K}_N \in \mathbb{R}^{77 \times F}$, each corresponding to a concept, with each key matrix containing 77 token embeddings, where $F$ is a feature dimension. For instance, if the sampled word for the third concept $C_3$ is 'white,' the corresponding key matrix $\mathbf{K}_3$ would be [<SOT>, <white>, , $\cdots$, ], where <SOT> and  are key-projected embeddings of special tokens. We only extract the embedding of the semantic token <white> from $\mathbf{K}_3$, denoted as $\widehat{\mathbf{K}}_3 \in \mathbb{R}^{n_3 \times F}$, where $n_3 = 1$ since there is only one semantic token, <white>. Similarly, we extract the embeddings for the other concepts, obtaining $\widehat{\mathbf{K}}_1, \widehat{\mathbf{K}}_2, \ldots, \widehat{\mathbf{K}}_N$. These $N$ embeddings are then concatenated to form $\widehat{\mathbf{K}} = [\widehat{\mathbf{K}}_1, \widehat{\mathbf{K}}_2, \ldots, \widehat{\mathbf{K}}_N] \in \mathbb{R}^{N' \times F}$, where $N'$ represents the total number of semantic tokens across all $N$ concepts ($N' = N$ if each concept-word consists of a single token).

**Updating of the head relevance vectors:** We calculate the cross-attention (CA) maps $\widehat{\mathbf{M}} \in \mathbb{R}^{R^2 \times N'}$, where $R$ is the width or height of the image latent, by applying Eq. 1 using the con-

catenated token embeddings $\widehat{\mathbf{K}} \in \mathbb{R}^{N' \times F}$, in place of $\mathbf{K}$, and the image query $\mathbf{Q}^{(h)} \in \mathbb{R}^{R^2 \times F}$ from $h$-th head position. This measures the correlation between the visual information at $h$-th head position and the textual information of the $N$ visual concepts, resulting in $N$ groups of CA maps. If a word consists of multiple tokens, we average the CA maps across the token dimension so that each word corresponds to a single CA map. This process produces a matrix of shape $\mathbb{R}^{R^2 \times N}$. This matrix is then averaged across spatial dimension (i.e., $R^2$) to yield a single strength value for each visual concept $C_n$, $n = 1, \ldots, N$. We use the argmax operation on these $N$ strength values to identify the visual concept with the largest value, which eliminates the problem of different representation scales across $H$ CA heads (Appendix B.2). Then, we update that visual concept's HRV by increasing its $h$-th component by one. This update process is repeated over all head positions $h = 1, \ldots, 128$ and all timesteps $t = 1, \ldots, 50$ for 2100 random image generations. At the end, each HRV is normalized to have its $L1$ norm equivalent to $H = 128$. Pseudo-code is provided in Appendix B.1.

## 4   ORDERED WEAKENING ANALYSIS OF HEAD RELEVANCE VECTORS

In this section, we investigate whether the constructed HRVs can effectively and reliably serve as interpretable features. As our primary analysis tool, we introduce ordered weakening across the $H$ cross-attention heads by multiplying $-2$ to a target head's CA maps. This weakening is applied consistently across all timesteps during image generation, but only to the CA maps of all semantic tokens, leaving special tokens unchanged. Using this approach, we compare images generated under two different head-weakening orders. In the most relevant head positions first (MoRHF), we weaken $H$ heads starting from the strongest to the weakest, where the strength of head $h$ is determined by the value of the $h$-th element in the HRV. In the least relevant head positions first (LeRHF), the order is reversed, from the weakest to the strongest. If an HRV reliably represents a visual concept, we expect MoRHF to impact the corresponding concept in the generated images more quickly than LeRHF. The head weakening is inspired by a rescaling technique in P2P (Hertz et al., 2022) and the ordering was inspired by the metrics defined in Samek et al. (2016) and Tomsett et al. (2020).

Analysis results for three visual concepts are shown in Figure 3. Comparison of generated images are shown in Figure 3a. In the top case, where the visual concept of Material is weakened, the characteristics of copper in the generated image already disappeared when the most relevant 11 heads are weakened. In contrast, the copper remains visible until the least relevant 71 heads are weakened. A similar observation can be made for the visual concepts of Animals and Geometric Patterns. An additional 33 examples can be found in Figures 11 and 13–15 of Appendix C. It is noted that a caution is required when analyzing the results, especially for LeRHF. For the HRV of a given visual concept, the least relevant heads have little effect on the concept of interest but could be significantly relevant to other visual concepts. Therefore, interpreting the changes observed in the LeRHF-generated images requires careful consideration of other visual concepts.

We have also plotted the trends of CLIP image-text similarity in Figure 3b. CLIP similarity was measured between the generated images and the concept-words used in the prompts. For each data point in the CLIP score plots, we used between 30 and 150 images, depending on the specific visual concept. The prompt templates and concept-words used for each visual concept are detailed in Appendix C.1. These plots clearly show that relevant concepts are removed significantly faster during MoRHF weakening, while they are preserved for a longer duration in LeRHF weakening. Additional similarity plots for six more visual concepts can be found in Figure 12 of Appendix C.2. Additionally, we compare HRV-based ordered weakening with random order weakening in Appendix C.3, further supporting HRV's concept-awareness.

## 5   STEERING VISUAL CONCEPTS IN THREE VISUAL GENERATIVE TASKS

Head relevance vectors are not only valuable as interpretable features but can also be used to steer visual concepts in generative tasks. In this section, we demonstrate that polysemous word challenges can be addressed and that leading methods, such as P2P (Hertz et al., 2022) for image editing and Attend-and-Excite (A&E) (Chefer et al., 2023) for multi-concept generation, can be significantly enhanced. All of our experiments are conducted using Stable Diffusion v1.4, 50 timesteps with PNDM sampling (Liu et al., 2022), and classifier-free guidance at a scale of 7.5. All CLIP-based metrics are calculated using the OpenCLIP ViT-H/14 model (Ilharco et al., 2021).

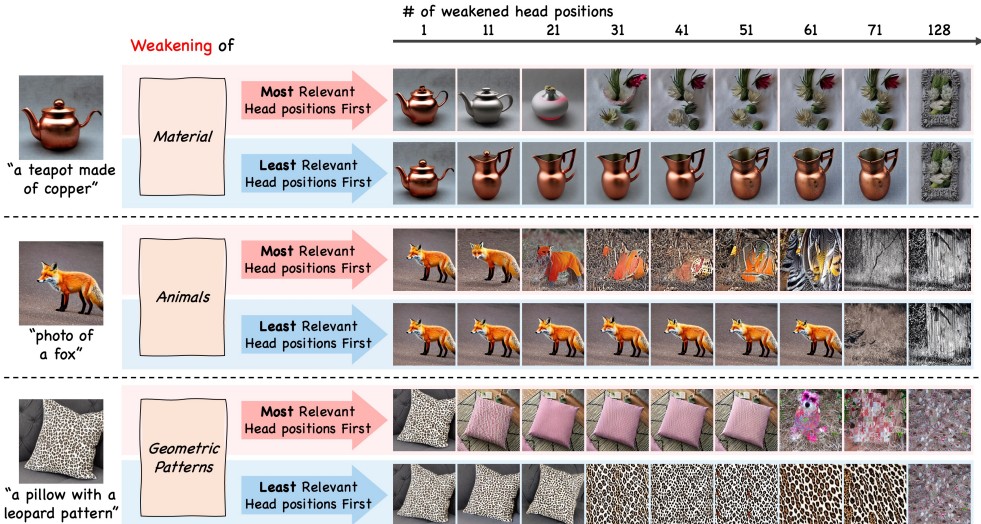

(a) Generated images as weakening progresses in either MoRHF or LeRHF order.

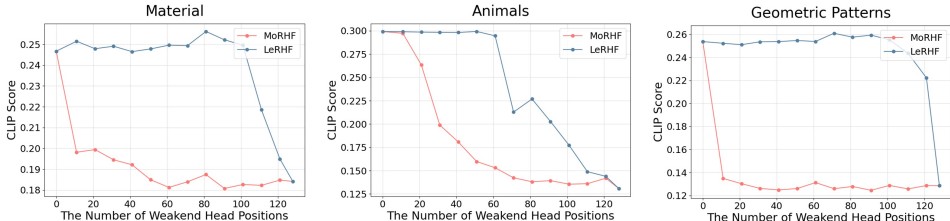

(b) Change in CLIP image-text similarity score as weakening progresses in either MoRHF or LeRHF order.

Figure 3: Ordered weakening analysis for three visual concepts: The visual concept of interest disappears significantly faster with MoRHF, where the most relevant heads in the corresponding HRV are weakened first. Note that 128 corresponds to the weakening of all heads.

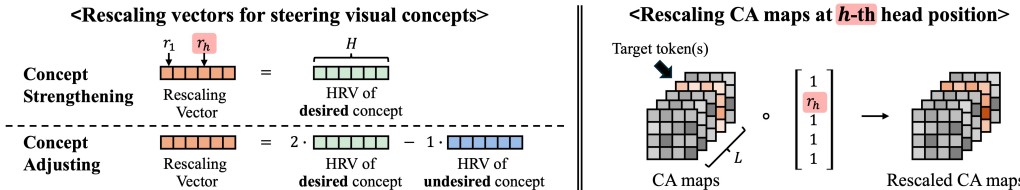

Figure 4: Two rescaling vectors for visual concept steering. **Left:** *Concept strengthening* uses HRV of a desired visual concept as the rescaling vector. *Concept adjusting* combines HRVs of a desired and an undesired visual concepts to define the rescaling vector: $2 \cdot$ (HRV of desired concept) $- 1 \cdot$ (HRV of undesired concept). Here, $H = 128$ denotes the number of CA heads. **Right:** For both concept steering methods, the $h$-th CA map of a target token is rescaled using $r_h$, the $h$-th element of the rescaling vector, where $h = 1, \ldots, H$. Here, $L = 77$ denotes the token length.

**Two rescaling vectors for visual concept steering – concept strengthening and concept adjusting:** We define two rescaling vectors as illustrated in Figure 4, both utilizing pre-constructed HRVs. In Section 3, we have constructed 34 HRVs for the 34 visual concepts listed in Table 3 of Appendix A. We utilize the pre-constructed HRVs to steer the corresponding concepts. For some visual generative tasks, only a desired concept can be identified, and we apply *concept strengthening*. In other tasks, both a desired and an undesired concept can be identified, typically in cases where the generative model fails to meet the user's intention. In such cases, we apply *concept adjusting*. The rescaling vector of concept adjusting is designed to be closely related to that of concept strengthening; the two vectors become equivalent when desired and undesired concepts are identical.

## 5.1 IMAGE GENERATION − REDUCING MISINTERPRETATION OF POLYSEMOUS WORDS

The same word can have different meanings depending on the context. Stable Diffusion (SD) models are known for misinterpreting such polysemous words, often generating images that do not comply with the user's intended meaning. Two examples are shown in Figure 5. SD fails to recognize that 'lavender' clearly refers to a color and 'Apple' to an electronic device. This misinterpretation by SD may arise from limitations in the CLIP text encoder, which has been reported to exhibit bag-of-words issues (Yuksekgonul et al., 2023a). The problem can be resolved by adopting *concept adjusting* to SD as shown in Figure 5, where we name our method as SD-HRV. For the lavender case, SD-HRV resolves this issue by applying con-

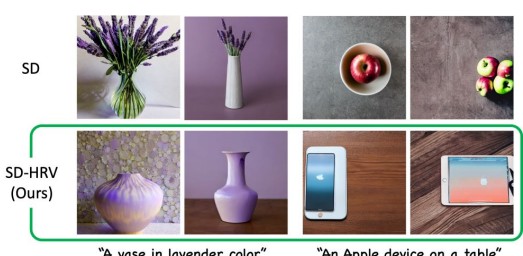

Figure 5: Examples of image generations from Stable Diffusion (SD) and SD-HRV (ours) using prompts frequently misinterpreted by T2I models. SD-HRV effectively reduces misinterpretation compared to SD.

cept adjusting to the token 'lavender,' using Color as the desired visual concept and Plants as the undesired visual concept. For the Apple case, concept adjusting is applied to the token 'Apple,' using Brand Logos as the desired and Fruits as the undesired. Examples of 10 cases, including the 2 cases shown in Figure 5, can be found in Figures 18–19 of Appendix D.1. There, we provide 10 images generated with 10 random seeds for each case. Additionally, we investigated the performance of concept strengthening compared to concept adjusting and found, as expected, that concept adjusting performs better. The details can be found in Appendix D.3.

We have also performed a *human evaluation* over the 10 cases, each with 10 random seeds, and the details can be found in Appendix D.2. Based on the human evaluation, the human perceived misinterpretation rate dropped significantly from 63.0% to 15.9% when SD-HRV was adopted.

## 5.2 IMAGE EDITING − SUCCESSFUL EDITING FOR FIVE CHALLENGING VISUAL CONCEPTS

Image editing involves generating an image that aligns with the target prompt while minimizing structural changes from the source image. Although recently developed methods excel at image editing, certain visual concepts remain challenging to edit. For example, concepts related to materials, geometric patterns, image styles, and weather conditions are known to be particularly difficult to edit. To address this problem, we propose applying *concept strengthening* to P2P (Hertz et al., 2022). We refer to our method as P2P-HRV. The key idea is to apply concept strengthening to the edited token, the token that describes how the attribute of the source image should be changed, thereby strengthening the concept related to the editing target. The detailed explanations of P2P-HRV method are provided in Appendix E.1.

**Experimental settings:** We focus on five challenging visual concepts as editing targets, including three object attributes (Color, Material, and Geometric Patterns) and two image attributes (Image Styles and Weather Conditions). We compare P2P-HRV with SDEdit (Meng et al., 2021), P2P (Hertz et al., 2022), PnP (Tumanyan et al., 2023), MasaCtrl (Cao et al., 2023), and FPE (Liu et al., 2024). For each method, we generate 500 edited images for each editing target (250 for Weather Conditions) using the prompts described in Appendix E.2. For object attributes (Color, Material, and Geometric Patterns), we evaluated performance using both CLIP (Radford et al., 2021) and BG-DINO scores. The CLIP score evaluates CLIP image-text similarity between the edited image and the target prompt. The BG-DINO score assesses structure preservation using Grounded-SAM-2 (Ravi et al., 2024; Ren et al., 2024) for extracting non-object parts from the source and edited images and then comparing them with DINOv2 (Oquab et al., 2023) embeddings. The BG-DINO score is inspired by the *segment-and-embed* metrics used in prior research (Parmar et al., 2023; Kim et al., 2024). For image attributes (Image Styles and Weather Conditions), we conducted a *human evaluation* to assess human preference (HP) scores, as BG-DINO is not well-suited for these edits, which involve modifying the entire image rather than preserving non-object parts. In the human evaluation, we compared P2P-HRV with the other methods by asking 'Which edited image better matches the target description, while maintaining essential details of the source image?' We normalized the HP-score, setting P2P-HRV's preference score to 100.

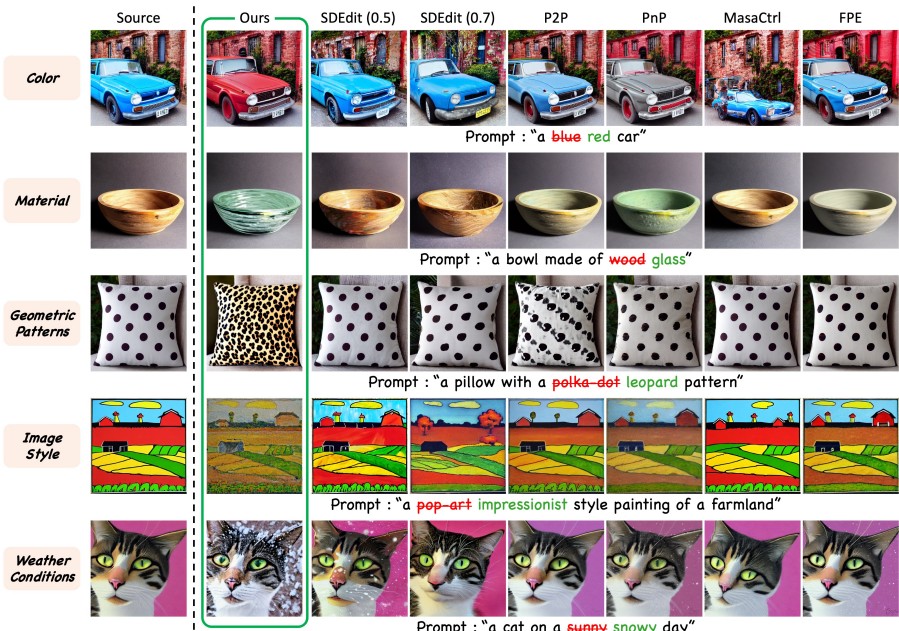

Figure 6: Examples of image editing for the five challenging visual concepts. In these examples, all comparison methods frequently fail to make the desired edits, whereas our P2P-HRV successfully achieves the intended modifications.

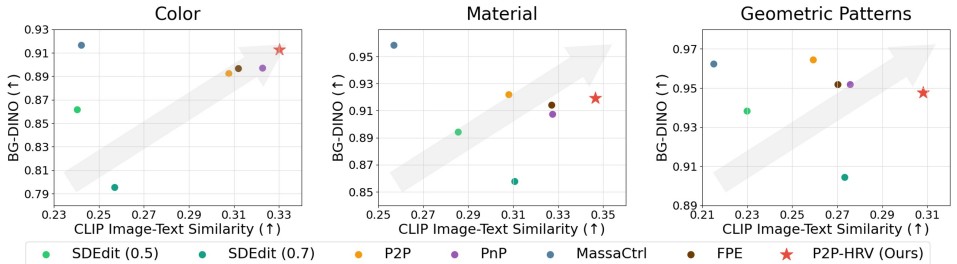

Figure 7: Quantitative comparison of image editing methods for three object attributes using CLIP and BG-DINO scores.

Table 1: CLIP scores and human evaluation scores for two image attributes. The best and second-best results are highlighted in **bold** and underlined, respectively.

| Method | Image Attribute | | | |
|---|---|---|---|---|
| | Image Style | | Weather Conditions | |
| | CLIP | HP-score | CLIP | HP-score |
| SDEdit (0.5) | 0.2938 | - | 0.2817 | - |
| SDEdit (0.7) | 0.3217 | 15.8 | 0.2908 | 39.5 |
| P2P | 0.3120 | 30.6 | 0.2788 | 33.9 |
| PnP | 0.3286 | 41.9 | 0.3046 | 35.0 |
| MassaCtrl | 0.2722 | - | 0.2524 | - |
| FPE | 0.3236 | 25.7 | 0.2962 | 35.5 |
| P2P-HRV (Ours) | **0.3424** | **100** | **0.3348** | **100** |

**Experimental results:** Figure 6 presents five exemplary cases of image editing across the five challenging visual concepts. The figure demonstrates that our approach significantly improves image-text alignment compared to the previously known methods. While only five cases are shown in this figure, an extensive list of additional examples can be found in Figures 24–34 of Appendix E.5. For the editing of three object attributes, CLIP similarity and BG-DINO scores are presented in Figure 7. P2P-HRV achieves Pareto-optimal performance compared to previous SOTA methods across all three object attributes. For the editing of two image attributes, CLIP similarity and human preference performance are presented in Table 1. Our P2P-HRV improves CLIP perfor-

mance by 4.20% on Image Style and 9.91% on Weather Conditions compared to the previous SOTA, PnP. In the human evaluation, we have found that P2P-HRV received 2.39 and 2.53 times more votes in HP-scores for Image Style and Weather Conditions, respectively, compared to the second-best methods.

## 5.3 MULTI-CONCEPT GENERATION – REDUCING CATASTROPHIC NEGLECT

T2I generative models often struggle with multi-concept generation, failing to capture all the specified subjects or attributes in a prompt. To address this issue, known as catastrophic neglect, Attend-and-Excite (A&E) (Chefer et al., 2023) iteratively updates the image latent using gradients derived from the CA maps of selected tokens during image generation. To further improve A&E, we propose incorporating our *concept strengthening* approach, which we refer to as A&E-HRV.

**Experimental settings:** We investigated A&E-HRV using two types of prompts: (i) 'a {*Animal A*} and a {*Animal B*}' (Type 1) and (ii) 'a {*Color A*} {*Animal A*} and a {*Color B*} {*Animal B*}' (Type 2), originally examined in the A&E work (Chefer et al., 2023). Type 1 evaluates multi-object generation, while Type 2 adds the challenge of binding color attributes to each animal. In these experiments, we used 12 animals and 10 colors, as detailed in Appendix F. We assessed 66 prompts for Type 1 and 150 for Type 2, with 30 random seeds applied across all methods. Following A&E, we adopted three evaluation metrics. Full prompt similarity measures the CLIP similarity between the generated image and the full prompt. Minimum object similarity is calculated by measuring the CLIP image-text similarity between the image and two sub-prompts, which are created by splitting the original prompt at 'and.' The lower similarity score between the two sub-prompts is then reported. BLIP-score measures the CLIP text-text similarity between the prompt and the image caption generated with BLIP-2 (Li et al., 2023).

**Experimental results:** Table 2 presents the quantitative comparisons, while Figure 8 shows qualitative results. As shown in Table 2, A&E-HRV outperforms other methods across all metrics and prompt types. In Figure 8, the top row shows results for Type 1 prompt, while the bottom row shows results for Type 2 prompt. The existing methods either neglect key visual concepts or fail to generate realistic images. In contrast, our approach captures all concepts and generates realistic images for both prompt types. Additional comparisons are provided in Figure 36 of Appendix F.2.

Table 2: Quantitative results of multi-concept generation. The best and second-best results are highlighted in **bold** and underlined, respectively. The percentage in parentheses indicates the improvement over the second best result, A&E.

| Method | Type1: a {*Animal A*} and a {*Animal B*} | | | Type2: a {*Color A*} {*Animal A*} and a {*Color B*} {*Animal B*} | | |
|---|---|---|---|---|---|---|
| | Full Prompt | Min. Object | BLIP-score | Full Prompt | Min. Object | BLIP-score |
| Stable Diffusion | 0.3000 | 0.1611 | 0.5934 | 0.3420 | 0.1458 | 0.5633 |
| Structured Diffusion | 0.2831 | 0.1545 | 0.5626 | 0.3307 | 0.1424 | 0.5508 |
| Attend-and-Excite (A&E) | 0.3544 | 0.2017 | 0.7049 | 0.3883 | 0.1972 | 0.6373 |
| A&E-HRV (Ours) | **0.3702** (+4.5%) | **0.2078** (+3.0%) | **0.7491** (+6.3%) | **0.3971** (+2.3%) | **0.2073** (+5.1%) | **0.6580** (+3.2%) |

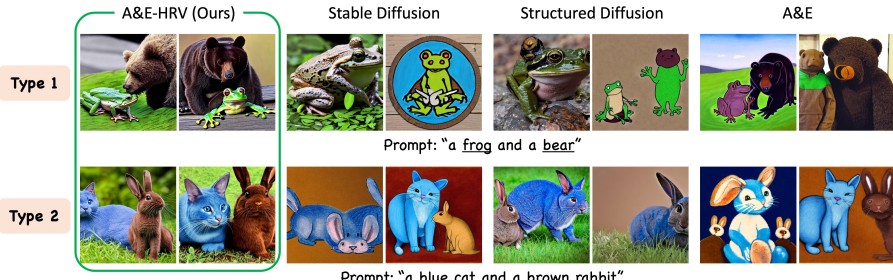

Figure 8: Examples of multi-concept generation for Type 1 and Type 2 prompts. We compare Stable Diffusion (Rombach et al., 2022), Structured Diffusion (Feng et al., 2022), and Attend-and-Excite (Chefer et al., 2023) with ours.

## 6    DISCUSSION

### 6.1    EXTENSIONS TO A LARGER ARCHITECTURE

The recently introduced Stable Diffusion XL (SDXL) (Podell et al., 2024) adopts a U-Net backbone that is three times larger than its predecessors, scaling up to 1300 cross-attention (CA) heads. To investigate the generalization capability of HRV, we have performed an extended study with SDXL. Using SDXL, we conducted the ordered weakening analysis to evaluate whether HRVs can serve as interpretable features. An exemplary result for the visual concept Furniture is shown in Figure 9. The sofa disappeared when the most relevant 211 heads were weakened. In contrast, the sofa remained visible until the least relevant 711 heads were weakened. Additional 45 examples and the similarity plots for nine visual concepts can be found in Figures 37–42 of Appendix G.1. For SDXL experiments, we utilize the SD-XL 1.0-base model.

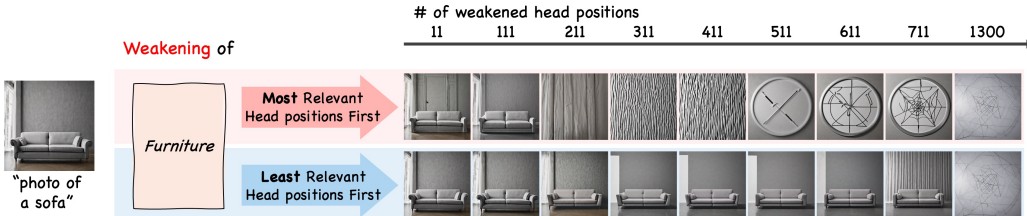

Figure 9: Ordered weakening analysis for SDXL: generated images are shown as weakening progresses in either MoRHF or LeRHF order.

### 6.2    DO GENERATION TIMESTEPS AFFECT HOW HEADS RELATE TO VISUAL CONCEPT?

Diffusion models generate images by iteratively processing an image latent through the same U-Net network. A natural question is whether the patterns of head relevance vectors change across different timesteps during generation. To explore this, we calculated head relevance vectors for each visual concept at every timestep, resulting in 1700 vectors (34 visual concepts×50 timesteps). Figure 10 presents a t-SNE (Van der Maaten & Hinton, 2008) plot of these 1700 vectors. In the t-SNE plot, visual concepts are clearly separated, while timesteps are not. This indicates that generation timesteps do not significantly affect the head relevance patterns of each visual concept. Further analysis, including cosine similarity plots, is provided in Appendix I.

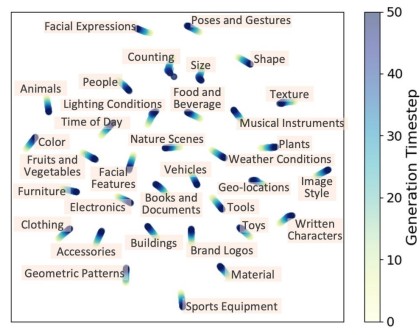

Figure 10: t-SNE plot of 1700 head relevance vectors across 34 visual concepts and 50 generation timesteps.

### 6.3    LIMITATION

We examined the 34 concepts listed in Table 3 of Appendix A and identified two types of failure cases. The first type arises from limitations in the underlying T2I model, while the second is related to our proposed algorithm or the concept-words used to represent the concept. Appendix H provides a detailed discussion of these failure cases, along with examples for each, shown in Figures 45-46.

## 7    CONCLUSION

In this work, we present findings from our exploration of cross-attention heads in T2I models. We demonstrate that head relevance vectors (HRVs) can be effectively and reliably constructed for human-specified visual concepts without requiring any modifications or fine-tuning of the T2I model. Furthermore, we show that HRVs can be successfully applied to improve performance of three visual generative tasks. Our work provides an advancement in understanding cross-attention layers and introduces novel approaches for exploiting these layers at the head-level.

## 8 REPRODUCIBILITY STATEMENT

We provide our core codebase in our code repository, which includes the methodology implementation, settings, generation prompts, and benchmarks for image editing and multi-concept generation.

## 9 ETHIC STATEMENT

Our work presents new techniques for controlling and refining text-to-image diffusion models, enhancing both image generation and editing capabilities. While these advancements hold significant potential for creative and practical applications, they also raise ethical concerns about the possible misuse of generative models, such as creating manipulated media for disinformation. It is important to recognize that any image editing or generation tool can be used for both positive and negative purposes, making responsible use essential. Fortunately, various research in detecting harmful content and preventing malicious editing are making significant progress. We believe our detailed analysis of cross-attention heads will contribute to these efforts by providing a deeper understanding of the mechanisms behind the text-to-image generative models.

## ACKNOWLEDGEMENTS

This work was partly supported by a National Research Foundation of Korea (NRF) grant funded by the Korea government (MSIT) (No. NRF-2020R1A2C2007139) and by Institute of Information & communications Technology Planning & Evaluation (IITP) grant funded by the Korea government (MSIT) ([NO.RS-2021-II211343, Artificial Intelligence Graduate School Program (Seoul National University)], [No. RS-2023-00235293, Development of autonomous driving big data processing, management, search, and sharing interface technology to provide autonomous driving data according to the purpose of usage]).

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

CONTENTS

# A 34 VISUAL CONCEPTS AND FULL LIST OF CONCEPT-WORDS

In this paper, we use 34 visual concepts, each paired with 10 concept-words, as shown in Table 3.

Table 3: 34 visual concepts and full list of concept-words.

| Visual Concept | Concept-words |
| --- | --- |
| Color | red, blue, green, yellow, black, white, purple, gray, pink, brown |
| Animals | dog, cat, elephant, lion, bird, fish, butterfly, bear, horse, cow |
| Plants | tree, flower, grass, bush, cactus, vine, oak tree, moss, tulip, rose |
| Fruits and Vegetables | apple, banana, carrot, tomato, broccoli, strawberry, potato, orange, lettuce, grapes |
| People | child, woman, man, elderly person, teenager, athlete, farmer, doctor, teacher, tourist |
| Vehicles | car, bicycle, airplane, train, boat, motorcycle, bus, truck, scooter, helicopter |
| Buildings | skyscraper, house, church, library, museum, school, hospital, warehouse, apartment, castle |
| Furniture | chair, table, sofa, bed, bookshelf, desk, wardrobe, dresser, cabinet, coffee table |
| Electronics | smartphone, laptop, television, camera, tablet, headphones, microwave, refrigerator, printer, smartwatch |
| Clothing | t-shirt, jeans, jacket, dress, skirt, sweater, shorts, coat, blouse, pants |
| Accessories | hat, scarf, sunglasses, watch, belt, necklace, earrings, bracelet, gloves, handbag |
| Tools | hammer, screwdriver, wrench, drill, saw, pliers, tape measure, chisel, level, shovel |
| Toys | doll, teddy bear, puzzle, action figure, toy car, lego, ball, kite, yo-yo, board game |
| Food and Beverages | pizza, salad, burger, pasta, soup, coffee, tea, juice, cake, sandwich |
| Books and Documents | novel, dictionary, textbook, magazine, newspaper, journal, notebook, manual, report, brochure |
| Sports Equipment | soccer ball, tennis racket, basketball, baseball glove, hockey stick, golf club, surfboard, ski poles, boxing gloves, cricket bat |
| Musical Instruments | guitar, piano, violin, drum, flute, saxophone, trumpet, harp, cello, clarinet |
| Written Characters | latin alphabet, greek alphabet, english alphabet, punctuation marks, mathematical symbols, uppercase letters, lowercase letters, digits, consonants, vowels |
| Shape | circle, square, triangle, rectangle, oval, hexagon, octagon, star, heart, diamond |
| Texture | smooth, rough, fuzzy, slippery, sticky, soft, hard, bumpy, grainy, glossy |
| Size | small, medium, large, tiny, huge, miniature, gigantic, petite, bulky, massive |
| Counting | one, two, three, four, five, six, seven, eight, nine, ten |
| Image style | abstract, realistic, minimalist, cartoon, vintage, modern, black and white, sepia, surreal, pop art |
| Material | wood, metal, plastic, glass, fabric, paper, stone, rubber, leather, ceramic |
| Lighting Conditions | bright, dim, dark, natural light, artificial light, backlit, soft light, harsh light, spotlight, twilight |
| Facial Expressions | happy, sad, angry, surprised, confused, laughing, crying, smiling, frowning, disgusted |
| Facial Features | eyes, nose, mouth, ears, eyebrows, lips, cheeks, chin, forehead, teeth |
| Poses and Gestures | standing, sitting, running, jumping, waving, clapping, pointing, hugging, dancing, kneeling |
| Nature Scenes | forest, beach, mountain, river, desert, meadow, waterfall, jungle, lake, canyon |
| Weather Conditions | windy, sunny, cloudy, stormy, foggy, rainy, snowy, hail, thunderstorm, drizzle |
| Time of Day | morning, noon, afternoon, evening, night, sunrise, sunset, midnight, dusk, dawn |
| Geo-locations | New York, Beijing, Paris, Tokyo, London, Sydney, Cairo, Rome, Moscow, Rio de Janeiro |
| Brand Logos | Nike, McDonald's, Coca-Cola, Google, Microsoft, Amazon, Adidas, Samsung, Starbucks, BMW |
| Geometric Patterns | stripes, polka dots, chevron, plaid, argyle, houndstooth, lattice, hexagons, spirals, waves |

# B   DETAILS OF HRV CONSTRUCTION

## B.1   PSEUDO-CODE FOR HRV CONSTRUCTION

---
**Algorithm 1** HRV construction

---
**Require:** $N$: Number of human-specified visual concepts
**Require:** $T$: Total number of generation timesteps
**Require:** $H$: Total number of CA heads
**Require:** $\mathbb{P}$: Set of prompts for random image generation
**Require:** $\mathbb{S}$: Set of concept-words covering $N$ visual concepts
**Require:** $\psi$: CLIP text-encoder
**Require:** $l_K^{(h)}$: Key projection layer for the $h$-th cross-attention (CA) head
**Require:** $\xi$: Function to extract for semantic token embeddings
**Require:** $\mathbf{Q}^{(t,h)}$: Image query matrix at timestep $t$ and the $h$-th CA head
 1: Initialize HRV matrix $\mathbf{V}$ as a zero matrix $\mathbf{0} \in \mathbb{R}^{N \times H}$
 2: **for** each prompt $P \in \mathbb{P}$ **do**
 3:    **while** generating a random image with prompt $P$ **do**
 4:      **for** all $t = 1, 2, \ldots, T$ **do**
 5:        **for** all $h = 1, 2, \ldots, H$ **do**
 6:          **for** all $n = 1, 2, \ldots, N$ **do**
 7:            Sample a concept-word $\mathbf{W}_n$ for visual concept $C_n$ from $\mathbb{S}$
 8:            Compute key-projected embedding of $\mathbf{W}_n$: $\mathbf{K}_n = l_K^{(h)}(\psi(\mathbf{W}_n)) \in \mathbb{R}^{77 \times F}$
 9:            Extract semantic token embeddings: $\widehat{\mathbf{K}}_n = \xi(\mathbf{K}_n)$
10:          **end for**
11:          Concatenate $\widehat{\mathbf{K}}_1, \widehat{\mathbf{K}}_2, \ldots, \widehat{\mathbf{K}}_N$ along the token dimension:

$$\widehat{\mathbf{K}} = [\widehat{\mathbf{K}}_1, \widehat{\mathbf{K}}_2, \ldots, \widehat{\mathbf{K}}_N] \in \mathbb{R}^{N' \times F} \tag{2}$$

12:          Calculate the CA map $\widehat{\mathbf{M}}$ using $\mathbf{K} = \widehat{\mathbf{K}}$ and $\mathbf{Q} = \mathbf{Q}^{(t,h)} \in \mathbb{R}^{R^2 \times F}$:

$$\widehat{\mathbf{M}} = \mathrm{softmax}\left(\frac{\mathbf{Q}^{(t,h)} \cdot \widehat{\mathbf{K}}^T}{\sqrt{d}}\right) \in \mathbb{R}^{R^2 \times N'} \tag{3}$$

13:          Average $\widehat{\mathbf{M}}$ along the token dimension for multi-token concept-words, resulting in a matrix of shape $\mathbb{R}^{R^2 \times N}$
14:          Average the resulting matrix over the spatial dimension ($R^2$), producing $\widetilde{\mathbf{M}} \in \mathbb{R}^N$
15:          Apply an argmax operation over the token dimension ($N$):

$$\widetilde{\mathbf{M}} \leftarrow \mathrm{argmax}(\widetilde{\mathbf{M}}) \tag{4}$$

16:          Update the $h$-th column of the HRV matrix $\mathbf{V}$ by adding $\widetilde{\mathbf{M}}$:

$$\mathbf{V}[:, h] \leftarrow \mathbf{V}[:, h] + \widetilde{\mathbf{M}} \tag{5}$$

17:        **end for**
18:      **end for**
19:    **end while**
20: **end for**
21: **Return:** HRV matrix $\mathbf{V} \in \mathbb{R}^{N \times H}$

---

B.2 ROLE OF THE ARGMAX OPERATION IN HRV CONSTRUCTION

During HRV construction, we apply the argmax operation to the averaged CA maps before using them to update the HRV matrix (see Eq. 4). This step addresses the varying representation scales across $H$ CA heads. To demonstrate these scale differences, we compute the averaged L1-norm of the CA maps before applying the softmax operation (refer to Eq. 3 for notations):

$$\mathbf{L}^{(t,h)} = \frac{1}{R^2 \cdot N'} \cdot \sum_{R^2,N'} \left\| \left( \frac{\mathbf{Q}^{(t,h)} \cdot \widehat{\mathbf{K}}^T}{\sqrt{d}} \right) \right\|_1 \tag{6}$$

In Table 4, we show the mean and standard deviation of $\mathbf{L}^{(t,h)}$ across 2100 generation prompts and 50 timesteps for each CA head in Stable Diffusion v1.4. The CA heads exhibit variation in their representation scales, with the head having the largest scale showing a mean value 8.1 times higher than that of the smallest scale. Since the softmax operation maps large-scale values closer to a Dirac-delta distribution and small-scale values closer to a uniform distribution, it is necessary to align the scales between CA heads before accumulating the information into the HRV matrix. We achieve this by simply applying the argmax operation, as shown in Eq. 4, which resolves the issue of differing representation scales across the CA heads.

As explained in the previous paragraph, the softmax operation introduces an imbalance when summing without the argmax operation, where CA heads with larger scales produce vectors closer to a Dirac-delta distribution, while those with smaller scales produce vectors closer to a uniform distribution. This imbalance favors CA heads with larger representation scales, leading to an overemphasis on the largest concept chosen by these larger-scale heads compared to the largest concept chosen by smaller-scale CA heads. For example, as shown in Table 4, the largest concept chosen by the CA head at [Layer 15-Head 4] would be overemphasized compared to the largest concept chosen by the CA head at [Layer 16-Head 1]. A similar issue arises when using the max operation instead of argmax, as the maximum values from larger-scale CA heads are much larger than those from smaller-scale CA heads. To address this, we apply the argmax operation before summation, ensuring that the largest concept from each CA head contributes a value of 1 to the HRV matrix, regardless of its scale. This straightforward approach eliminates the bias toward larger-scale CA heads.

Table 4: Mean and standard deviation of the averaged L1-norm, $\mathbf{L}^{(t,h)}$, of CA maps before applying the softmax operation. The statistics are calculated over 2100 generation prompts and 50 timesteps. The largest and smallest mean values are highlighted in **bold**.

|          | Head 1 | Head 2 | Head 3 | Head 4 | Head 5 | Head 6 | Head 7 | Head 8 |
|----------|--------|--------|--------|--------|--------|--------|--------|--------|
| **Layer 1**  | $1.16 \pm 0.16$ | $1.58 \pm 0.19$ | $1.08 \pm 0.26$ | $1.45 \pm 0.18$ | $1.73 \pm 0.36$ | $1.75 \pm 0.39$ | $1.89 \pm 1.26$ | $0.92 \pm 0.22$ |
| **Layer 2**  | $1.24 \pm 0.21$ | $1.19 \pm 0.29$ | $1.30 \pm 0.24$ | $1.43 \pm 0.20$ | $0.99 \pm 0.34$ | $1.08 \pm 0.24$ | $0.89 \pm 0.12$ | $1.07 \pm 0.28$ |
| **Layer 3**  | $1.48 \pm 0.12$ | $2.42 \pm 0.35$ | $1.33 \pm 0.14$ | $1.95 \pm 0.38$ | $2.25 \pm 0.47$ | $1.69 \pm 0.29$ | $1.51 \pm 0.20$ | $1.69 \pm 0.38$ |
| **Layer 4**  | $1.79 \pm 0.25$ | $2.10 \pm 0.32$ | $1.10 \pm 0.28$ | $1.13 \pm 0.14$ | $1.39 \pm 0.16$ | $2.18 \pm 0.70$ | $1.57 \pm 0.24$ | $1.14 \pm 0.16$ |
| **Layer 5**  | $1.38 \pm 0.16$ | $1.55 \pm 0.20$ | $1.57 \pm 0.36$ | $1.44 \pm 0.18$ | $1.35 \pm 0.21$ | $1.46 \pm 0.33$ | $1.72 \pm 0.48$ | $1.30 \pm 0.28$ |
| **Layer 6**  | $1.65 \pm 0.20$ | $1.87 \pm 0.56$ | $1.41 \pm 0.27$ | $1.50 \pm 0.17$ | $2.30 \pm 0.29$ | $2.05 \pm 0.63$ | $1.65 \pm 0.21$ | $1.84 \pm 0.62$ |
| **Layer 7**  | $1.64 \pm 0.52$ | $1.79 \pm 0.70$ | $1.30 \pm 0.19$ | $1.34 \pm 0.22$ | $2.37 \pm 0.36$ | $2.37 \pm 0.30$ | $3.04 \pm 1.33$ | $1.90 \pm 0.54$ |
| **Layer 8**  | $1.24 \pm 0.14$ | $2.06 \pm 0.26$ | $1.53 \pm 0.20$ | $1.64 \pm 0.19$ | $1.28 \pm 0.21$ | $1.66 \pm 0.22$ | $1.82 \pm 0.20$ | $2.14 \pm 0.31$ |
| **Layer 9**  | $2.10 \pm 0.26$ | $2.19 \pm 0.29$ | $1.74 \pm 0.22$ | $2.10 \pm 0.25$ | $1.56 \pm 0.21$ | $1.32 \pm 0.15$ | $1.61 \pm 0.19$ | $2.00 \pm 0.28$ |
| **Layer 10** | $1.89 \pm 0.23$ | $1.14 \pm 0.12$ | $1.47 \pm 0.18$ | $2.17 \pm 0.21$ | $1.39 \pm 0.16$ | $1.50 \pm 0.19$ | $2.12 \pm 0.21$ | $1.95 \pm 0.29$ |
| **Layer 11** | $2.19 \pm 0.20$ | $1.64 \pm 0.19$ | $2.38 \pm 0.25$ | $2.15 \pm 0.22$ | $2.36 \pm 0.26$ | $2.37 \pm 0.38$ | $2.18 \pm 0.21$ | $2.15 \pm 0.29$ |
| **Layer 12** | $1.26 \pm 0.13$ | $2.36 \pm 0.62$ | $1.49 \pm 0.21$ | $1.49 \pm 0.15$ | $1.44 \pm 0.18$ | $1.78 \pm 0.33$ | $1.26 \pm 0.14$ | $2.10 \pm 0.28$ |
| **Layer 13** | $1.46 \pm 0.21$ | $1.40 \pm 0.25$ | $1.65 \pm 0.22$ | $1.44 \pm 0.18$ | $1.66 \pm 0.47$ | $1.51 \pm 0.22$ | $1.47 \pm 0.22$ | $1.27 \pm 0.16$ |
| **Layer 14** | $1.17 \pm 0.30$ | $1.19 \pm 0.34$ | $1.59 \pm 0.49$ | $2.10 \pm 1.05$ | $0.98 \pm 0.29$ | $1.51 \pm 0.34$ | $1.02 \pm 0.47$ | $2.35 \pm 0.55$ |
| **Layer 15** | $1.20 \pm 0.38$ | $2.16 \pm 0.46$ | $1.88 \pm 0.40$ | $\mathbf{4.11} \pm 1.65$ | $1.62 \pm 0.39$ | $0.76 \pm 0.13$ | $1.84 \pm 0.28$ | $1.48 \pm 0.37$ |
| **Layer 16** | $\mathbf{0.51} \pm 0.04$ | $1.80 \pm 0.33$ | $1.14 \pm 0.31$ | $1.84 \pm 0.33$ | $0.91 \pm 0.38$ | $1.15 \pm 0.18$ | $1.17 \pm 0.11$ | $1.06 \pm 0.21$ |

## C   Details and additional results on ordered weakening analysis

In this section, we present the generation prompts used for the ordered weakening analysis introduced in Section 4. Additionally, we provide MoRHF and LeRHF plots for 6 more visual concepts, along with more detailed qualitative results.

### C.1   Prompts used for ordered weakening analysis

We conducted the ordered weakening analysis across 9 visual concepts: *Animals*, *Color*, *Fruits and Vegetables*, *Furniture*, *Geometric Patterns*, *Image Style*, *Material*, *Nature Scenes*, and *Weather conditions*. The prompt templates and words for each concept are listed in Table 5. For each prompt, we use 3 random seeds to generate images. For instance, the concept *Color* used the prompt template "a {*Color*} {*Objects A*}" with 3 random seeds, covering 10 colors and 5 objects. This results in 150 generated images per data point in the line plot shown in Figure 3b of the manuscript.

Table 5: Prompt and word list for ordered weakening analysis. Visual concepts marked with an asterisk($^*$) use words that do not overlap with the concept-word list in Table 3.

| Visual Concept | Prompt Template / Words |
|---|---|
| Animals* | *photo of a {**Animals**}* |
| | • {**Animals**}: rabbit, frog, sheep, pig, chicken, dolphin, goat, duck, deer, fox |
| Color* | *a {**Color**} {Objects}* |
| | • {**Color**}: coral, beige, violet, cyan, magenta, indigo, orange, turquoise, teal, khaki |
| | • {*Objects*}: car, bench, bowl, balloon, ball |
| Fruits and Vegetables* | *photo of {**Fruits and Vegetables**}* |
| | • {**Fruits and Vegetables**}: lemons, blueberries, onions, raspberries, pineapples, cherries, cucumbers, bell peppers, cauliflowers, mangoes |
| Furniture | *photo of a {**Furniture**}* |
| | • {**Furniture**}: bed, table, chair, sofa, recliner, bookshelf, dresser, wardrobe, coffee table, TV stand |
| Geometric Patterns | *a {Objects} with a {**Geometric Patterns**} pattern* |
| | • {*Objects*}: T-shirt, pillow, wallpaper, umbrella, blanket |
| | • {**Geometric Patterns**}: polka-dot, leopard, stripe, greek-key, plaid |
| Image Style | *a {**Image Style**} style painting of a {Landscapes}* |
| | • {**Image Style**}: cubist, pop art, steampunk, impressionist, black-and-white, watercolor, cartoon, minimalist, sepia, sketch |
| | • {*Landscapes*}: castle, mountain, cityscape, farmland, forest |
| Material* | *a {Objects} made of {**Material**}* |
| | • {*Objects*}: bowl, cup, table, ball, teapot |
| | • {**Material**}: copper, marble, jade, gold, basalt, silver, clay, steel, tin, bronze |
| Nature Scenes* | *photo of a {**Nature Scenes**}* |
| | • {**Nature Scenes**}: glacier, coral reef, swamp, pond, fjord, rainforest, grassland, marsh, creek, island |
| Weather Conditions | *a {Animals} on a {**Weather Conditions**}* |
| | • {*Animals*}: cat, dog, rabbit, frog, bird |
| | • {**Weather Conditions**}: snowy, rainy, foggy, stormy |

## C.2   ADDITIONAL RESULTS ON ORDERED WEAKENING ANALYSIS

Figure 11 presents the randomly selected examples of the ordered weakening analysis for 6 additional visual concepts: *Animals*, *Fruits and Vegetables*, *Furniture*, *Material*, *Nature Scenes*, and *Weather Conditions*. The MoRHF weakening rapidly removes concept-relevant content, whereas the LeRHF weakening either preserves the original image longer or removes irrelevant content first. Figure 12 shows changes in CLIP image-text similarity scores for MoRHF and LeRHF weakening across these six visual concepts, showing consistent trends. Overall, these results demonstrate how the head relevance vector (HRV) effectively prioritizes heads based on their relevance to each visual concept. Additional qualitative examples are provided in Figure 13–15.

## C.3   COMPARISON WITH RANDOM ORDER WEAKENING

As an additional analysis, we compare HRV-based ordered weakening with random order weakening by calculating the area between the LeRHF and MoRHF line plots. The definitions of MoRHF and LeRHF for HRV-based ordered weakening are provided in Section 4. For random order weakening, the $H$ cross-attention heads are first ordered randomly, and then MoRHF is defined as the first-to-last order and LeRHF as the last-to-first order based on this random ordering. A larger (LeRHF − MoRHF) area indicates that the ordering of CA heads better reflects the relevance of the corresponding concept. Table 6 compares HRV-based ordered weakening and three random weakening approaches across six visual concepts. The results show that HRV-based ordered weakening achieves a higher (LeRHF − MoRHF) area, demonstrating its effectiveness in ordering heads based on their relevance to the given concept.

Table 6: Comparison of (LeRHF − MoRHF) areas between HRV-based ordered weakening and three random weakening cases across six visual concepts. Larger values indicate better alignment of CA head ordering with the relevance of the corresponding concept. *Random Order - Mean* represents the average value across the three random order cases. The highest value for each concept is highlighted in **bold**.

|  | Material | Geometric Patterns | Furniture | Image Style | Color | Animals | Average |
|---|---|---|---|---|---|---|---|
| Ours (HRV) | **6.63** | **14.75** | **14.42** | **9.46** | **7.33** | **8.13** | **10.12** |
| Random Order - Case 1 | -1.94 | 4.29 | -5.48 | -1.81 | -1.83 | 3.02 | -0.63 |
| Random Order - Case 2 | 5.85 | 0.14 | 8.38 | -1.89 | -3.39 | 0.19 | 1.55 |
| Random Order - Case 3 | 1.68 | -3.61 | 2.99 | 2.20 | 5.33 | -2.91 | 0.95 |
| Random Order - Mean | 1.86 | 0.27 | 1.96 | -0.50 | 0.04 | 0.10 | 0.62 |

## C.4   ORDERED RESCALING WITH VARIED RESCALING FACTORS

In the ordered weakening analysis, we selected $-2$ as the rescaling factor. This choice is inspired by P2P-rescaling (Hertz et al., 2022), which uses factors in the range of $[-2, 2]$ to adjust the CA maps of the U-Net for image editing. To explore the impact of different rescaling factors, we present two examples in Figures 16-17. A rescaling factor of 1 leaves the original image generation process unchanged, while factors greater than 1 strengthen the concept and factors smaller than 1 weaken it. Strengthening produces minimal changes, likely because the concept is already present in the image. Weakening works effectively with factors below 0, with stronger effects observed as the factor decreases further.

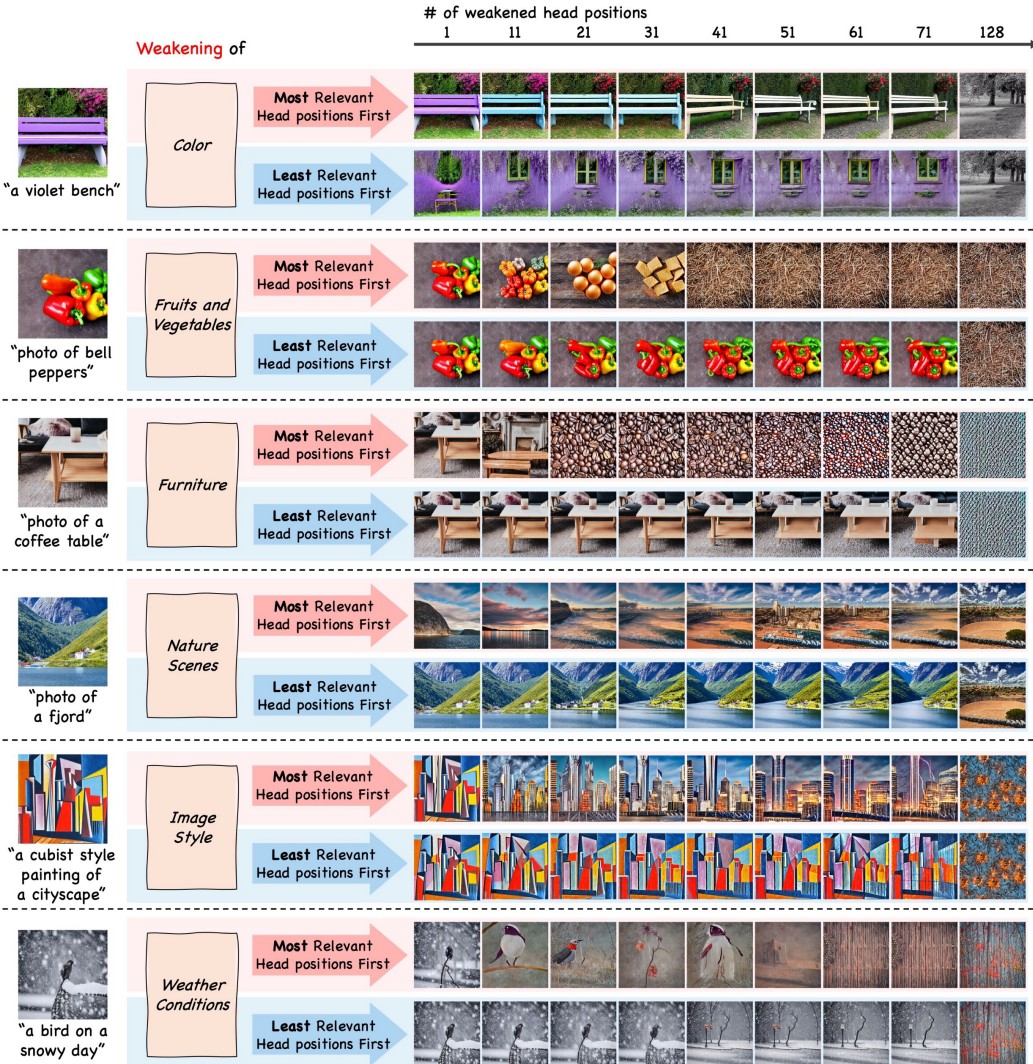

Figure 11: Ordered weakening analysis of six additional concepts: qualitative results using Stable Diffusion v1.

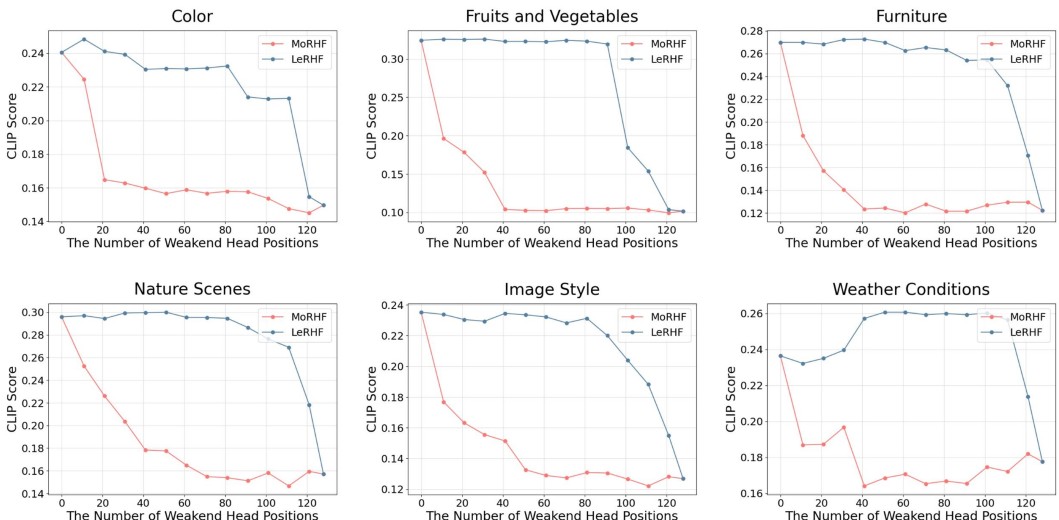

Figure 12: Ordered weakening analysis of six additional concepts: quantitative results using Stable Diffusion v1.

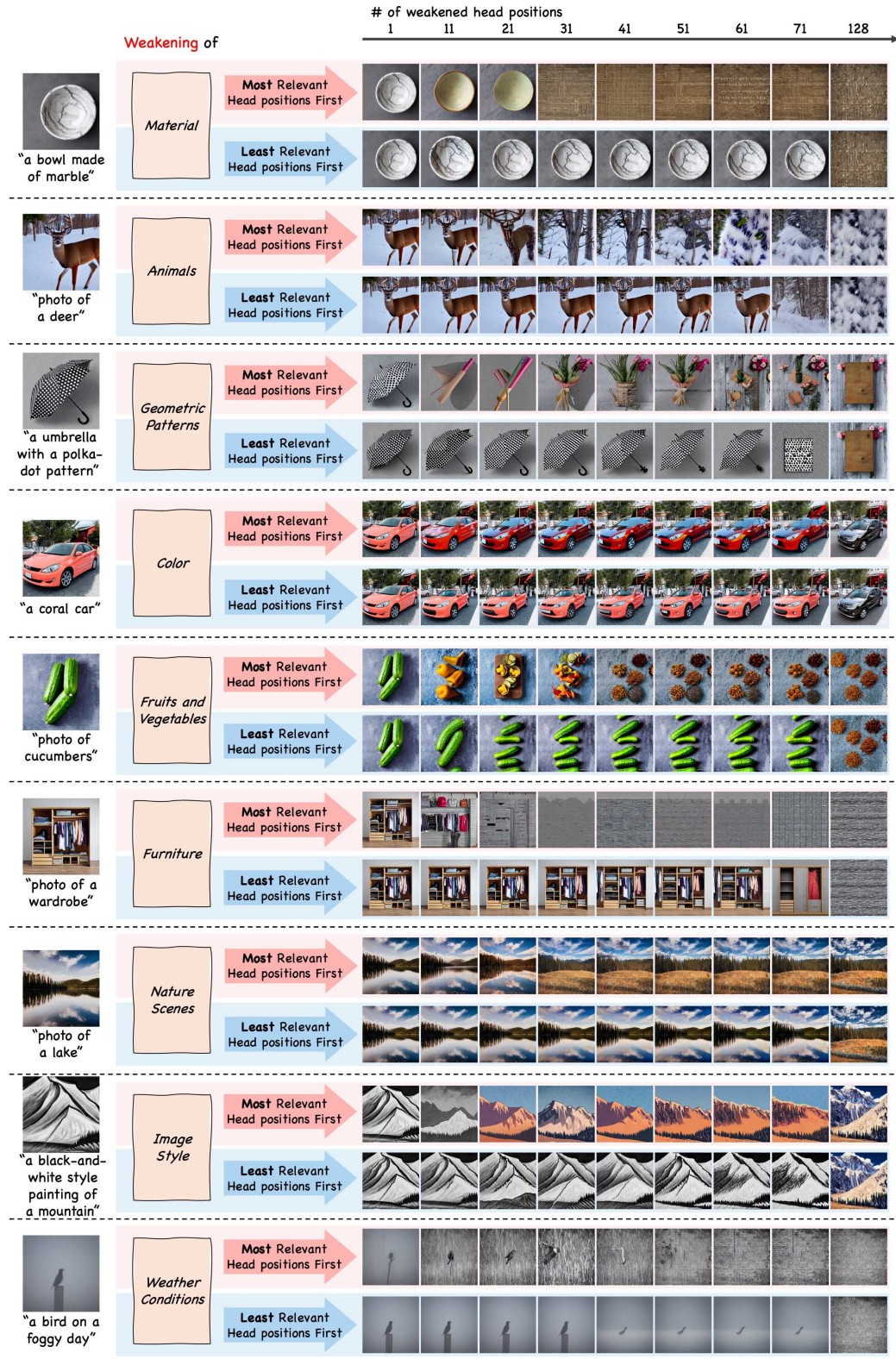

Figure 13: Ordered weakening analysis of nine concepts with additional examples: qualitative results using Stable Diffusion v1 (Part 1 of 3).

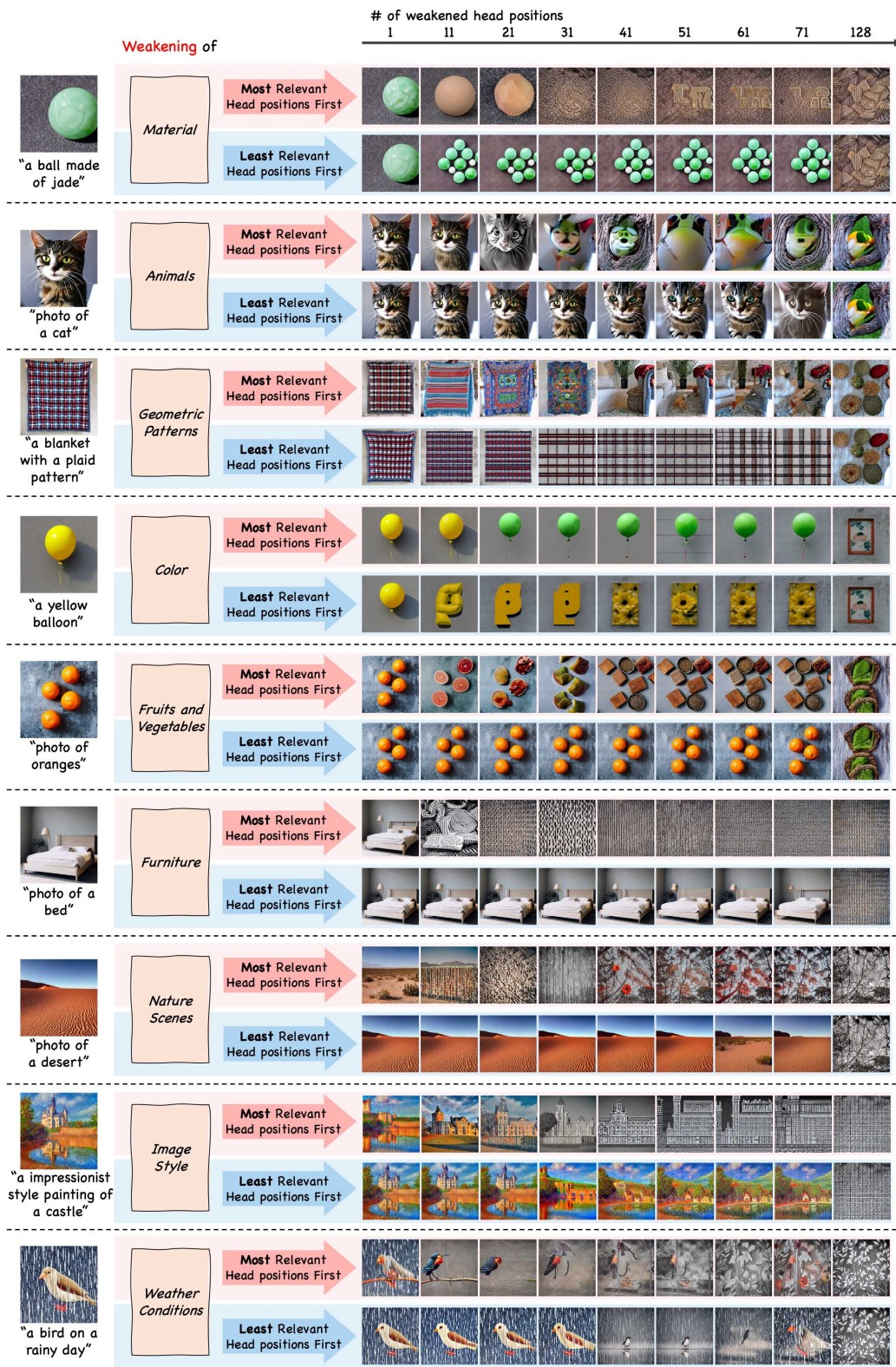

Figure 14: Ordered weakening analysis of nine concepts with additional examples: qualitative results using Stable Diffusion v1 (Part 2 of 3).

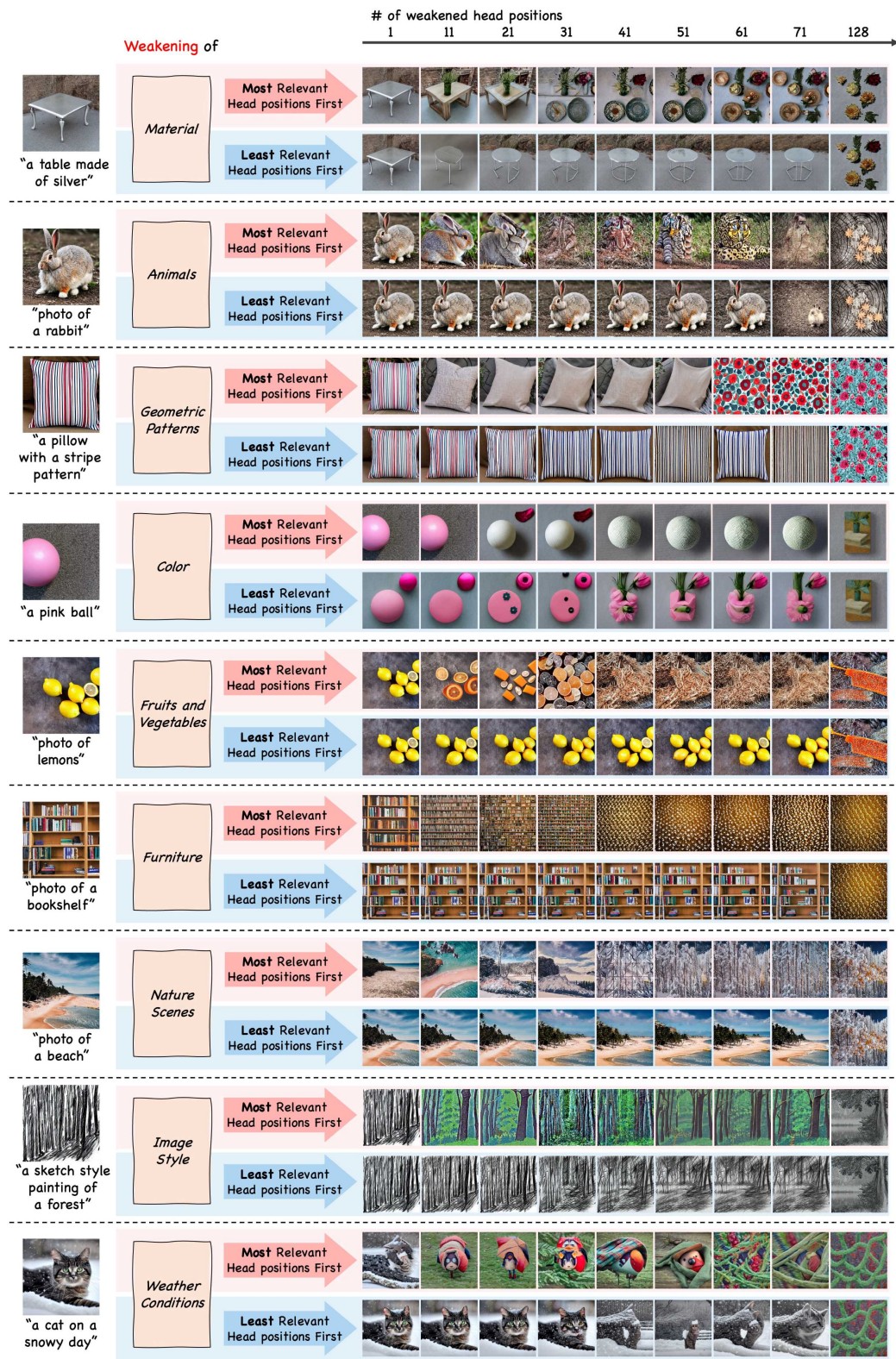

Figure 15: Ordered weakening analysis of nine concepts with additional examples: qualitative results using Stable Diffusion v1 (Part 3 of 3).

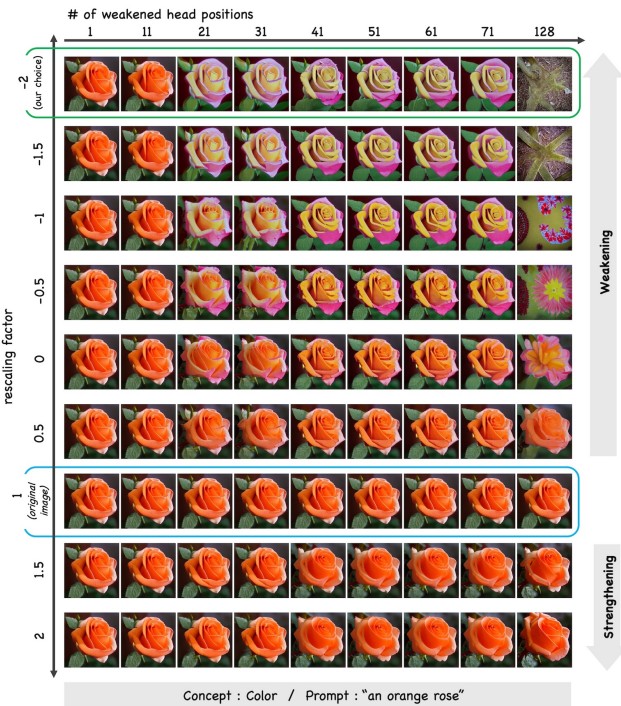

Figure 16: Ordered rescaling with varying rescaling factors, using HRV for the *Color* concept and the generation prompt 'an orange rose.' As the rescaling factor decreases, the weakening effect becomes more pronounced.

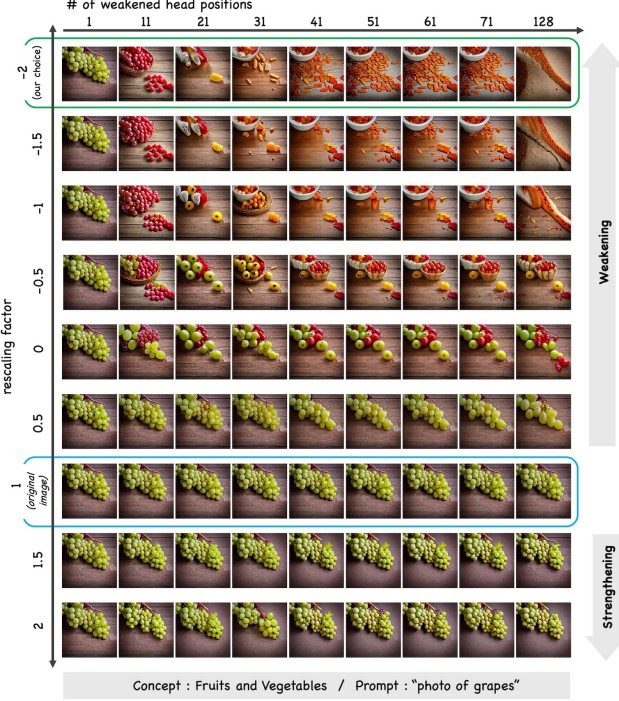

Figure 17: Ordered rescaling with varying rescaling factors, using HRV for the *Fruits and Vegetables* concept and the generation prompt 'photo of grapes.' As the rescaling factor decreases, the weakening effect becomes more pronounced.

# D DETAILS ON REDUCING MISINTERPRETATION OF POLYSEMOUS WORDS

## D.1 PROMPTS AND SELECTED CONCEPTS FOR REDUCING MISINTERPRETATION

We identified 10 prompts that the text-to-image (T2I) generative model frequently misinterprets and carefully selected desired and undesired concepts from our 34 visual concepts to help reduce these misinterpretations. Table 7 lists these 10 prompts, along with the desired and undesired concepts for each polysemous word. For both Stable Diffusion (SD) and SD-HRV, we generated 100 images using these 10 prompts with 10 random seeds. The full set of generated images is shown in Figures 18 and 19. We categorized the misinterpretation into three types: (i) containing the undesired meaning, (ii) missing the desired meaning, and (iii) both, and mark the images showing any of these misinterpretations. For the last prompt, 'A single rusted nut,' where 'nut' was misinterpreted as *Food and Beverages* instead of *Tools*, SD-HRV only partially resolved the issue by removing 'nut' as *Food and Beverages* but failed to generate it as *Tools*. This suggests that SD-HRV is not perfect, and there is still room for improvement in addressing such misinterpretations. Our current implementation for SD-HRV requires manual settings for the target token, as well as the desired and undesired concepts. However, our tests with an LLM show that it effectively identifies the inputs needed for SD-HRV, suggesting that constructing an automatic pipeline using LLMs is feasible.

Table 7: The list of prompts often misinterpreted, with polysemous words underlined.

| Prompt | Desired Concept | Undesired Concept |
|---|---|---|
| A vase in lavender color | Color | Plants |
| An Apple device on a table | Brand Logos | Fruits and Vegetables |
| A rose-colored vase | Color | Plants |
| A single orange-colored plate | Color | Fruits and Vegetables |
| A crane flying over a grass field | Animals | Tools |
| A bowl in mint color on a table | Color | Fruits and Vegetables |
| An olive-colored plate on a table | Color | Fruits and Vegetables |
| A plum-colored bowl on a table | Color | Fruits and Vegetables |
| An apricot-colored bowl on a table | Color | Fruits and Vegetables |
| A single rusted nut | Tools | Food and Beverages |

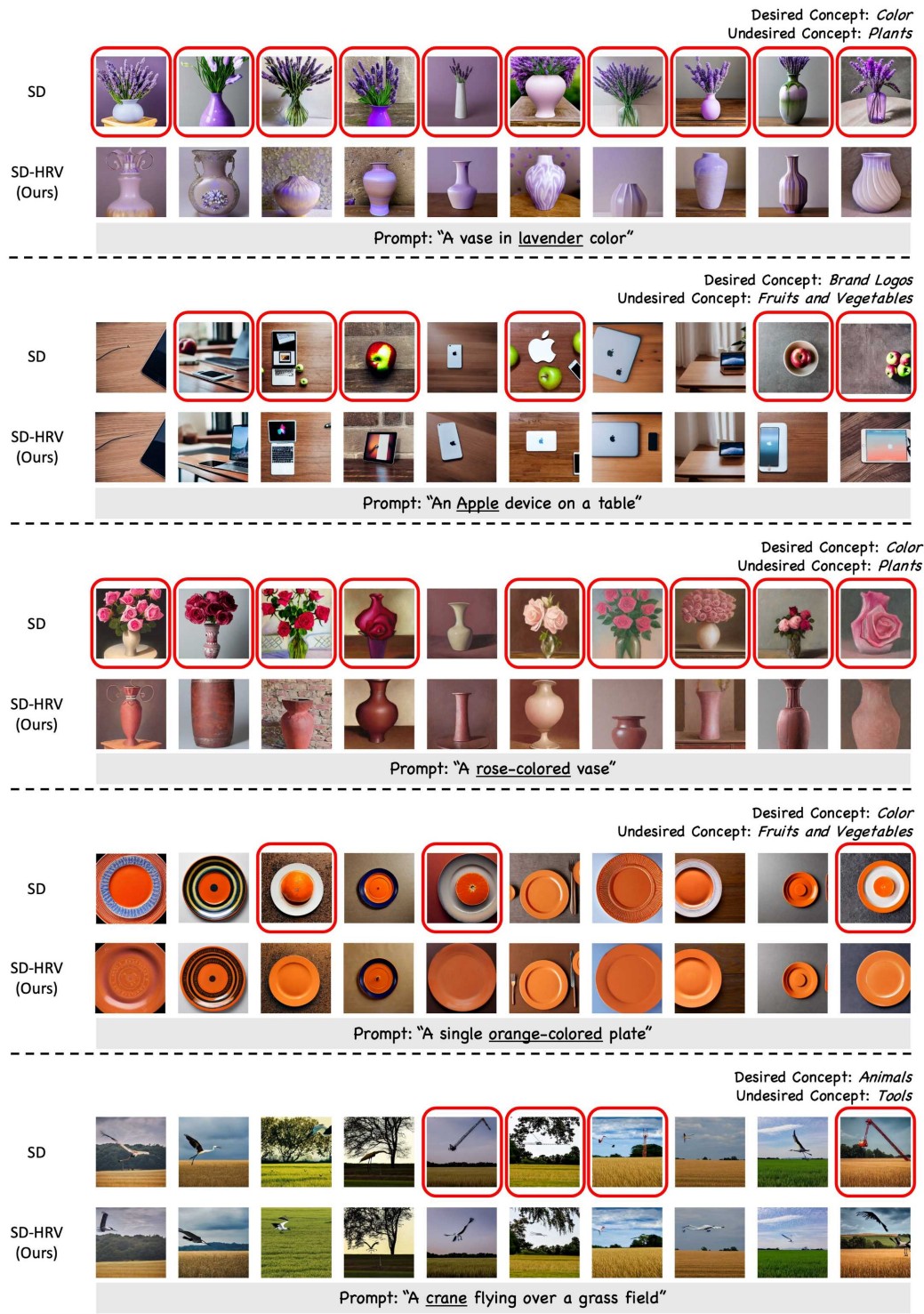

Figure 18: Complete set of generated images used for the human evaluation (Part 1 of 2). Images showing misinterpretations are marked with red boxes.

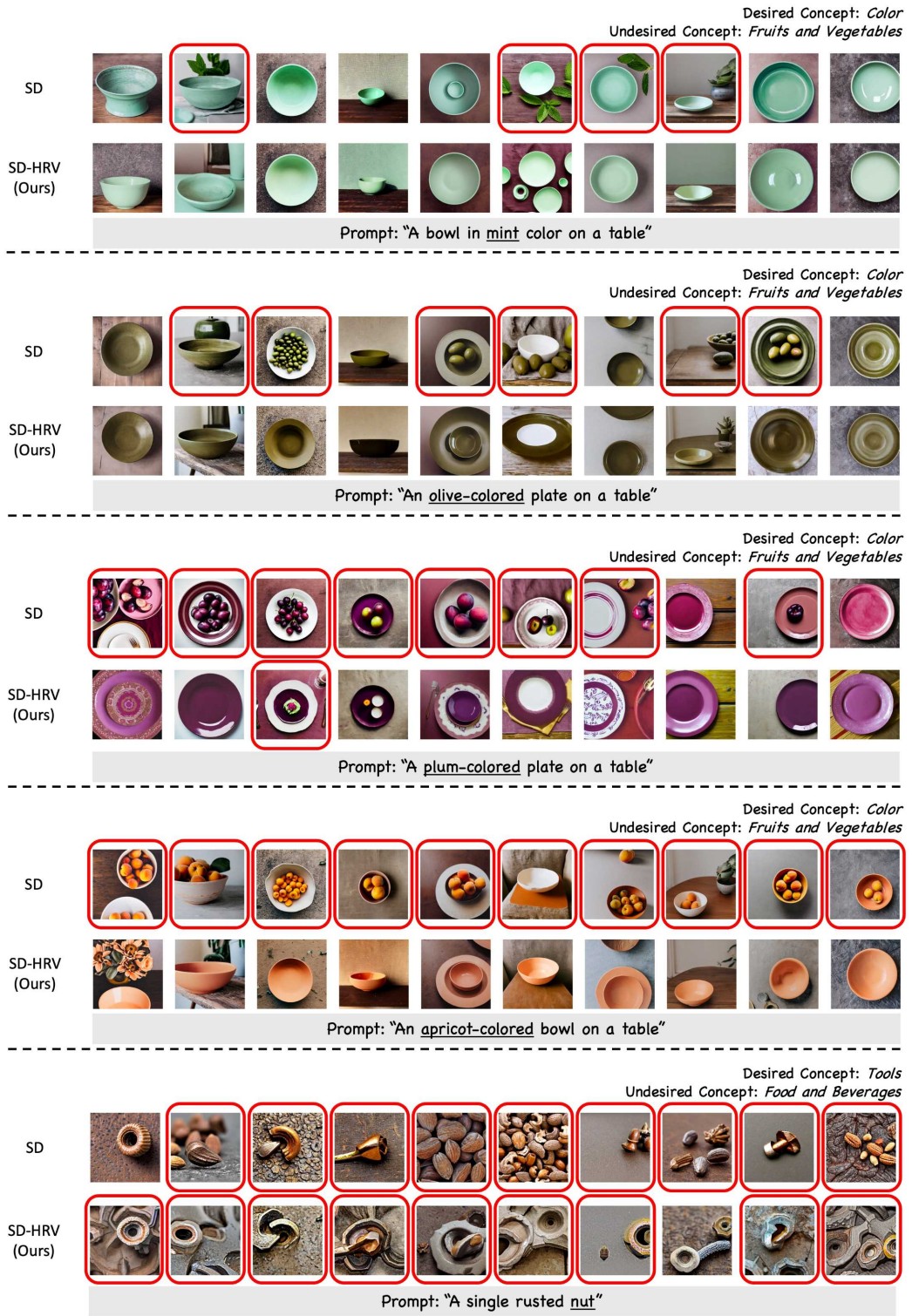

Figure 19: Complete set of generated images used for the human evaluation (Part 2 of 2). Images showing misinterpretations are marked with red boxes.

## D.2 HUMAN EVALUATION

We evaluate the human perceived misinterpretation rate using Amazon Mechanical Turk (AMT), requiring participants to have over 500 HIT approvals, an approval rate above 98%, and live in the US. The survey begins with a sample question accompanied by its correct answer, which is repeated at the end without the answer. Participants who missed the sample question are excluded, leaving 36 valid responses. The misinterpretation rate measures how often polysemous words are misinterpreted in the generated images. We use 10 prompts from Table 7 and 10 random seeds to generate 100 images for each T2I model. This results in 200 total images for comparison between Stable Diffusion (SD) and SD-HRV (Ours). These images are organized into 10 problem sets, each containing 20 images generated with the same prompt using either SD or SD-HRV. Each problem set consists of 4 questions, with each question presenting 5 images generated using the same T2I model but with different random seeds. Each participant receives 3 randomly selected problem sets, containing 12 questions and 60 images. Details of the human evaluation setup are summarized in Table 8. For each question, participants are shown 5 images and asked to count how many depict the intended meaning of the polysemous word without including the unintended meaning: "Count how many of the following five images contain {intended meaning of the polysemous word} but no {unintended meaning of the polysemous word}." This count is then subtracted from 5 to determine the count of images with misinterpretations. After applying *concept adjusting* with our head relevance vectors on Stable Diffusion, the misinterpretation rate drops from 63.0% to 15.9%.

Table 8: Overview of human evaluation details for assessing misinterpretation.

| Number of valid responses | Type of questions | Number of questions | Number of questions per participant | Filtering process |
|---|---|---|---|---|
| 36 | Counting images that satisfy the given condition | 40 | 12 questions with 60 generated images | O |

## D.3 COMPARISON OF CONCEPT STRENGTHENING AND CONCEPT ADJUSTING

We can also apply *concept strengthening*, instead of *concept adjusting*, on Stable Diffusion to reduce misinterpretations. While this approach resolves misinterpretations in some prompts, it is not fully effective in others. Figure 20 shows two cases: the left column shows where concept strengthening fails, and the right column shows where it succeeds. In contrast, concept adjusting succeeds in both cases. This is likely because, in some instances, the undesired concepts are relatively strong, requiring explicit redirection of the T2I model away from those undesired concepts.

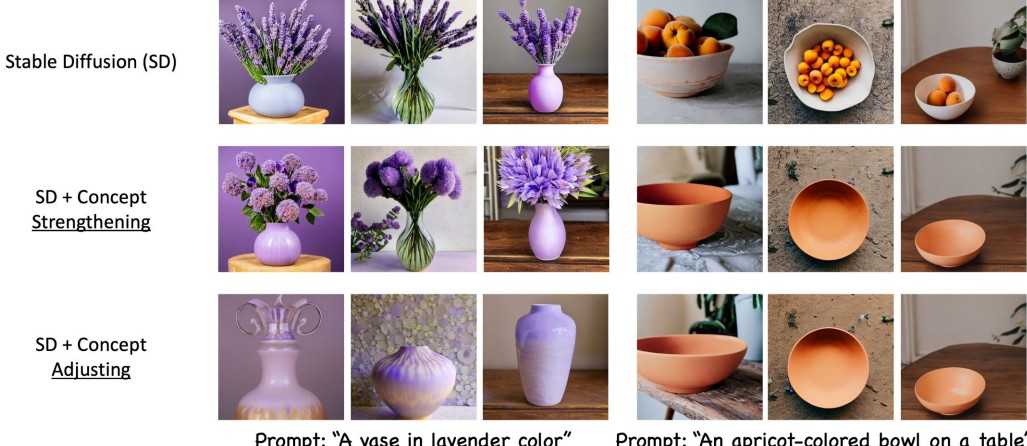

Prompt: "A vase in lavender color"    Prompt: "An apricot-colored bowl on a table"

Figure 20: Comparison of concept strengthening and concept adjusting. Concept strengthening fails in the left case, while concept adjusting succeeds in both cases.

# E DETAILS AND ADDITIONAL RESULTS ON IMAGE EDITING

## E.1 DETAILED EXPLANATIONS ON P2P-HRV

**Brief overview of P2P replacement.** P2P generates target images using the CA maps calculated during the source image generation. Given a source prompt $P$ and a target prompt $P^*$, P2P simultaneously generates images for both prompts, starting from the same Gaussian noise $\mathbf{Z}_1$ and the same random seed $s$. The diffusion denoising process unfolds over timesteps $t = 1, 2, \ldots, T$, where $t = 1$ represents pure noise and $t = T$ the fully denoised image. Let $\tau_c$ denote the CA replacement steps in P2P. During the first $\tau_c$ timesteps ($t \leq \tau_c$), P2P injects structural information from the source image into the target by replacing the CA maps of the target prompt with those from the source prompt. For the remaining timesteps ($t > \tau_c$), target image is generated using its own CA maps without replacement. At each timestep $t$, if $\mathbf{M}_t$ and $\mathbf{M}_t^*$ represent CA maps from the source and target prompts, respectively, and $\widetilde{\mathbf{M}}_t$ denotes the modified CA maps used for generating the target image, then $\widetilde{\mathbf{M}}_t$ is calculated according to the following equation:

$$\widetilde{\mathbf{M}}_t(\mathbf{M}_t, \mathbf{M}_t^*, t) = \begin{cases} \mathbf{M}_t, & \text{if } t \leq \tau_c \\ \mathbf{M}_t^*, & \text{otherwise} \end{cases} \tag{7}$$

In short, P2P uses these modified CA maps $\widetilde{\mathbf{M}}_t$ and the target prompt $P^*$ to generate target images.

**P2P-HRV.** We enhance P2P by applying concept strengthening on the edited token. Consider the source prompt $P$ = 'a blue car' and the target prompt $P^*$ = 'a red car'. During the first $\tau_c$ timesteps, P2P replaces the CA maps of $P^*$ with those of $P$ while generating the target image. However, this can lead to a mismatch with the target prompt, as the CA maps for 'blue' may interfere with properly changing the color from 'blue' to 'red.' To address this, we leave the CA maps for the *edited token* ('red' in this case) unchanged during the first $\tau_c$ timesteps, while replacing the CA maps for the other tokens. This ensures that the CA maps for 'blue' are excluded from the target image generation. The calculation of the modified CA maps $\widetilde{\mathbf{M}}_t$ is therefore adjusted as

$$(\widetilde{\mathbf{M}}_t(\mathbf{M}_t, \mathbf{M}_t^*, t))_{i,j^*}^h = \begin{cases} c_h \cdot (\mathbf{M}_t^*)_{i,j^*}^h, & \text{if } t \leq \tau_c, \ j^* \neq j \\ (\mathbf{M}_t)_{i,j^*}^h, & \text{if } t \leq \tau_c, \ j^* = j \\ (\mathbf{M}_t^*)_{i,j^*}^h, & \text{otherwise}, \end{cases} \tag{8}$$

where $i$ represents a pixel value, $j$ a source text token, $j^*$ a target text token, $h$ a CA head position index, and $c_h = 1$ for all $h = 1, \cdots, 128$. The term $(\widetilde{\mathbf{M}}_t)_{i,j^*}^h$ denotes the $(i, j^*)$-component of the modified $h$-th CA map $(\widetilde{\mathbf{M}}_t)^h$. To further steer the model to focus on the concept being edited, we apply *concept strengthening* by setting $c_h = r_h$, where $r_h$ is the $h$-th component of the rescale vector defined in Figure 4 of Section 5. This is applied across all head positions $h = 1, \cdots, 128$. This final method is referred to as *P2P-HRV*.

## E.2 PROMPTS FOR IMAGE EDITING

We compare P2P-HRV with several state-of-the-art image editing methods across five editing targets, including three object attributes—*Color*, *Material*, and *Geometric Patterns*—and two image attributes—*Image Style* and *Weather Conditions*. The prompt template and concept-words for each visual concept are listed in Table 9. For each prompt, we generate images using 10 random seeds. For example, in the *Color* editing task, 10 random seeds are used with the prompt template 'a {*Color A*} {*Objects*}' (source prompt) → 'a {*Color B*} {*Objects*}' (target prompt), covering 10 color pairs (*Color A*, *Color B*) and 5 objects *(Objects)*, resulting in 500 generated images for each T2I model. The words for *Color A* and *Color B* are sampled from the concept-word set of the visual concept *Color* in the first row of Table 9. The words for *Objects* are sampled similarly. The same process is applied to the other editing tasks, except for *Weather Conditions*, which uses 5 attribute pairs (*Weather Condition A*, *Weather Condition B*), generating 250 images for each T2I model. The full list of prompts and attribute pairs for all five editing tasks can be found in our core codebase.

Table 9: Prompt and word list for image editing

| Visual Concept | Prompt Template |
| --- | --- |
| | **Words** |
| Color | *a {**Color A**} {Objects}* |
| | *→ a {**Color B**} {Objects}* |
| | • {**Color**}: blue, brown, red, purple, pink, yellow, green, white, gray, black |
| | • {Objects}: car, bench, bowl, balloon, ball |
| Material | *a {Objects} made of {**Material A**}* |
| | *→ a {Objects} made of {**Material B**}* |
| | • {**Objects**}: bowl, cup, table, ball, teapot |
| | • {**Material**}: wood, glass, steel, copper, silver, marble, paper, jade, gold, basalt, granite, clay, leather |
| Geometric Patterns | *a {Objects} with a {**Geometric Patterns A**} pattern* |
| | *→ a {Objects} with a {**Geometric Patterns B**} pattern* |
| | • {Objects}: T-shirt, pillow, wallpaper, umbrella, blanket |
| | • {**Geometric Patterns**}: polka-dot, leopard, stripe, greek-key, plaid |
| Image Style | *a {**Image Style A**} style painting of a {Landscapes}* |
| | *→ a {**Image Style B**} style painting of a {Landscapes}* |
| | • {**Image Style**}: cubist, pop art, steampunk, impressionist, black-and-white, watercolor, cartoon, minimalist, sepia, sketch |
| | • {Landscapes}: castle, mountain, cityscape, farmland, forest |
| Weather Conditions | *a {Animals} on a {**Weather Conditions A**} day* |
| | *→ a {Animals} on a {**Weather Conditions B**} day* |
| | • {Animals}: cat, dog, rabbit, frog, bird |
| | • {**Weather Conditions**}: snowy, rainy, foggy, stormy |

### E.3 METRICS FOR IMAGE EDITING AND HUMAN EVALUATION ON TWO IMAGE ATTRIBUTES

For object attributes (*Color*, *Material*, and *Geometric Patterns*), we evaluated performance using CLIP (Radford et al., 2021) and BG-DINO scores. The CLIP score measures the CLIP image-text similarity between the edited image and the target prompt, assessing how well the edited image aligns with the target prompt. Meanwhile, the BG-DINO score assesses structure preservation, focusing only on the non-object parts of the image, as the editing targets are restricted to the objects themselves. To compute the BG-DINO score, we use Grounded-SAM-2 (Ravi et al., 2024; Ren et al., 2024) to extract non-object parts from the source and edited images, process these segmented images with the DINOv2 (Oquab et al., 2023) model to obtain embeddings, and calculate cosine similarity between these two embeddings. While prior works (Parmar et al., 2023; Kim et al., 2024) have employed *segment-and-embed* metrics using LPIPS (Zhang et al., 2018) or CLIP embeddings to focus on specific image regions, we adopt DINOv2 embeddings because the DINOv2 model is trained with a self-supervised objective that enables it to capture unique image characteristics.

For image attributes (*Image Styles* and *Weather Conditions*), we conducted a human evaluation to assess human preference (HP) scores, rather than using the BG-DINO score, as BG-DINO is not suitable for evaluating structure preservation in these cases, which involve edits across the entire image. Using Amazon Mechanical Turk (AMT), we measured HP scores while ensuring quality by requiring participants to have over 500 HIT approvals, an approval rate above 98%, and live in the US. Each survey begins with a sample question that includes the correct answer, which is repeated at the end without the answer provided. After filtering out raters who missed the sample question, we collect 28 valid responses for *Image Style* and 35 for *Weather Conditions*.

We use 50 prompt pairs for *Image Style* and 25 for *Weather Conditions*, as presented in Table 9. For human evaluation, we randomly select a seed previously used to measure CLIP image-text similarities. Images are then generated for each prompt pair using P2P-HRV and four other high-performing

methods, resulting in 250 images for *Image Style* and 125 for *Weather Conditions*. This creates 200 binary choice questions for *Image Style* and 100 for *Weather Conditions*, with each participant answering 20 randomly selected questions. Details of the human evaluation setup are summarized in Table 10. In each question, we present participants with two images—one generated using our approach and the other by a different method—and ask, 'Which edited image better matches the target description, while maintaining essential details of the source image?' If participants cannot decide, they can select the option 'Cannot Determine / Both Equally.' The results are shown in Table 1 of Section 5.2. The HP-score in Table 1 is calculated by dividing the number of selections for the other method by the number of selections for ours and multiplying by 100. For example, an HP score of 35.0 for PnP (Tumanyan et al., 2023) in *Weather Conditions* editing indicates that our method received 2.86 (= 100/35.0) times more votes than PnP in this editing task.

Table 10: Overview of human evaluation details for two image attribute editing.

| Editing Target | Number of valid responses | Type of questions | Number of questions | Number of questions per participant | Filtering process |
|---|---|---|---|---|---|
| Image Style | 28 | Binary choice question | 200 | 20 | O |
| Weather Conditions | 35 | Binary choice question | 100 | 20 | O |

### E.4 TRADE-OFF EFFECT OF SELF-ATTENTION REPLACEMENT IN P2P AND P2P-HRV

While P2P primarily focuses on cross-attention (CA) map replacement, it also shows that adjusting the self-attention (SA) replacement rates can enhance structure preservation. The SA replacement rate determines the initial timesteps during which the SA maps of the edited images are replaced with those of the source images. Higher replacement rates enhance structure preservation by incorporating more structural information from the source images but can reduce image-text alignment due to the increased reliance on source image data. This trade-off effect is illustrated in Figure 21, where both P2P and P2P-HRV are evaluated in *Color* editing benchmark with varying SA replacement rates. In this figure, BG-DINO measures structure preservation, while the CLIP image-text similarity measures image-text alignment. Notably, P2P-HRV consistently achieves significantly higher image-text alignment across all SA replacement rates compared to P2P. This result shows clear Pareto-optimal improvements of P2P-HRV over P2P. For all editing benchmarks in this paper, we use an SA replacement rate of 0.4 for P2P and 0.9 for P2P-HRV, as both provide a balanced trade-off between the two metrics. Examples of images generated with varying SA replacement rates are shown in Figures 22 and 23.

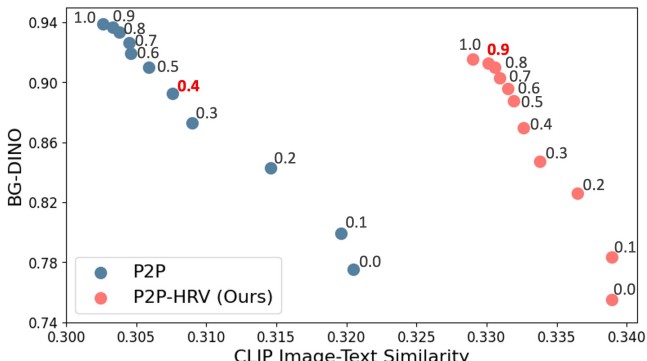

Figure 21: Trade-off effect of self-attention replacement in P2P and P2P-HRV (Ours). Both methods are evaluated on the *Color* editing benchmark with SA replacement rates varying from 0.0 to 1.0. Red-highlighted SA replacement rates indicate points where P2P and P2P-HRV achieve a balanced trade-off between the CLIP and BG-DINO scores. These values are used for P2P and P2P-HRV in all experiments presented in this paper.

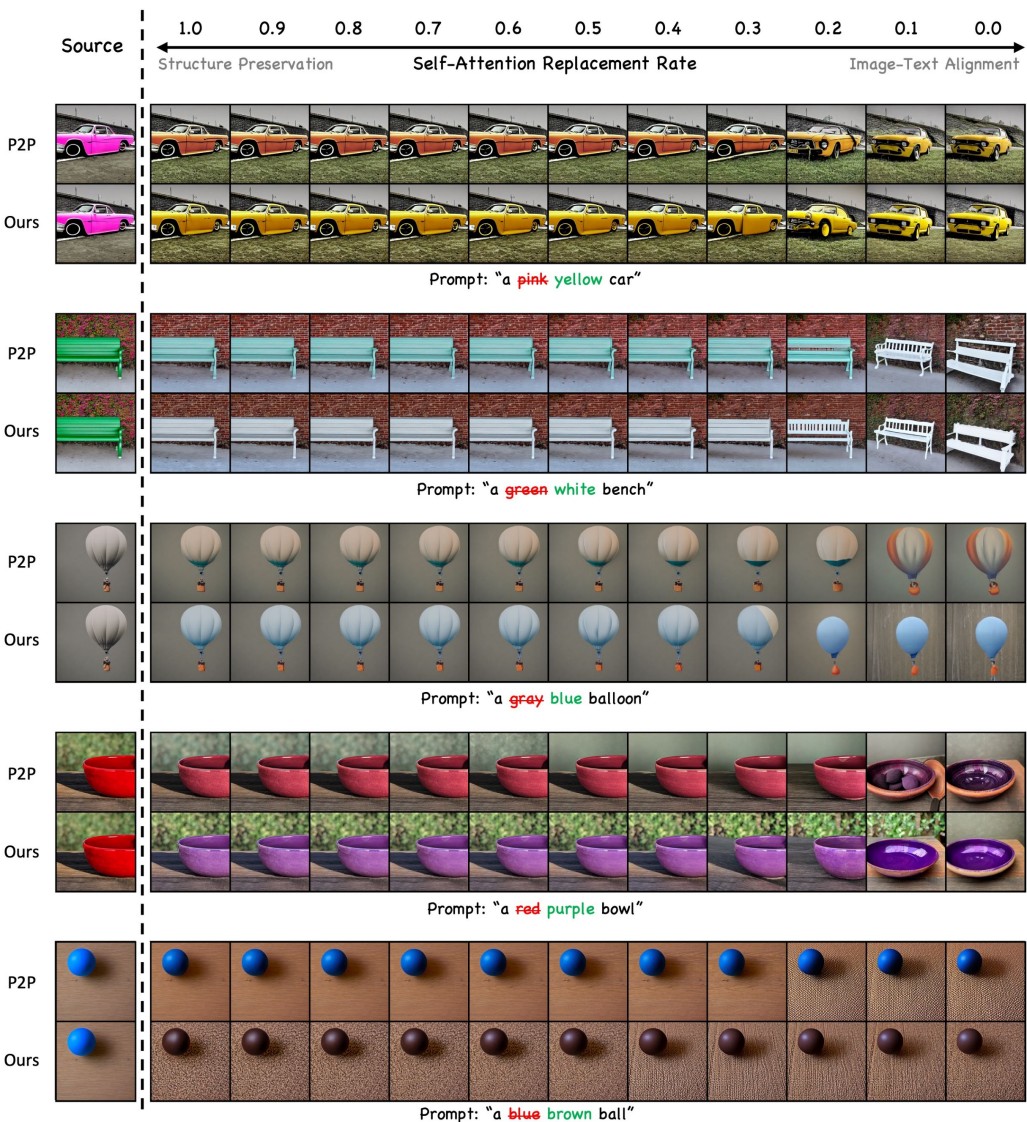

Figure 22: Qualitative results of image editing comparing P2P (Hertz et al., 2022) and ours, based on the variation of self-attention replacement rate (Part 1 of 2).

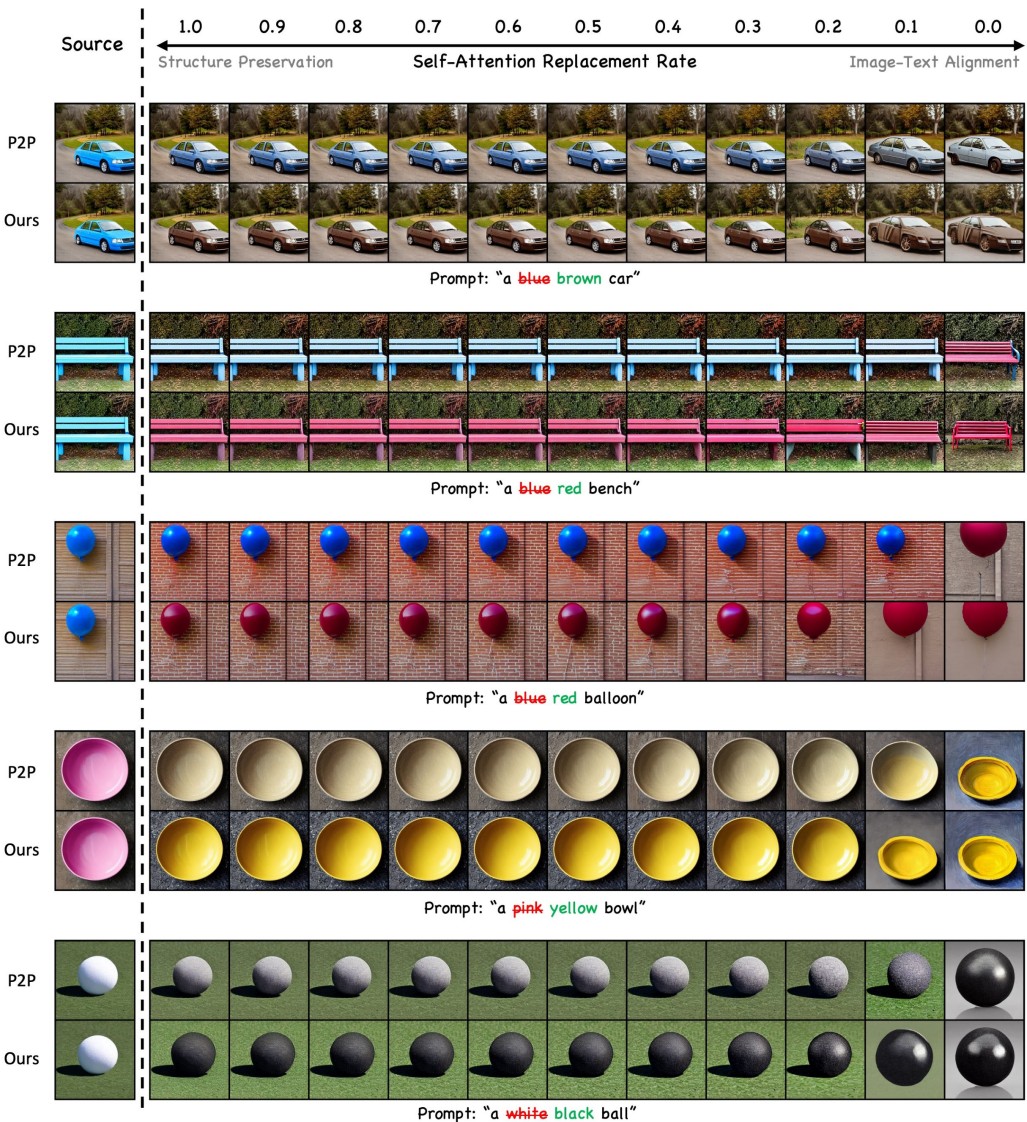

Figure 23: Qualitative results of image editing comparing P2P (Hertz et al., 2022) and ours, based on the variation of self-attention replacement rate (Part 2 of 2).

### E.5 Additional results on image editing

Figures 24–34 present additional qualitative results of image editing for three object attributes and two image attributes.

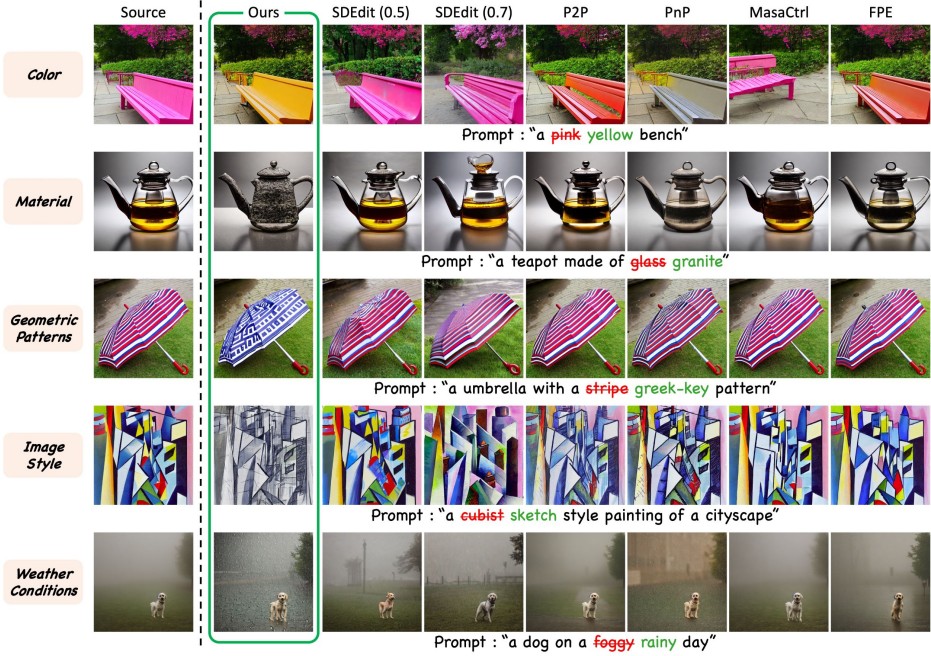

Figure 24: Qualitative results of image editing for three object attributes and two image attributes (Part 1 of 2).

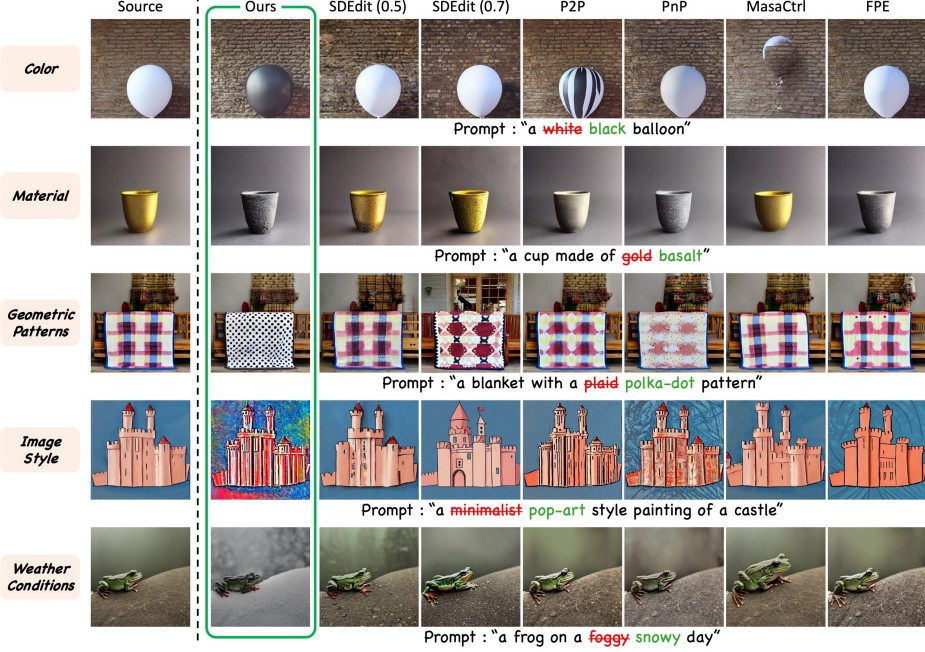

Figure 25: Qualitative results of image editing for three object attributes and two image attributes (Part 2 of 2).

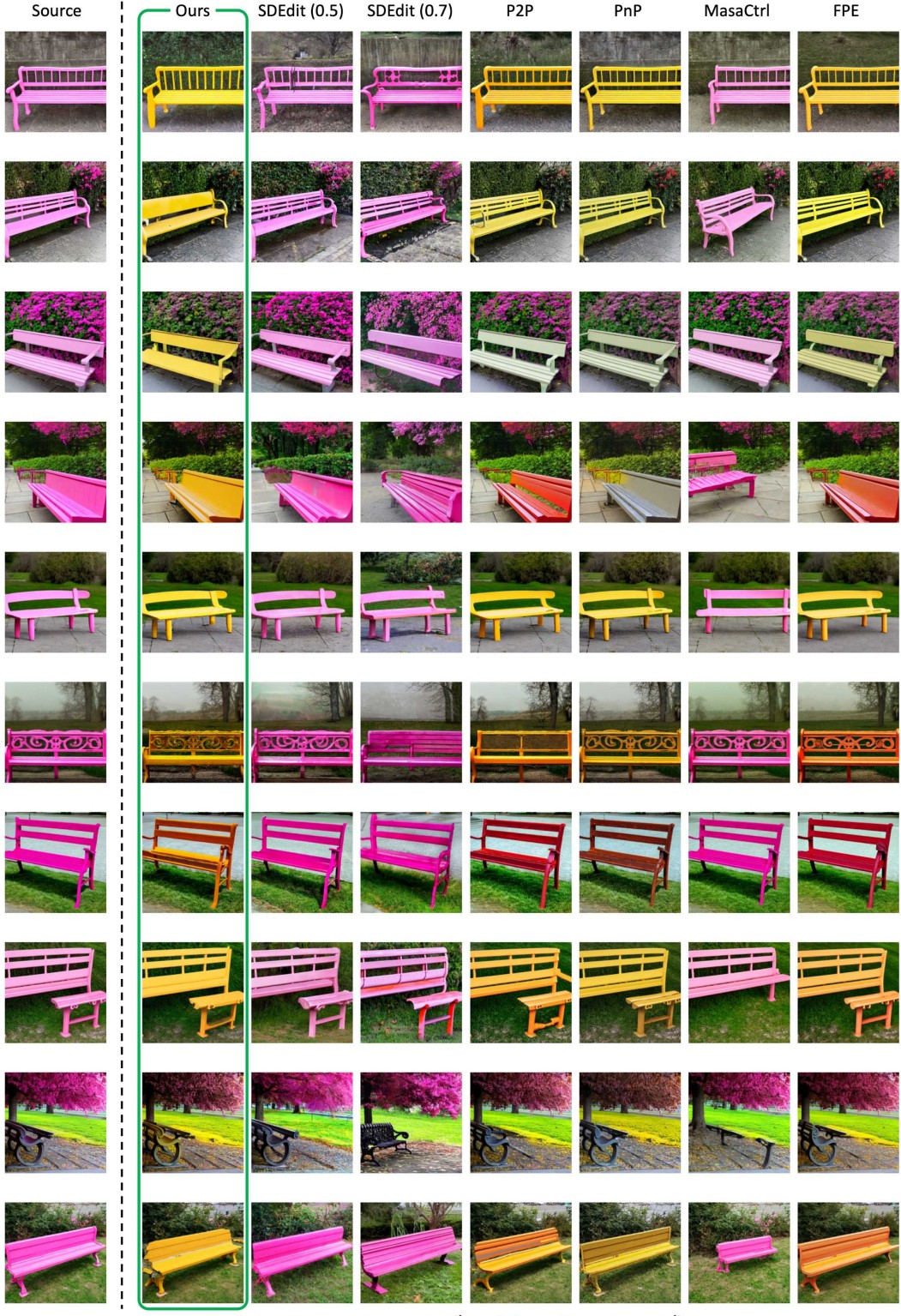

Figure 26: Qualitative results of image editing for the visual concept *Color* (Part 1 of 2). The results were generated using 10 random seeds and were used in the quantitative evaluation.

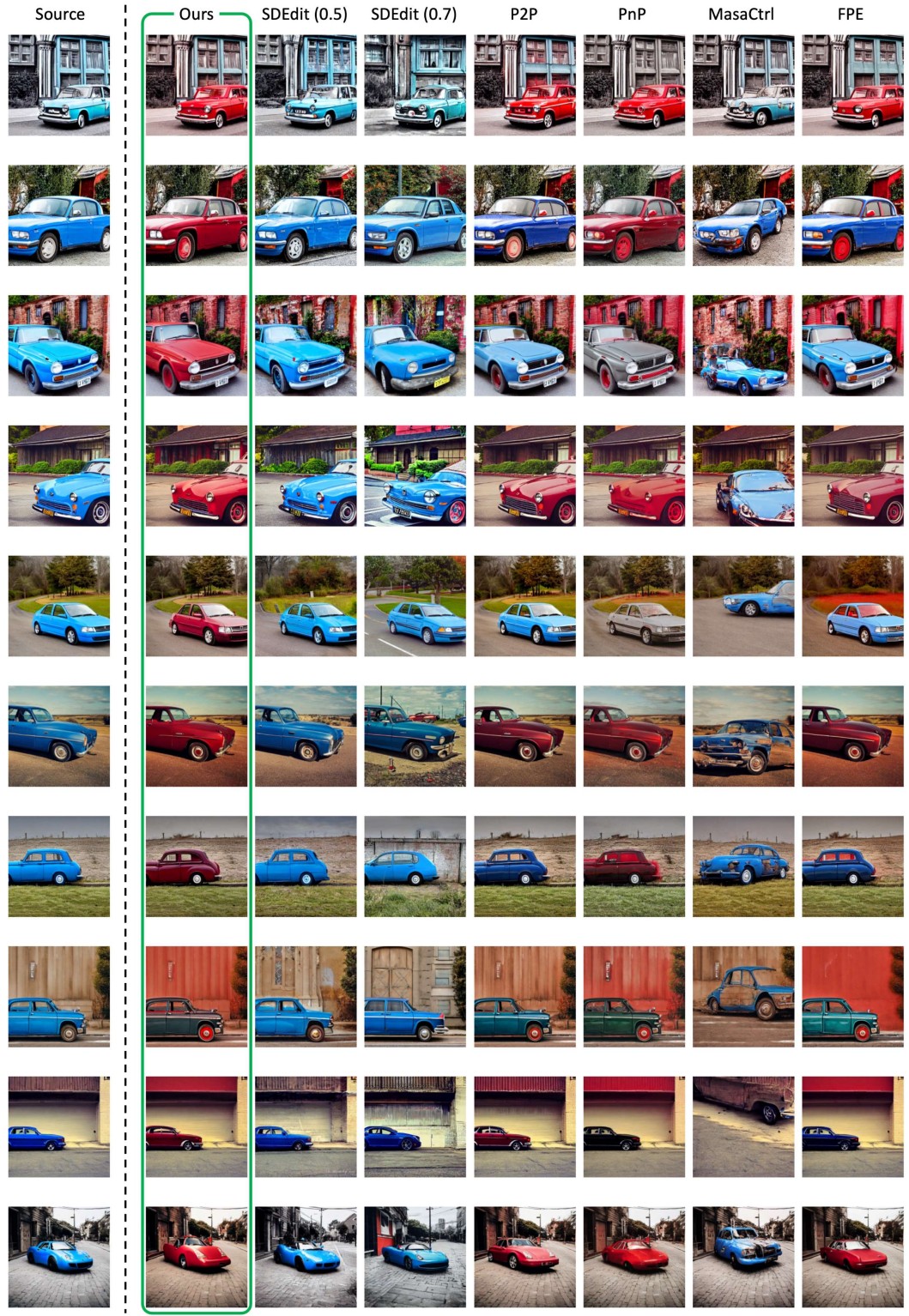

Figure 27: Qualitative results of image editing for the visual concept *Color* (Part 2 of 2). The results were generated using 10 random seeds and were used in the quantitative evaluation.

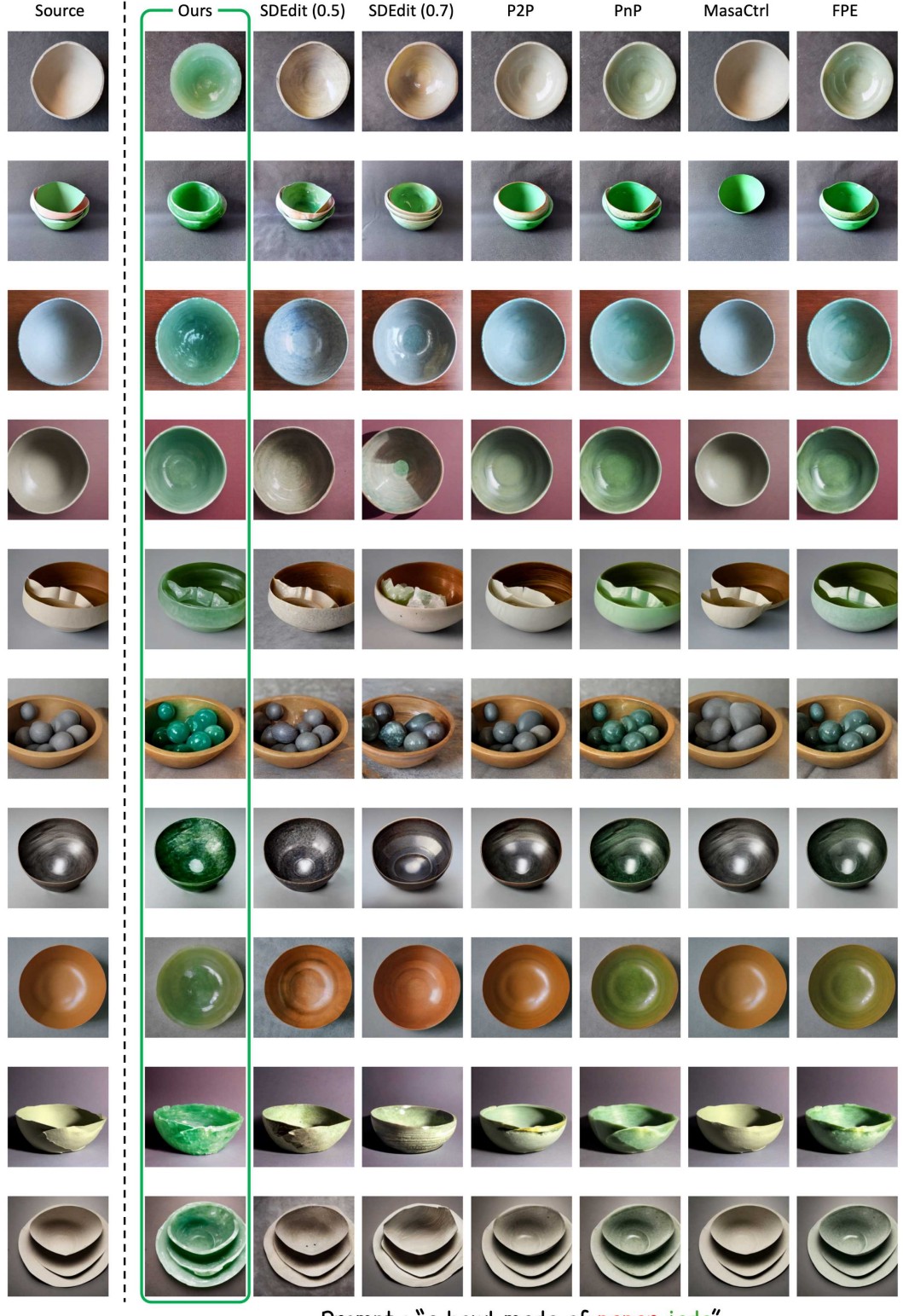

Figure 28: Qualitative results of image editing for the visual concept *Material* (Part 1 of 2). The results were generated using 10 random seeds and were used in the quantitative evaluation.

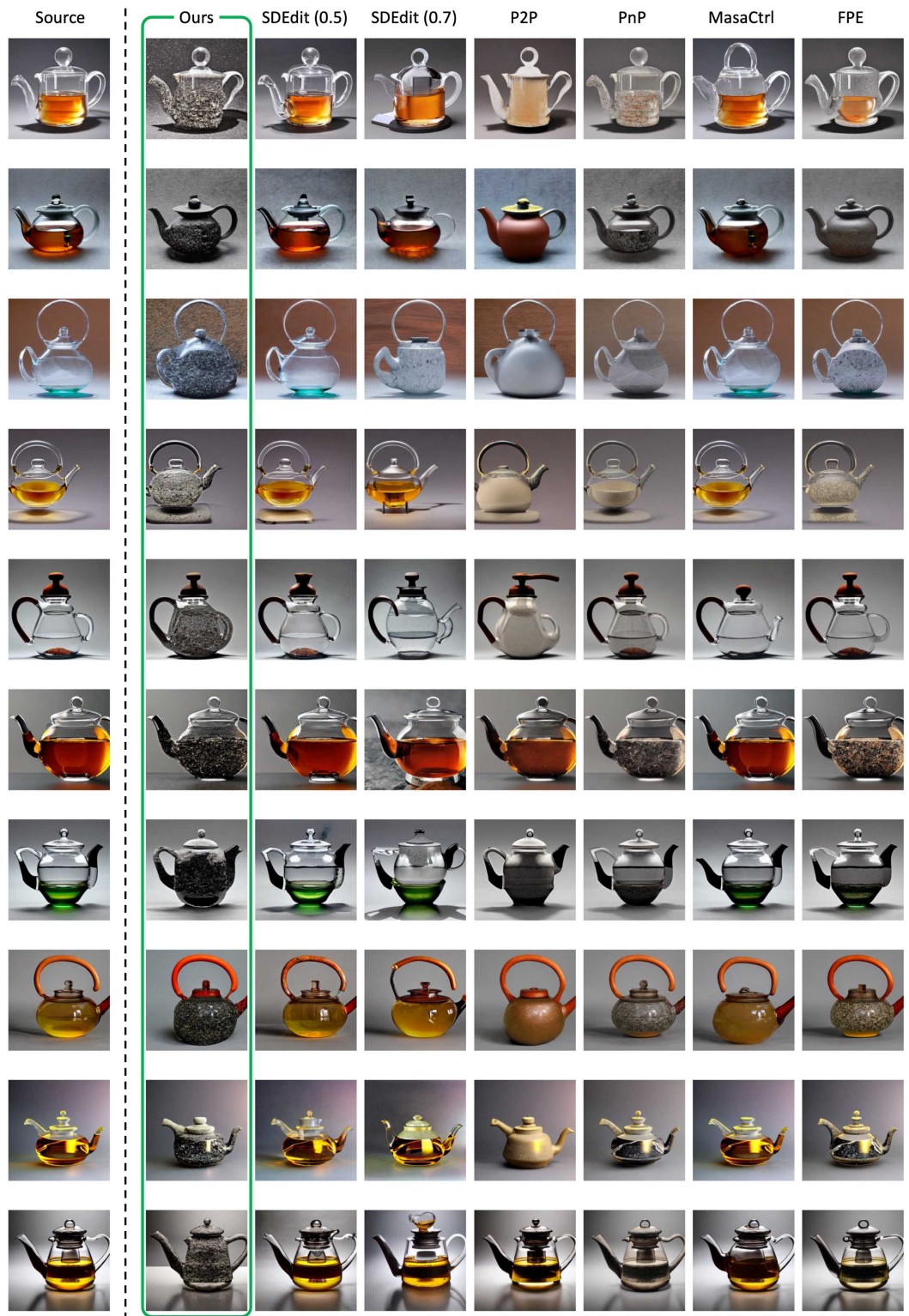

Figure 29: Qualitative results of image editing for the visual concept *Material* (Part 2 of 2). The results were generated using 10 random seeds and were used in the quantitative evaluation.

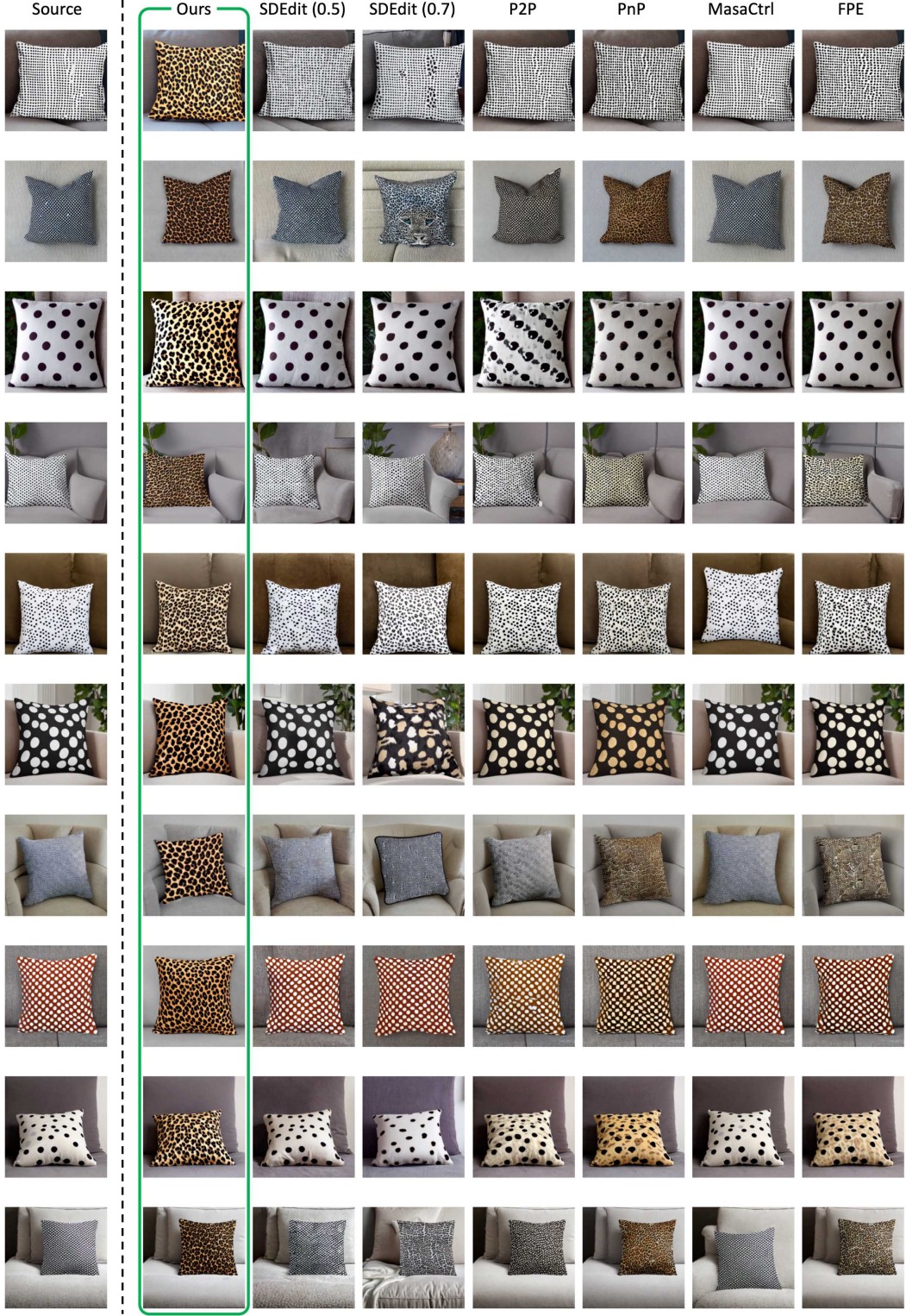

Prompt : "a pillow with a ~~polka-dot~~ leopard pattern"

Figure 30: Qualitative results of image editing for the visual concept *Geometric Patterns* (Part 1 of 2). The results were generated using 10 random seeds and were used in the quantitative evaluation.

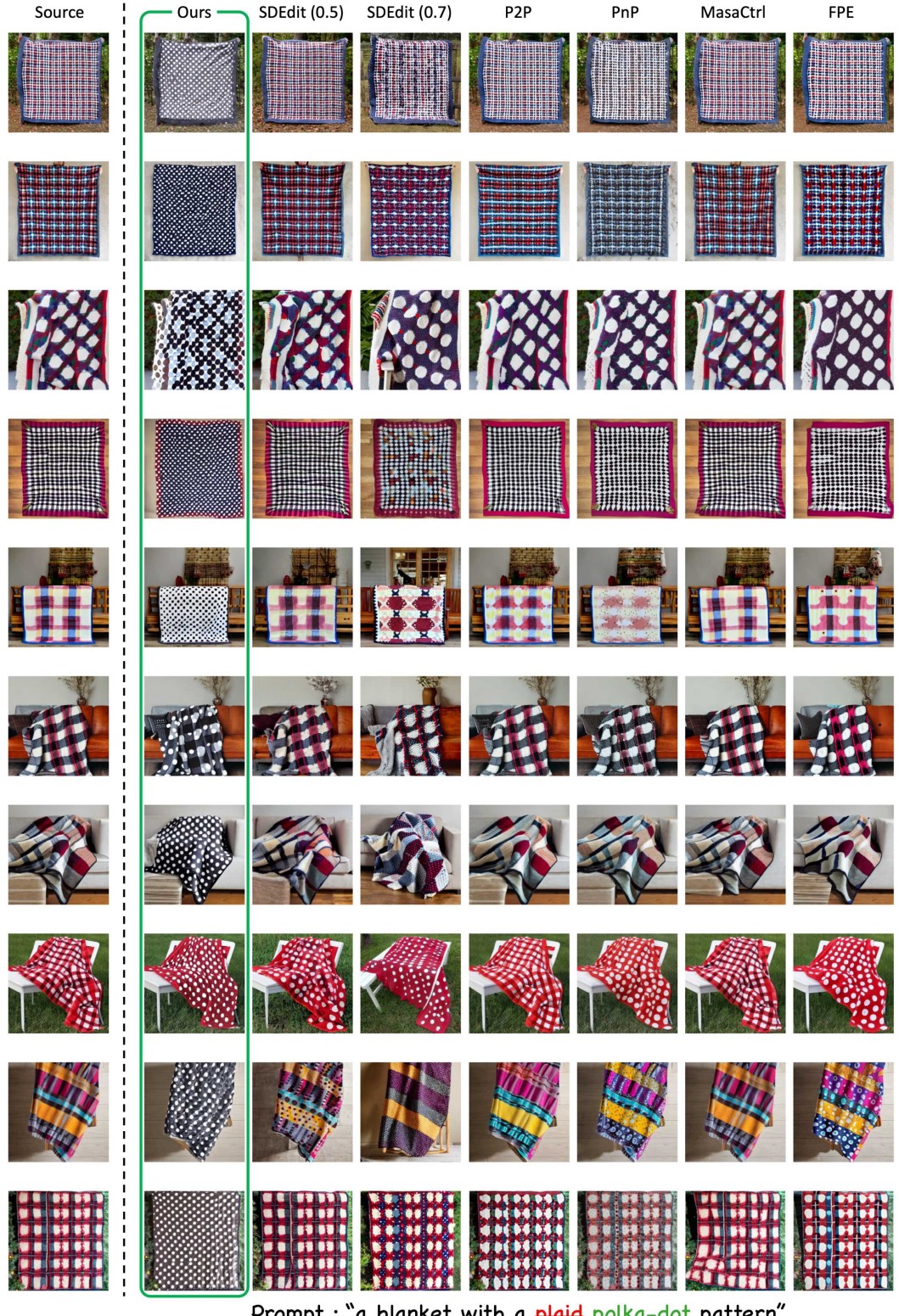

Figure 31: Qualitative results of image editing for the visual concept *Geometric Patterns* (Part 2 of 2). The results were generated using 10 random seeds and were used in the quantitative evaluation.

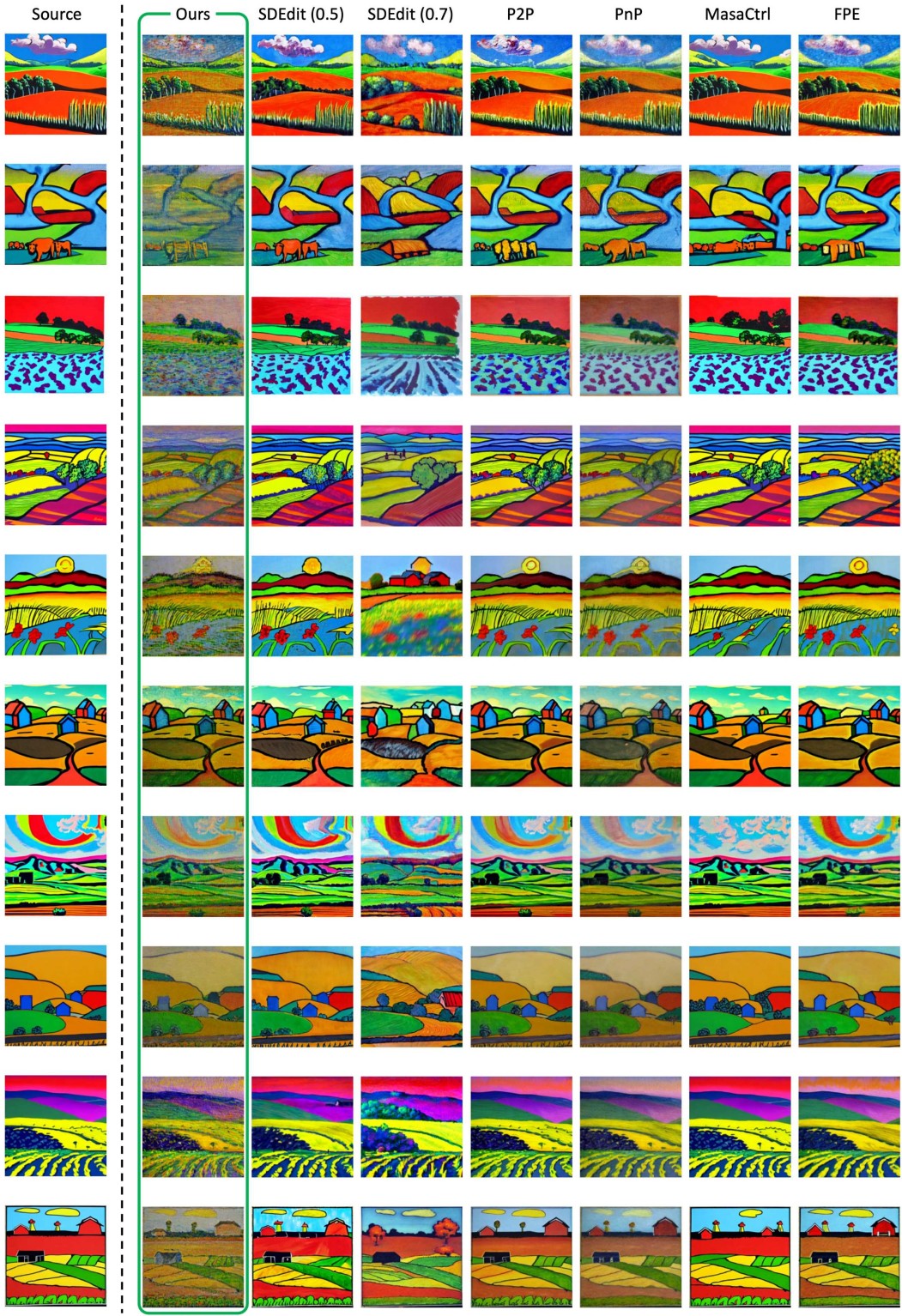

Figure 32: Qualitative results of image editing for the visual concept *Image Style* (Part 1 of 2). The results were generated using 10 random seeds and were used in the quantitative evaluation.

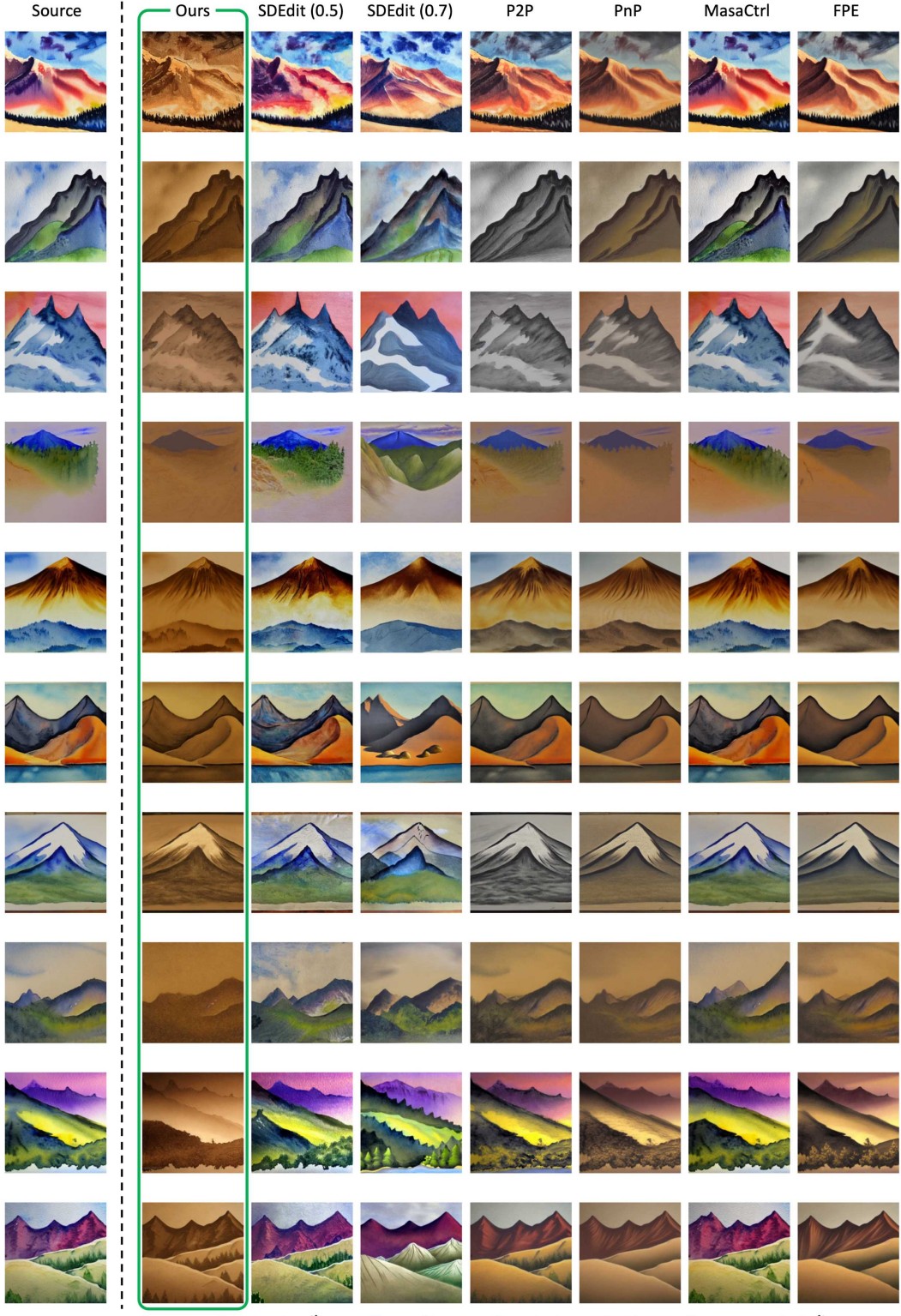

Figure 33: Qualitative results of image editing for the visual concept *Image Style* (Part 2 of 2). The results were generated using 10 random seeds and were used in the quantitative evaluation.

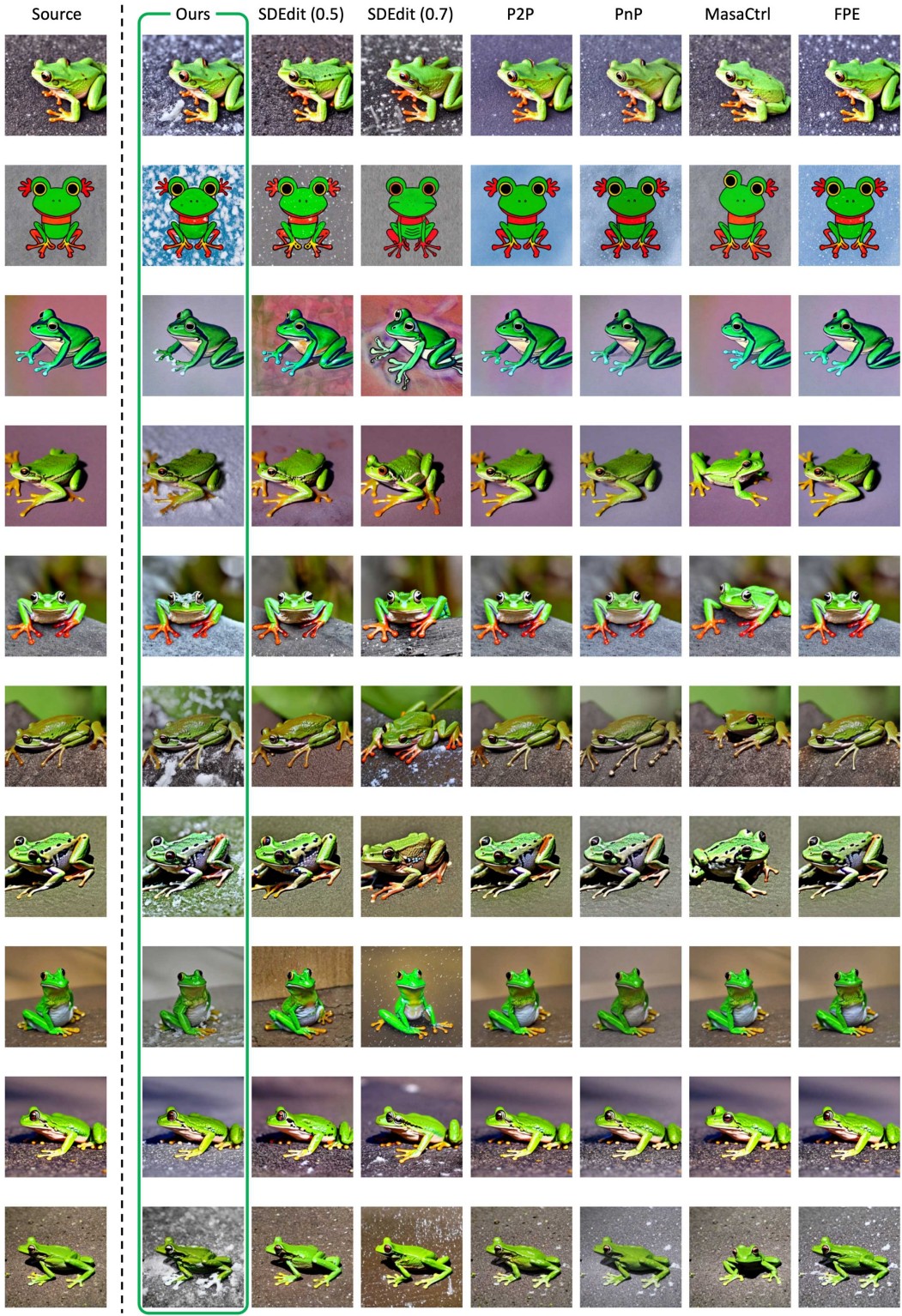

Figure 34: Qualitative results of image editing for the visual concept *Weather Conditions*. The results were generated using 10 random seeds and were used in the quantitative evaluation.

# F DETAILS AND ADDITIONAL EXPERIMENTS ON MULTI-CONCEPT GENERATION

Attend-and-Excite (A&E) excites signals for subjects using averaged cross-attention maps, which are averaged across different CA layers. A&E-HRV extends this by using HRVs to re-weight each cross-attention map before averaging. Both A&E and A&E-HRV are applied only to noun tokens, as in the original Attend-and-Excite. In our multi-concept generation experiments, we used two types of prompts: (i) 'a {*Animal A*} and a {*Animal B*}' (Type 1) and (ii) 'a {*Color A*} {*Animal A*} and a {*Color B*} {*Animal B*}' (Type 2). Table 11 lists the 12 animals and 10 colors used to generate these prompts, with the full prompt list available in our core codebase.

Table 11: Word list for multi-concept generation

| Visual Concept | Words |
|---|---|
| Animals | dog, cat, squirrel, fox, lion, frog, deer, penguin, bird, horse, bear, fish |
| Color | blue, brown, red, purple, pink, yellow, green, white, gray, black |

## F.1 COMPARISON WITH ATTRIBUTE-BINDING METHOD

In this section, we compare A&E-HRV with the recently proposed attribute-binding method, SynGen (Rassin et al., 2024). Since SynGen requires attribute words to be included in the prompt, we focus our comparison on Type 2 prompts. The quantitative results, shown in Table 12, show that our approach consistently outperforms SynGen across all three metrics–full prompt similarity, minimum object similarity, and BLIP-score–by margins of 2.8% to 5.8%. The qualitative results in Figure 35 show that SynGen often fails to generate both objects, while our approach generates object concepts more reliably.

Table 12: Type 2 results: Multi-concept generation using SynGen and our method. The percentage in parentheses indicates the improvement over the result of SynGen.

| Method | Type2: a {*Color A*} {*Animal A*} and a {*Color B*} {*Animal B*} | | |
|---|---|---|---|
| | Full Prompt | Min. Object | BLIP-score |
| SynGen | 0.3820 | 0.1960 | 0.6398 |
| A&E-HRV (Ours) | **0.3971 (+4.0%)** | **0.2073 (+5.8%)** | **0.6580 (+2.8%)** |

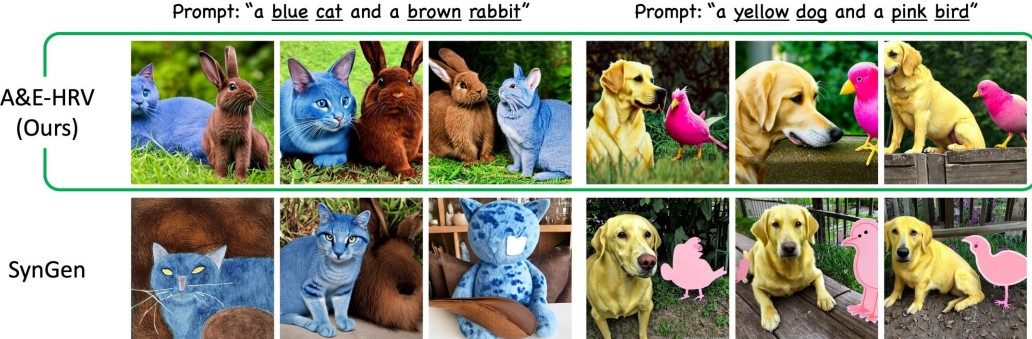

Figure 35: Qualitative comparison of the results for Type 2 prompts between SynGen (Rassin et al., 2024) and ours.

## F.2 Additional results on multi-concept generation

Figure 36 presents additional qualitative results of multi-concept generation for both Type 1 and Type 2 prompts.

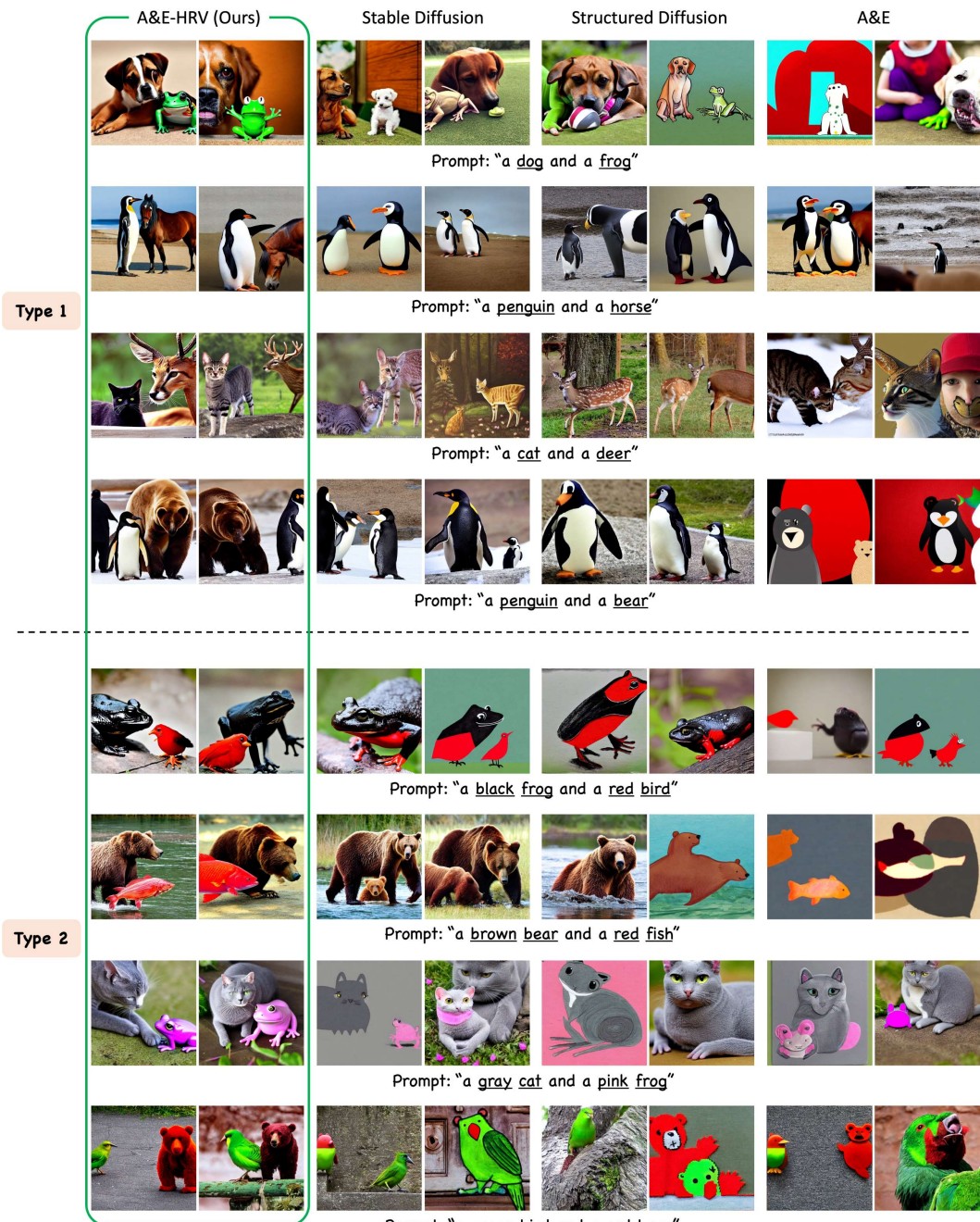

Figure 36: Qualitative comparison of the results for Type 1 and Type 2 prompts. We compare Stable Diffusion (Rombach et al., 2022), Structured Diffusion (Feng et al., 2022), and Attend-and-Excite (Chefer et al., 2023) with ours.

# G  ADDITIONAL RESULTS USING SDXL

## G.1  ADDITIONAL RESULTS ON ORDERED WEAKENING ANALYSIS

Figures 37–42 present additional results from the ordered weakening analysis on Stable Diffusion XL (SDXL) (Podell et al., 2024).

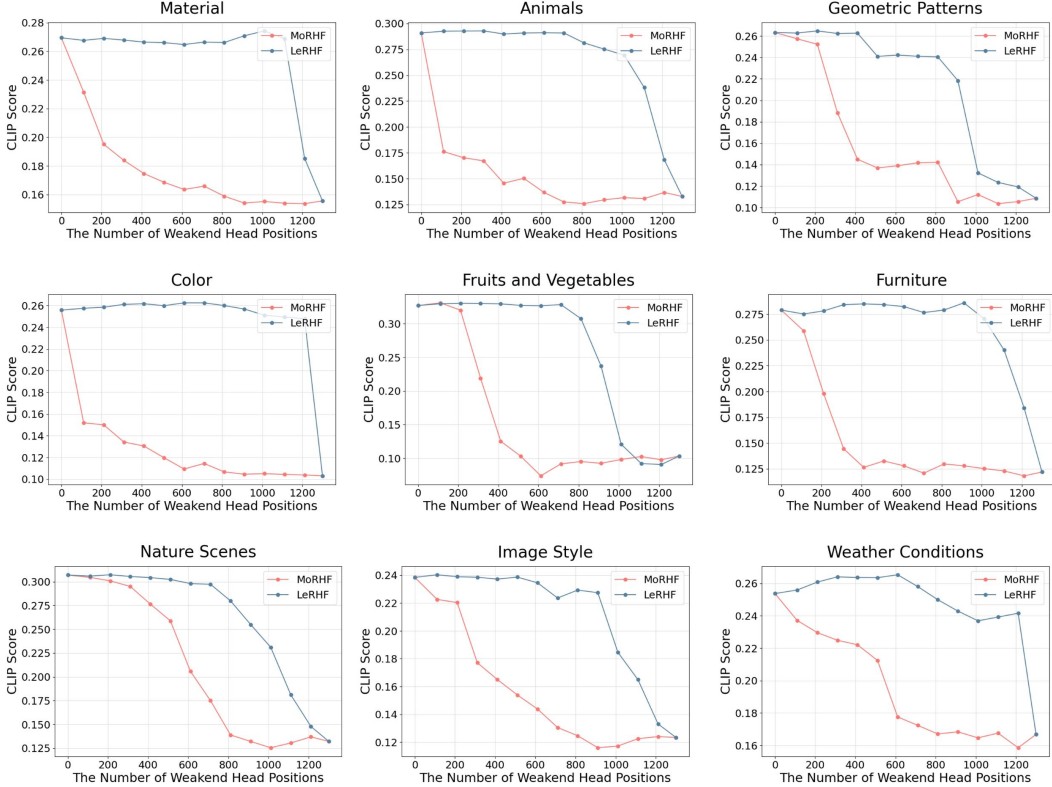

Figure 37: Ordered weakening analysis using SDXL: Change in CLIP image-text similarity score as weakening progresses in either MoRHF or LeRHF order.

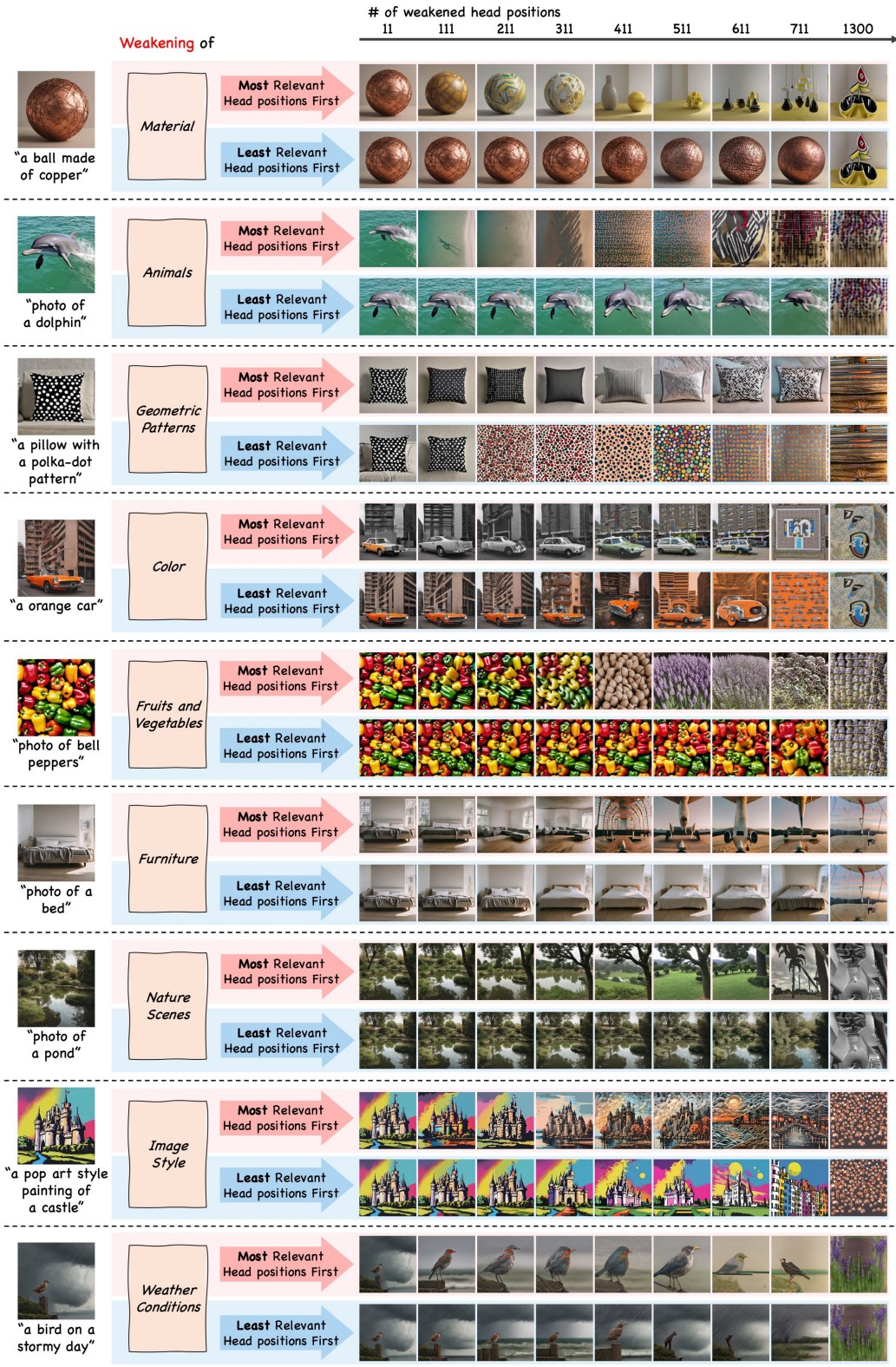

Figure 38: Ordered weakening analysis using SDXL: Generated images as weakening progresses in either MoRHF or LeRHF order (Part 1 of 5).

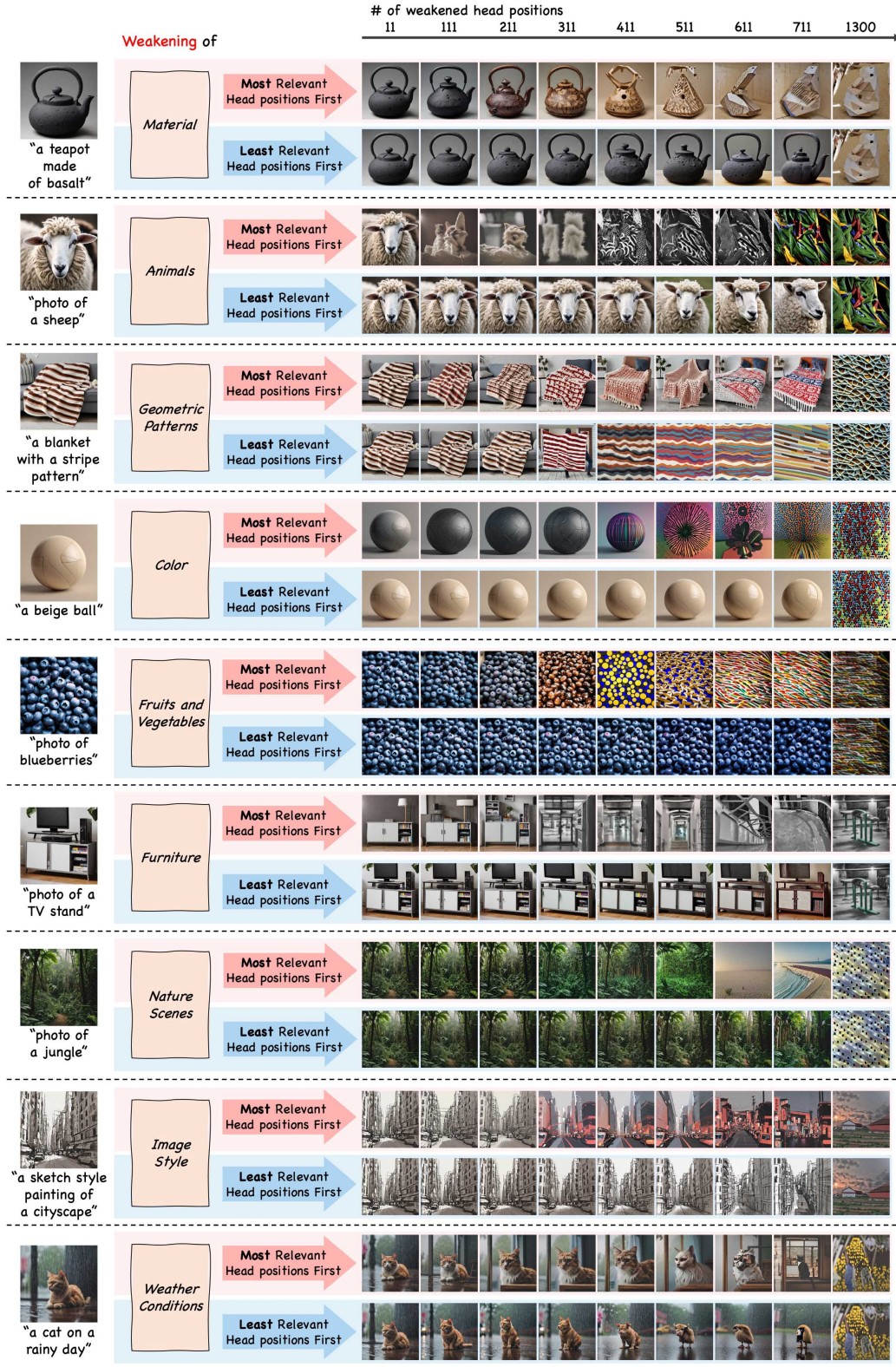

Figure 39: Ordered weakening analysis using SDXL: Generated images as weakening progresses in either MoRHF or LeRHF order (Part 2 of 5).

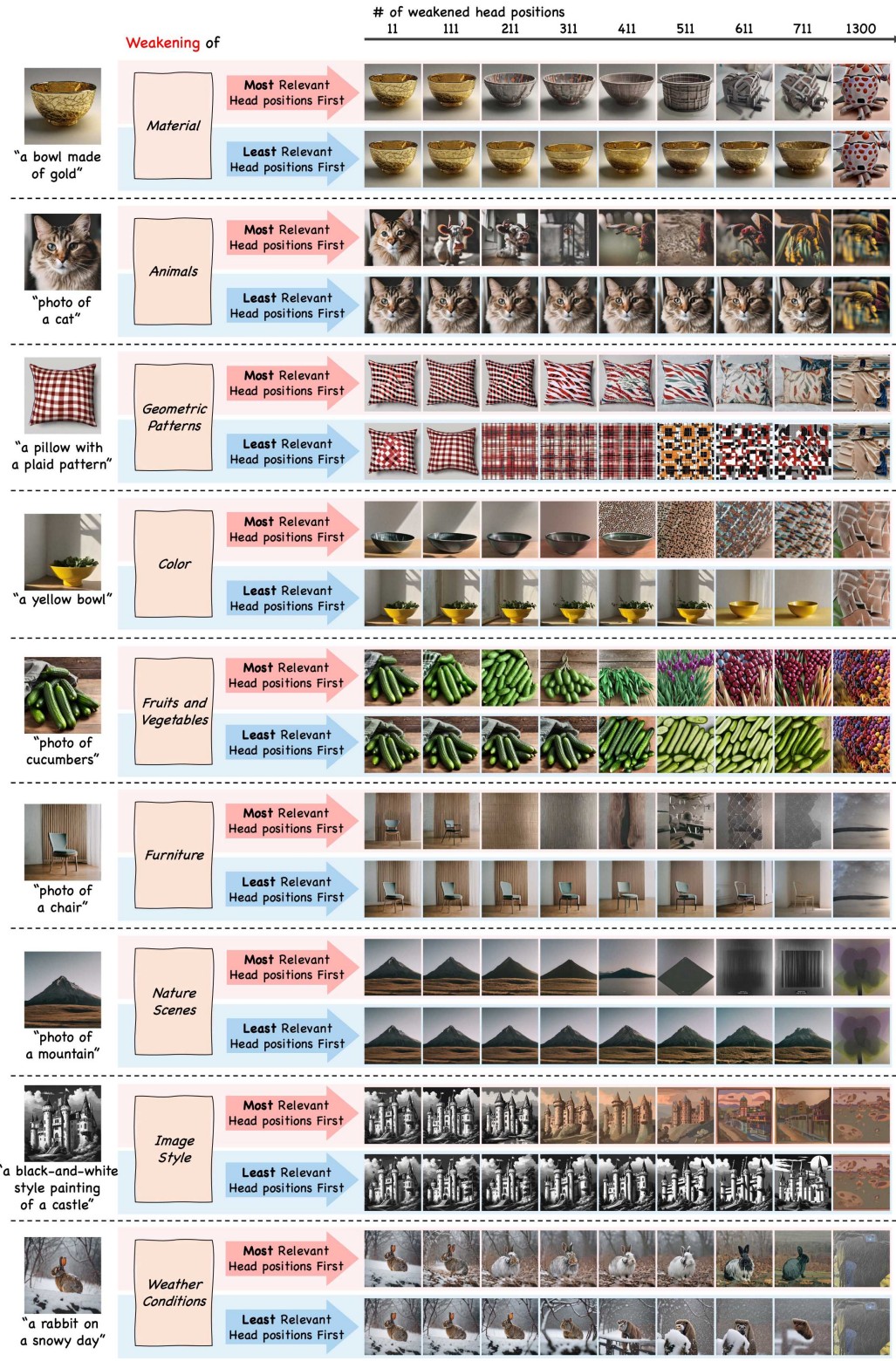

Figure 40: Ordered weakening analysis using SDXL: Generated images as weakening progresses in either MoRHF or LeRHF order (Part 3 of 5).

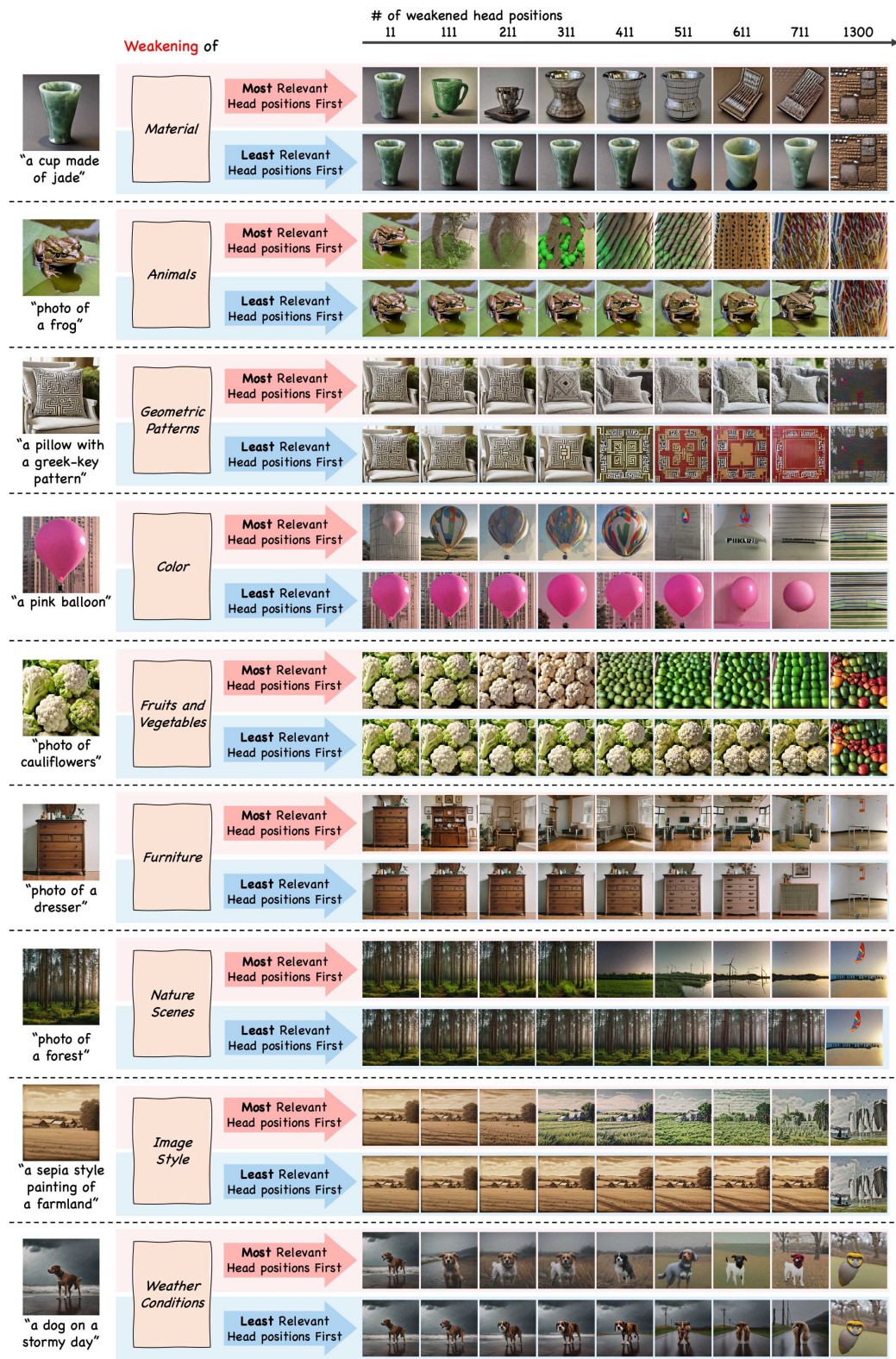

Figure 41: Ordered weakening analysis using SDXL: Generated images as weakening progresses in either MoRHF or LeRHF order (Part 4 of 5).

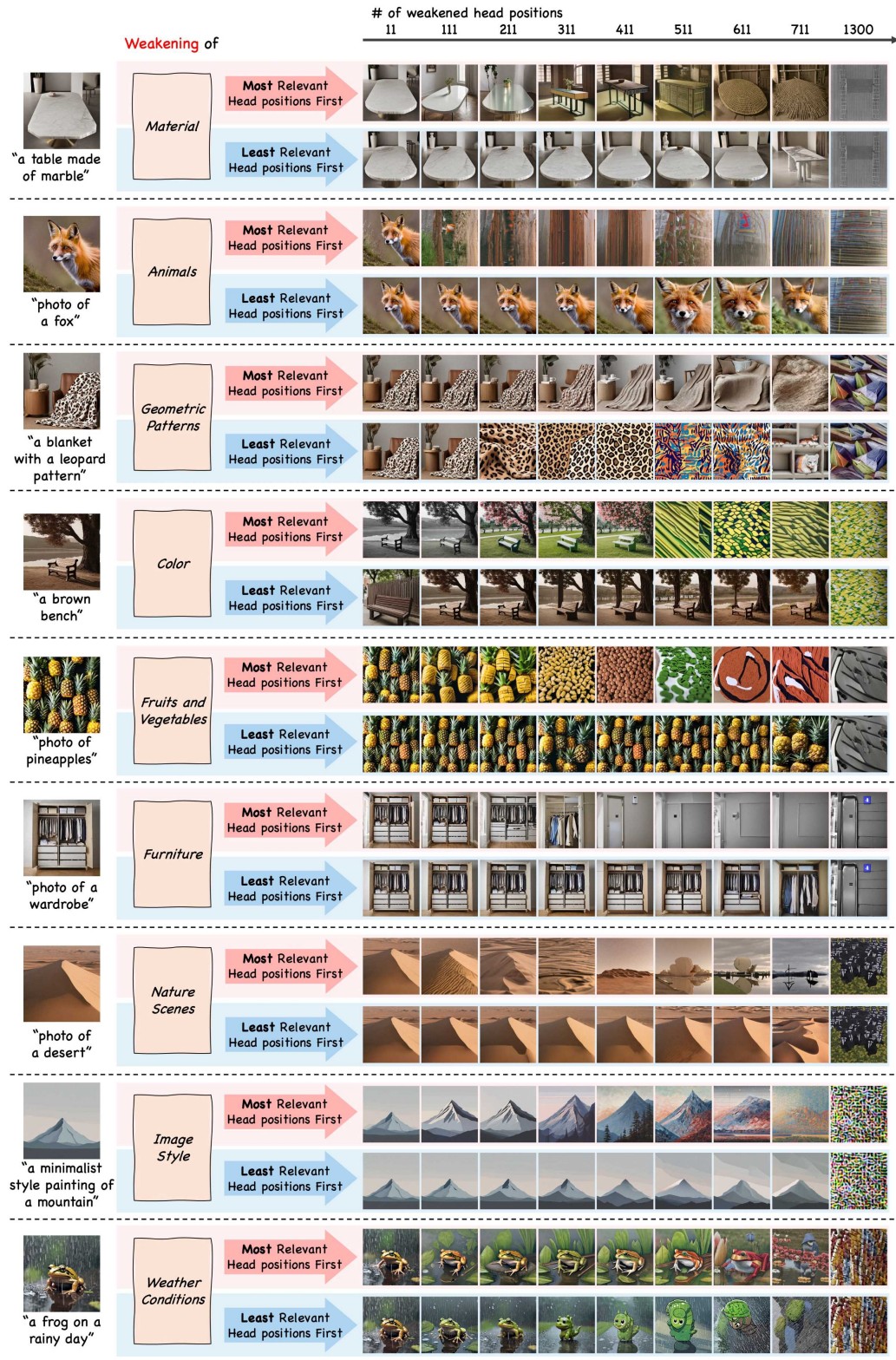

Figure 42: Ordered weakening analysis using SDXL: Generated images as weakening progresses in either MoRHF or LeRHF order (Part 5 of 5).

### G.2 ORDERED WEAKENING ANALYSIS WITH MORE COMPLEX IMAGES

In this section, we present two examples of ordered weakening analysis applied to more complex images using the prompts 'a plastic car melting' and 'a metal chair rusting.' Before starting the analysis, we introduce a new concept, *Physical and Chemical Processes*, into our set of 34 concepts, adding five corresponding concept-words: melting, rusting, boiling, freezing, and burning. We then re-compute HRV vectors. In the top example of Figure 43, we perform ordered weakening analysis with the generation prompt 'a plastic car melting' by weakening either the *Physical and Chemical Processes* or the *Vehicles* concept. When weakening *Physical and Chemical Processes* in MoRHF order, the concept of 'melting' is eliminated first, while the 'car' persists for a longer period. In contrast, when weakening *Vehicles*, the concept of 'car' is eliminated first, and 'melting' is preserved longer. Notably, the entangled property 'plastic' is initially affected when weakening 'melting,' but it is removed more slowly from the image. This is even more apparent in the bottom example of Figure 43, where weakening *Physical and Chemical Processes* eliminates 'rusting' first, while the concept of 'metal' is retained longer. These examples demonstrate that our HRV and ordered weakening analysis work well with more complex images.

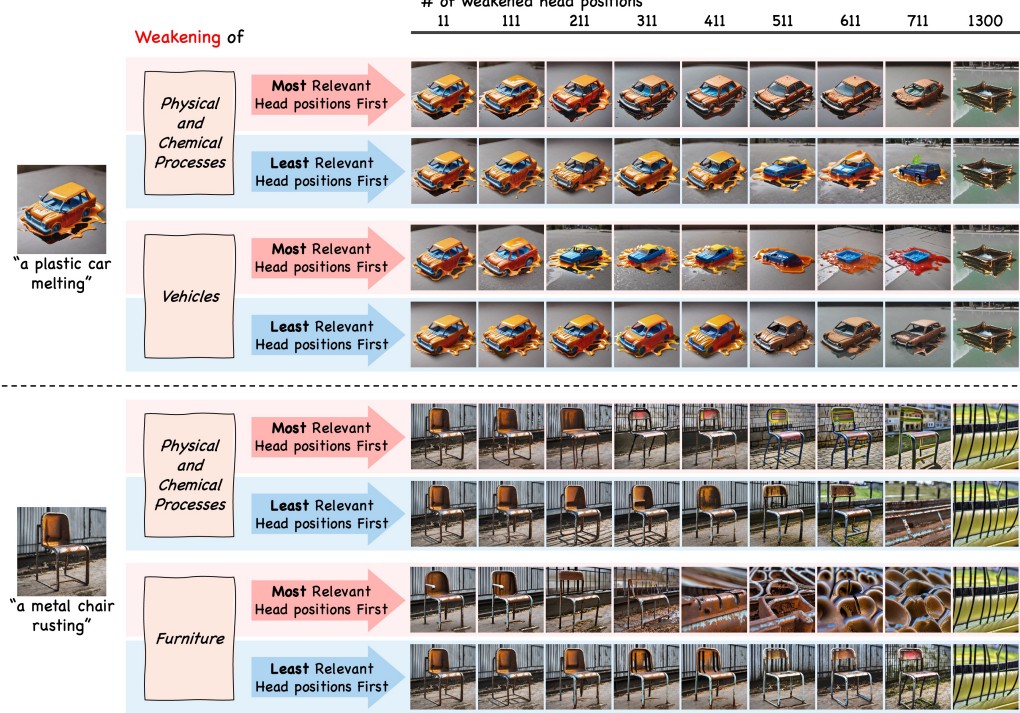

Figure 43: Ordered weakening analysis with more complex images: Generated images as weakening progresses in either MoRHF or LeRHF order using SDXL.

## G.3  REDUCING MISINTERPRETATION IN SDXL

SDXL significantly improves image generation performance compared to SD v1 models, thanks to its three times larger U-Net backbone and two CLIP text encoders. It also reduces misinterpretation issues with prompts from Table 7, but the problem is not entirely resolved, as undesired concepts still appear in generated images. For instance, among 100 images generated using these prompts, nearly all included the desired concepts, but about 35 also contained undesired concepts. Figure 44 illustrates this with two prompts: 'An Apple device on a table' and 'A rose-colored vase.' With SDXL, desired concepts (Apple device or rose-colored vase) are consistently generated; however, one image for the first prompt included the undesired concept of a fruit apple, and nine images for the second prompt included the undesired concept of a flower rose. In both cases, SDXL-HRV (SDXL with *concept adjusting*) effectively reduces misinterpretation by preventing the generation of undesired concepts.

In Figure 44, SDXL-HRV images for 'A rose-colored vase' tend to exhibit rose coloring across most parts of the images. We suspect this issue is related to the normalization of HRV vectors, which are currently normalized to have an L1 norm equal to their length, $H$. This approach is based on the fact that a vector with all elements set to one also has an L1 norm of $H$, and using this vector as a rescaling factor does not alter the image generation process. For SD v1.4, $H = 128$, but for SDXL, $H = 1300$, which may cause some HRV vector elements to become too large when rescaling CA maps. One possible solution is to clamp each HRV element to an upper bound $b$, replacing any value greater than $b$ with $b$. Future work will explore this and other normalization strategies to identify approaches better suited to high $H$ value.

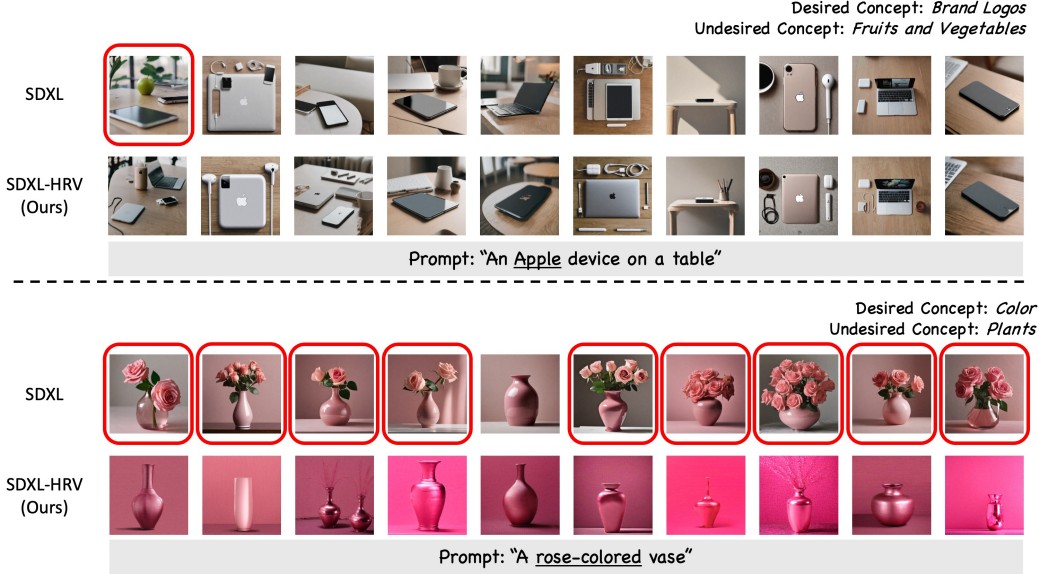

Figure 44: Two examples on misinterpretation reduction in SDXL. Images showing misinterpretations are marked with red boxes.

## H    LIMITATION

We examined 34 concepts listed in Table 3 and identified two types of failure cases. The first type stems from limitations in the underlying T2I model, where it struggles to correctly understand certain concepts. For example, in *Counting* and *Lighting Conditions*, the model fails to generate accurate outputs, as shown in Figure 45. The second type is related to our HRV or concept-words. Here, the T2I model correctly understands the concept, but MoRHF and LeRHF weakening fail to produce meaningful differences. An example of this is *Facial Expression*, with related failure cases shown in Figure 46.

In Figures 45-46, we generate images using SDXL with the same random seed for three prompts in each concept case. For the first type of failure, shown in Figure 45, the model often struggles to understand certain concepts, failing to distinguish between words like 'three' and 'four' in the *Counting* examples or 'natural light,' 'spotlight,' and 'dark light' in the *Lighting Conditions* examples. These issues make it difficult to assess whether HRVs identify appropriate CA head orderings relevant to these concepts, as the base model, SDXL, does not reliably generate the intended outputs. We believe such failures could be addressed in the future with more advanced T2I models. For the second type of failure, shown in Figure 46, SDXL correctly generates facial expressions that match the prompts. However, HRV fails to find meaningful CA head orderings, preventing it from distinguishing between MoRHF and LeRHF weakening. This may be due to the concept-words used for *Facial Expression* being too broad to represent the concept effectively. The concept-words for *Facial Expression* include Happy, Sad, Angry, Surprised, Confused, Laughing, Crying, Smiling, Frowning, Disgusted. We will further explore this limitation in future work.

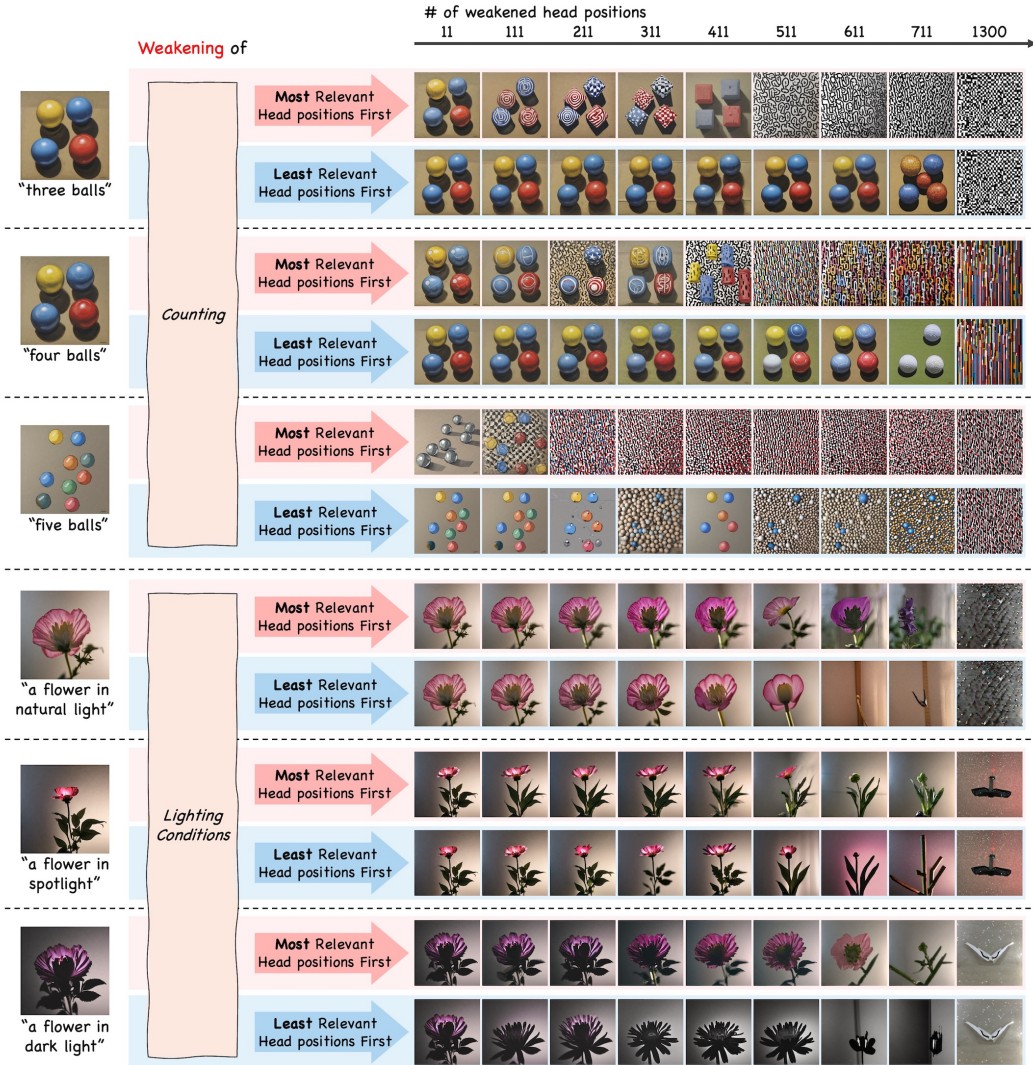

Figure 45: First type of failure cases: The baseline T2I model, SDXL, struggles to correctly understand the concepts, making it difficult to assess whether HRVs identify appropriate CA head orderings relevant to the corresponding concepts. The images are generated with SDXL.

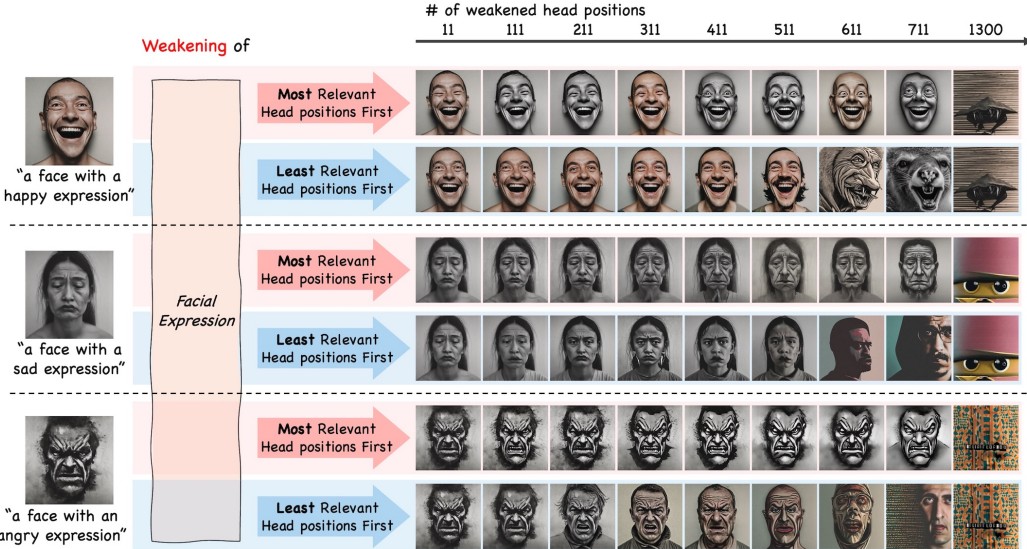

Figure 46: Second type of failure cases: Our HRV fails to identify the relevant CA head ordering for the corresponding concept, preventing it from distinguishing between MoRHF and LeRHF weakening. The images are generated with SDXL.

## I    ADDITIONAL ANALYSIS: THE EFFECT OF TIMESTEPS ON HEAD RELEVANCE VECTORS

In this section, we further analyze 1700 vectors (34 visual concepts×50 timesteps) obtained from Section 6.2. We start by reshaping these vectors into a tensor with dimensions 34×50×128, where 34 represents visual concepts, 50 represents timesteps, and 128 represents the number of CA head positions in the T2I model. We then average this tensor over the timestep dimension to obtain 34 vectors, each with a size of 128, corresponding to the head relevance vectors (HRVs). Similarly, averaging over the visual concept dimension yields 50 vectors, also of size 128, which we refer to as *timestep vectors*. To examine directional variations in the 34 HRVs, we compute and visualize their cosine similarities in Figure 47a. We also visualize the cosine similarities between the 50 timestep vectors in Figure 47b. Compared to Figure 47a, Figure 47b shows almost no directional variation between the 50 timestep vectors (note the colorbar scale). This supports the conclusion of Section 6.2, where we suggested that *the generation timesteps do not significantly affect the head relevance patterns of each visual concept*.

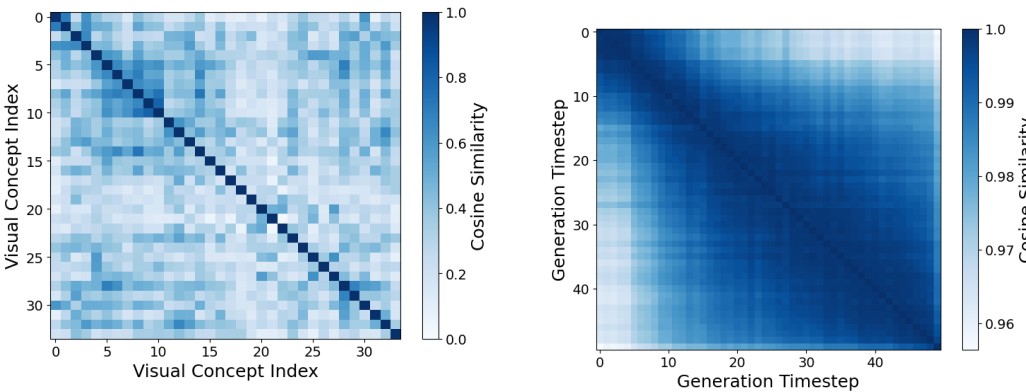

(a) Cosine similarities of 34 head relevance vectors (HRVs).

(b) Cosine similarities of 50 timestep vectors.

Figure 47: Cosine similarity plots of (a) 34 head relevance vectors and (b) 50 timestep vectors. Visual concepts are clearly separated, while timesteps are not.

## J    EXTENDING HUMAN VISUAL CONCEPTS

In this paper, we use 34 visual concepts to construct head relevance vectors (HRVs), but users can flexibly add or remove visual concepts as needed. In this section, we explore the effect of adding a new visual concept to the existing set of 34. To demonstrate this, we add the concept *Tableware*, creating a set of 35 extended visual concepts. We then construct HRVs individually for both the 34-concept and 35-concept sets and compare them through visualization. Stable Diffusion v1 has 16 multi-head CA layers, each containing 8 CA heads, for a total of 128 heads, making HRV visualization straightforward. In Figure 49, we visualize each set of HRVs, with darker colors representing higher values. The two sets of HRVs for the original 34 visual concepts (Figure 48a and Figure 48b) are highly similar. This indicates that adding the new concept *Tableware* does not significantly alter the patterns of HRVs for each concept. Additionally, Figure 49 shows 2 examples of ordered weakening analysis with the added concept *Tableware*, demonstrating that the HRV for the new concept *Tableware* is effective.

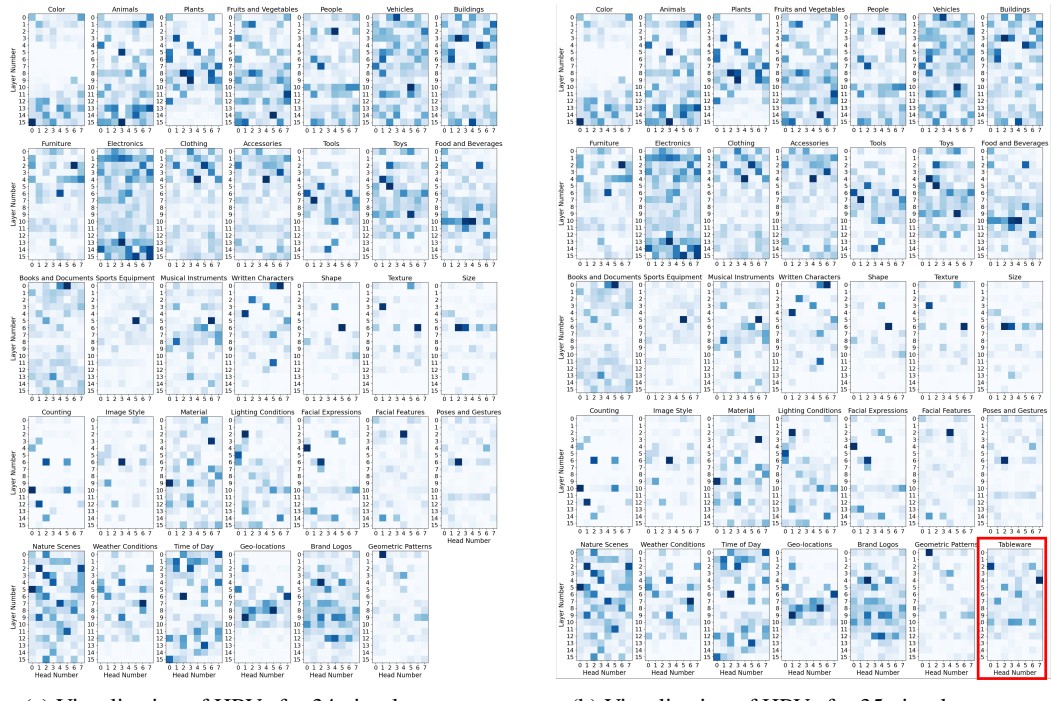

(a) Visualization of HRVs for 34 visual concepts     (b) Visualization of HRVs for 35 visual concepts

Figure 48: Visualization of head relevance vectors (HRVs) for (a) 34 visual concepts used in this paper, and (b) 35 extended visual concepts (the original 34 visual concepts plus the *Tableware* concept). HRVs for (a) and (b) are constructed individually (Best viewed with zoom).

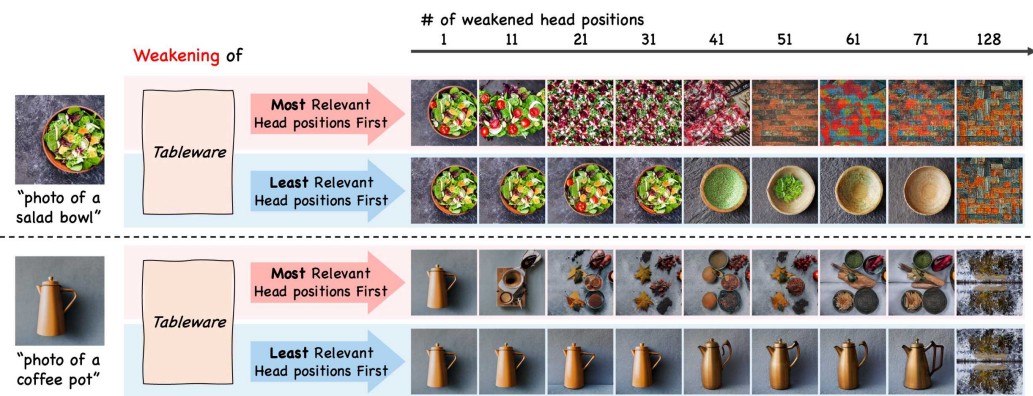

Figure 49: Ordered weakening analysis for *Tableware* concept: Generated images as weakening progresses in either MoRHF or LeRHF order using Stable Diffusion v1.4.

