# OpenReview forum: "Cross-Attention Head Position Patterns Can Align with Human Visual Concepts in Text-to-Image Generative Models"
_ICLR.cc/2025/Conference — ICLR 2025 Poster_

### Official Review · Reviewer_Efqm · 2024-10-25

**Soundness:** 3
**Presentation:** 2
**Contribution:** 3
**Rating:** 6
**Confidence:** 4

**Summary:**

This paper mainly focuses on the explainability of text-to-image diffusion model. The authors propose a new metric based on the cross-attention heads in the diffusion UNet to illustrate the correlation between each attention head and visual concepts. Based on the proposed Head Relevance Vectors, the authors further propose several applications including solving polysemous words problems and image editing.

**Strengths:**

1. The idea of correlating visual concepts with diffusion models is interesting.

**Weaknesses:**

1. I suggest the authors add textual description of the proposed HRV instead of directly showing Fig. 2 and Fig. 4 for better understanding.
2. I wonder why <SOT> and many  are required during update of HRV?
3. It would be better to used SDXL or some more recent models such as SD3 as primary model, given that SD1.5 is kind of outdated.
4. It would be better to add a random weakening baseline in Fig.3.
5. In Sec.5.1 the authors show that by utilizing HRV the SD can generate more proper concepts. I wonder if this method can be compared with using classifier guidance, where the model is encouraged to align the generated image with wanted concepts in terms of CLIP score.

**Questions:**

Please refer to the weaknesses.

---

> ### Author Response · Authors · 2024-11-22
> **Response to Reviewer Efqm**
>
> We are grateful for your constructive and valuable comments. We have revised our paper based on all of the reviewer’s suggestions and uploaded the updated version. The revisions are highlighted in blue for your convenience. Our point-by-point responses to your comments can be found below:
>
> 1. **Textual description of HRV for better understanding (W1)**:
> Thank you for your valuable feedback. In response, we have improved the explanation of HRV in Section 3 and included pseudo-code for HRV construction in Appendix B.1 of our revised version. We believe these updates enhance the clarity and readability of our work.
>
> 2. **<SOT> and many s during the update of HRV (W2)**:
> To clarify, our HRV construction does not require <SOT> and  tokens. In Stable Diffusion, CLIP text encoders map input prompts to textual embeddings of length 77, which include <SOT> and  token embeddings as padding. These special tokens are by-products of CLIP text encoders and are not part of our method. Instead, we update the HRV vectors using only the semantic token embeddings, excluding <SOT> and .
>
> 3. **SDXL or some more recent models as primary model (W3)**:
> Most image editing and multi-concept generation methods have been designed and implemented for SD v1, while well-functioning implementations for more recent models like SDXL or SD v3 are still rare. This is mainly because these SD models were introduced only recently and have not yet been thoroughly studied. For this reason, we integrated our algorithm into methods (P2P or A&E) based on SD v1, such that we can provide reliable analysis of how HRV can be effective for key applications. However, we believe our HRV can also improve performance on these newer models, given the architectural similarities between SD v1 and SDXL, as well as the success of our HRVs in identifying relevant CA head orderings, as shown by our ordered weakening analysis with SDXL. Also, please feel free to check out our responses to Reviewer d4iN where we summarize further investigation results. In future work, we look forward to applying our approach to algorithms based on these newer SD models.
>
> 4. **Add random weakening baseline in ordered weakening analysis (W4)**:
> We thank the reviewer for the insightful suggestion. We have added a new comparison with random order weakening to Table 6 in Appendix C.3 of our revised version, which is shown below:
> | |Material|Geometric Patterns|Furniture|Image Style|Color|Animals|Average|
> |--|--|--|--|--|--|--|--|
> | HRV (Ours)|**6.63**|**14.75**|**14.42**|**9.46**|**7.33**|**8.13**|**10.12**|
> | Random Order - Case 1|-1.94|4.29|-5.48|-1.81|-1.83|3.02|-0.63|
> | Random Order - Case 2|5.85|0.14|8.38|-1.89|-3.39|0.19|1.55|
> | Random Order - Case 3|1.68|-3.61|2.99|2.20|5.33|-2.91|0.95|
> | Random Order - Mean*|1.86|0.27|1.96|-0.50|0.04|0.10|0.62|
>
>    *'Random Order – Mean' represents the average value across the three random order cases.
>    This table compares HRV-based ordered weakening with three random weakening baselines for six visual concepts in the ordered weakening analysis. For better comparison, we calculate the area between the LeRHF (Least Relevant Head positions First) and MoRHF (Most Relevant Head positions First) line plots. Higher values indicate a CA head ordering that aligns more closely with the relevance of the corresponding concept. The results show that HRV-based ordered weakening identifies meaningful CA head orderings relevant to each visual concept.
>
> 5. **Compare SD-HRV with classifier guidance (W5)**:
>    - Thank you for your interesting suggestion. Current text-to-image diffusion models rely on classifier-free guidance, which approximates classifier guidance, to guide the generation process. Our approach, SD-HRV, addresses the issue of improper concept generation in these models by integrating concept adjusting technique into the classifier-free guidance component. The goal of SD-HRV is to reduce the misinterpretations that classifier-free guidance may encounter.
>    - For your reference, classifier guidance is rarely used in the community because it typically requires a pre-trained image classifier trained on images with varying levels of added Gaussian noise, introducing significant complexity. Furthermore, training a classifier for classifier guidance requires large annotated datasets based on a set of pre-defined classes. This is a strong constraint compared to HRV that can flexibly and easily incorporate any human-specified concepts, such as color, material, geometric patterns, or image style.
>
>
>
> We believe your comments and suggestions have greatly improved our work, and we would like to thank you once again for your constructive and helpful feedback.

---

> ### Author Response · Authors · 2024-12-01
> **Gentle reminder**
>
> Dear reviewer, thank you once again for your helpful review. We believe your feedback on the HRV explanation and randomly weakening baseline has improved the presentation and quality of the paper.
>
> We hope that our response and the revised paper have addressed all your concerns. As the author-reviewer discussion period is about to end, we would greatly appreciate any additional comments or questions you might have.

---

### Official Review · Reviewer_yNut · 2024-10-31

**Soundness:** 2
**Presentation:** 3
**Contribution:** 2
**Rating:** 6
**Confidence:** 5

**Summary:**

This paper tries to understand cross-attention (CA) layers *regarding attention heads*.
* The authors introduce N head relevance vectors (HRV) for N visual concepts.
* The strength of an attention head to the HRVs represent the relevance of the head to the concept.

Above properties are interpreted by *ordered weakening analysis*.
* Sequentially weaken the activations of CA heads to observe weakened visual concepts.

Boosting and reducing the strength of different heads control the strength of visual concepts. It helps three applications: 1) correcting mis-interpretation of words in t2i generation, 2) boosting prompt-to-prompt, 3) reducing neglect in multi-object generation.

**Strengths:**

1. This paper provides a new perspective in understanding the features in text-to-image generation: different heads.
2. Qualitative examples (Figure 3a) and CLIP similarities (Figure3b) along weakening MoRHF and LeRHF clearly show the effect of weakening different heads.
3. The appendix provides extensive qualitative results to remove doubt for cherry-picked results.
4. The proposed method is useful for three applications: 1) correcting mis-interpretation of words in t2i generation, 2) boosting prompt-to-prompt, 3) reducing neglect in multi-object generation.
5. Discussions resolve natural questions: extension to SDXL and effect across different timesteps.

**Weaknesses:**

1. The paper should provide principles of the proposed approaches.
    * L224 Why should we count each visual concept having the largest value to update the HRVs?
    * This is the most critical weakness for not giving a higher rating. I think the perspective is worth noticing but a solid paper should provide why/how it works.
    * Answering this question with theoretical justifications or intuition would strengthen the paper.
2. HRV should be described more clearly.
    * L205 a concatenation of token embeddings // concat along which axis? I guess the result of concatenation is $N \times (d + H)$. Then the query Q does not match the cross-attention operation because $Q\in R^d$. Am I missing something?
    * L210 K1, ..., KN should be denoted in Figure 2.
    * Adding equations and proper notations would help readers to understand the operation.
3. Human evaluation should be explained in more detail.  Appendix C.2 is not enough. Adding a table with Number of participants, Number and types of questions, Number of questions per participant, and Any quality control measures used would strengthen the user study.

Misc.: Related works -> Related work

**Questions:**

I wonder the interpolation between different strengths of a head. For example, interpolating  material=[-2, 2]?

---

> ### Author Response · Authors · 2024-11-22
> **Response 1 to Reviewer yNut**
>
> We are grateful for your constructive and valuable comments. We have revised our paper based on all of the reviewer’s suggestions and uploaded the updated version. The revisions are highlighted in blue for your convenience. Our point-by-point responses to your comments can be found below:
>
> 1. **Principles of our approaches (concerning the argmax operation) (W1)**:
> During HRV construction, we apply the argmax operation to the averaged cross-attention (CA) maps before using them to update the HRV matrix. **This step addresses the varying representation scales across $H$ CA heads**, and we are sorry for not providing further information in the original submission. To demonstrate these scale differences, we first compute the averaged L1-norm of the CA maps before applying the softmax operation for each CA head in Stable Diffusion v1.4. We then calculate **the mean and standard deviation** of these L1-norms across 2100 generation prompts and 50 timesteps. The table below shows some of these statistics for a few CA layers (a full table is provided in newly added Table 4 of Appendix B.2, along with further details on the L1-norm, including its mathematical expression, in Appendix B.2 of our revised version):
> | |Head 1|Head 2|Head 3|Head 4|Head 5|Head 6|Head 7|Head 8|
> |--|--|--|--|--|--|--|--|--|
> |**Layer 1**|1.16 ± 0.16|1.58 ± 0.19|1.08 ± 0.26|1.45 ± 0.18|1.73 ± 0.36|1.75 ± 0.39|1.89 ± 1.26|0.92 ± 0.22|
> |**Layer 2**|1.24 ± 0.21|1.19 ± 0.29|1.30 ± 0.24|1.43 ± 0.20|0.99 ± 0.34|1.08 ± 0.24|0.89 ± 0.12|1.07 ± 0.28|
> |**Layer 7**|1.64 ± 0.52|1.79 ± 0.70|1.30 ± 0.19|1.34 ± 0.22|2.37 ± 0.36|2.37 ± 0.30|3.04 ± 1.33|1.90 ± 0.54|
> |**Layer 8**|1.24 ± 0.14|2.06 ± 0.26|1.53 ± 0.20|1.64 ± 0.19|1.28 ± 0.21|1.66 ± 0.22|1.82 ± 0.20|2.14 ± 0.31|
> |**Layer 15**|1.20 ± 0.38|2.16 ± 0.46|1.88 ± 0.40|**4.11** ± 1.65|1.62 ± 0.39|0.76 ± 0.13|1.84 ± 0.28| 1.48 ± 0.37|
> |**Layer 16**|**0.51** ± 0.04|1.80 ± 0.33|1.14 ± 0.31|1.84 ± 0.33|0.91 ± 0.38|1.15 ± 0.18|1.17 ± 0.11| 1.06 ± 0.21|
>
>    The CA heads exhibit variation in their representation scales, with the head having the largest scale showing a mean value 8.1 times higher than that of the smallest scale. **Since the softmax operation maps large-scale values closer to a Dirac-delta distribution and small-scale values closer to a uniform distribution, it is necessary to align the scales between CA heads before accumulating the information into the HRV matrix.** We achieve this by simply applying the argmax operation, as shown in Eq. (4) of Appendix B.1 of our revised version.
>    We thank the reviewer for the helpful feedback, and we have added this explanation to Appendix B.2 of our revised version, which we believe strengthens our paper.
>
> 2. **HRV should be described more clearly (W2)**:
>    - **Concatenation of token embeddings**: Token embeddings are concatenated along the token dimension. Specifically, each key-projected embedding, $K_m$, has a shape of $77 \times F$, where $F$ is the feature dimension. From this, we extract the semantic token embedding, denoted as $\widehat{K}_m$, which has a shape of $n_m \times F$, where $n_m$ depends on the token length of the concept-words (e.g., for the concept-word 'white', $n_m=1$). These embeddings are then concatenated across the token dimension (corresponding to $n_m$), resulting in $\widehat{K}$ with a shape of $N' \times F$. The image query matrix, $Q$, has a shape of $R^2 \times F$, where $R$ represents the width (or height) of the image latent, and the attention map, $\widehat{M}$, has a shape of $R^2 \times N'$.
>    We have clarified the dimensions of all tensors and matrices in Section 3 of our revised version to improve readability.
>    - **$K_1, …, K_N$ in the figure**: Thank you for the helpful suggestion. We have added $K_1, \dots, K_N$ to Figure 2 in our revised version.
>    - **Adding equations and proper notations**: We appreciate the suggestion and have clarified the notations in Section 3 of our revised version. Additionally, **we have included pseudo-code for HRV construction in Appendix B.1 of our revised version.** The equations involved in HRV construction are now presented in the pseudo-code.
>    We believe these changes, prompted by the reviewer’s feedback, significantly enhance the readability of our work.
>
> 3. **Human evaluation detail (W3)**:
> Thank you for your suggestion. In our revised version, we have added more details about the human evaluation in Appendix D.2 and E.3, along with tables (Table 8 and Table 10) that summarize all the relevant information.

---

> ### Author Response · Authors · 2024-11-22
> **Response 2 to Reviewer yNut**
>
> 4. **Interpolation between different strengths of CA heads (Q1)**:
> Thank you for your insightful question. In response, we have added two examples in Figures 16-17 in Appendix C.4 of our revised version, demonstrating the interpolation of rescaling factors within the range $[-2, 2]$. These figures show that strengthening (rescaling factors greater than 1) produces minimal changes, likely because the concept is already well-presented in the original images. In contrast, weakening works effectively with factors below 0, with stronger effects observed as the factor decreases further.
>
> We believe your comments and suggestions have greatly improved our work, and we would like to thank you once again for your constructive and helpful feedback.

---

> > ### Comment · Reviewer_yNut · 2024-11-24
> >
> > I thank the authors for the rebuttal.
> >
> > 1. The answer is not directly connected to my question. Why should we count them rather than other choices such as sum, max, etc.?
> > 2. Great.
> > 3. Great.
> > 4. Great.

---

> ### Author Response · Authors · 2024-11-24
> **Thank the reviewer & follow-up response to the reviewer**
>
> Dear Reviewer yNut. We apologize for any confusion and have clarified our previous response regarding W1 below.
>
> As we mentioned earlier, the purpose of counting is to address the varying representation scales across the $H$ cross-attention (CA) heads, where $H$ is the number of CA heads in a T2I model. These differences in scales cause the $N$-length vectors calculated for each CA head (prior to the argmax operation shown in Figure 2 of our manuscript) to exhibit different distributions: **Due to the softmax operation, CA heads with larger-scale values produce vectors closer to a Dirac-delta distribution, while those with smaller-scale values produce vectors closer to a uniform distribution.** To clarify further, we denote the $N$-length vector for the $h$-th CA head as $\tilde{M}_h$, where $N$ is the number of concepts being compared.
>
> - If we **sum** these vectors to update the HRV matrix, $\tilde{M}_h$ vectors closer to a Dirac delta distribution overly emphasize their largest concept, whereas $\tilde{M}_h$ vectors closer to a uniform distribution underrepresent their largest concept. **This imbalance favors CA heads with larger representation scales.** For example, according to the table from our previous response, the largest concept chosen by the CA head at [Layer 15-Head 4] would be overemphasized compared to the largest concept chosen by the CA head at [Layer 16-Head 1].
>
> - Similarly, using [**max** -> sum] results in the same issue, as the maximum value of $\tilde{M}_h$ for CA heads with larger representation scales will be much higher than for those with smaller scales.
>
> - However, applying [**argmax** -> sum] (**equivalent to counting; our method**) ensures that the largest concept from each CA head contributes a value of 1 to the HRV matrix, **regardless of the representation scale of its corresponding CA head.**
>
> +) We have revised L224 of the original manuscript to clarify this point. In the revised version, the explanation can be found in L222–L225, with full details provided in Appendix B.2. Additionally, we have added a new explanation in Appendix B.2 to clarify why we should use [argmax->sum] (equivalent to counting) instead of [sum] or [max->sum].
>
> We appreciate your feedback and would be happy to provide further clarifications if needed.

---

> ### Author Response · Authors · 2024-12-01
> **Gentle reminder**
>
> Dear reviewer, thank you once again for your helpful review and follow-up feedback. We believe your comments on the three weaknesses in our original submission have greatly enhanced the quality of the paper.
>
> We hope that our follow-up response and the revised paper have addressed all your concerns. As the author-reviewer discussion period is about to end, we would greatly appreciate any additional comments or questions you might have.

---

### Official Review · Reviewer_d4iN · 2024-11-01

**Soundness:** 3
**Presentation:** 3
**Contribution:** 3
**Rating:** 8
**Confidence:** 4

**Summary:**

The paper proposes a method of constructing so called "HRV" vectors, which align with visual concepts. The authors leverage cross-attention layers of Stable Diffusion model to learn those vectors for predefined concepts. The proposed method helps to solve three known issues of the image synthesis task.

**Strengths:**

1. Good motivation and a clear idea
2. Comprehensive quantitative and qualitative comparisons with many other solutions
3. The experiments, settings, and other details are mainly clearly explained

**Weaknesses:**

1. It requires fixing a set of concepts beforehand for every HRV construction. Does not have a study of how the HRV matrix will be changed when some concepts are changed or replaced after the construction.
2. Manual settings, choice, and configuration are required for every concept (case) during inference (Sec 5.1, Fig 5).
3. Lack of failed cases, there are no details about the limitations of this method.
4. Even though there is a section for bigger / novel models (SDXL), all experiments, studies, and comparisons are based on SD v1. New models might eliminate many issues the proposed method tries to solve.

**Questions:**

1. Could you give more details about why there are some irrelevant concepts after a certain point of ordered weakening (Fig 9)?
2. Could you give more details about how the "h" is chosen/computed in the method of HRV updates?

---

> ### Author Response · Authors · 2024-11-22
> **Response 1 to Reviewer d4iN**
>
> We are grateful for your constructive and valuable comments. We have revised our paper based on all of the reviewer’s suggestions and uploaded the updated version. The revisions are highlighted in blue for your convenience. Our point-by-point responses to your comments can be found below:
>
> 1. **How the HRV matrix will be changed when some concepts are changed (W1)**:
> The core idea behind our HRV construction is to iteratively compare different concepts across a sufficiently large number of random image generations*. As shown in Appendix J of our revised version (Appendix I in the original manuscript), adding or removing a concept has minimal impact on the constructed HRV vectors. Figure 48 in Appendix J illustrates this by comparing two sets of HRVs: one constructed with 34 visual concepts and another with 35 concepts (the original 34 plus a new concept, ‘Tableware’). The figure shows that the HRVs for the original 34 visual concepts remain nearly identical, even when HRVs for the expanded 35 are constructed independently from the scratch. This suggests that users can either retain previously calculated HRV vectors or recompute them when making minor updates to the set of visual concepts.
> *Constructing HRV vectors for a set of concepts takes about 25 minutes using 10 RTX 3090 GPUs with SD v1. There is no limit to the number of concepts that can be compared.
>
> 2. **Manual settings for reducing misinterpretation (W2)**:
> As the reviewer pointed out, our current implementation for reducing misinterpretation requires manual settings for the target token, as well as the desired and undesired concepts. We did not explore automating this process because our primary focus was on analyzing HRVs and demonstrating their effectiveness in various visual generative applications. However, constructing an automatic pipeline using LLMs is feasible. To explore this, we queried GPT-4o to see if it could extract the target token, desired concepts, and undesired concepts for use in our SD-HRV algorithm. Below, we provide an example of the LLM query and its response:
> | Query to the LLM |
> | -- |
> | Consider the following generation prompt for Stable Diffusion: **A vase in lavender color.** The model might misinterpret some words, leading to inaccurate image generation. Can you identify the polysemous word that Stable Diffusion might misinterpret? Refer to the following concepts that Stable Diffusion might confuse: {List of the 34 visual concepts used in our work.}|
>
>    | Response from the LLM |
>    | -- |
>    | The polysemous word in the generation prompt "A vase in lavender color" that Stable Diffusion might misinterpret is **"lavender."** This word can refer to multiple concepts:
>    **Color**: "Lavender" is commonly understood as a shade of purple.
>    **Plant**: "Lavender" is also a type of flowering plant.
>    Given that Stable Diffusion may associate "lavender" with either the color or the plant, it could generate an image that reflects either of these meanings, leading to a misinterpretation depending on the context. |
>
>    As shown in the LLM's response, it accurately identifies the inputs needed for SD-HRV. This suggests that an automatic pipeline could be developed by integrating the LLM with SD-HRV. We have added a comment in our paper that this automation can be possible with an advanced LLM.
>    Thank you very much for your feedback.
>
> 3. **Lack of failed cases (W3)**:
>  In Appendix H of our revised version, we have expanded the limitation section to discuss two types of HRV failure cases identified during our examination of all 34 visual concepts used in our study. The first type, shown in Figure 45, involves 'Counting' and 'Lighting Conditions,' and stems from limitations in the underlying T2I models. Specifically, SDXL struggles to accurately interpret these visual concepts, generating inaccurate outputs and complicating the evaluation of HRVs. The second type, shown in Figure 46, relates to 'Facial Expression,' where HRV fails to identify a relevant CA head ordering. We suspect this issue arises from the concept-words used for 'Facial Expression' being too broad to effectively capture the concept. We will further address this limitation in future works.
> We believe the newly added Figures 45-46 better clarify the limitations of our work, and we appreciate your helpful comment

---

> ### Author Response · Authors · 2024-11-22
> **Response 2 to Reviewer d4iN**
>
> 4. **SDXL might eliminate many issues (W4)**:
> As the reviewer noted, SDXL significantly improves image generation performance, and this enhanced capability is likely to extend to other visual generative applications. However, our tests on SDXL for misinterpretation and multi-concept generation tasks revealed that further improvements are still needed as summarized below.
>    - For misinterpretation, SDXL generates desired concepts more effectively than the earlier SD v1.4, but it still tends to produce undesired concepts. Figure 44 in Appendix G.3 of our revised version illustrates this issue, which is effectively addressed by our SDXL-HRV (SDXL with our concept-adjusting method).
>    -  For multi-concept generation, we evaluated SDXL on two benchmark types used in our study, with results shown in the tables below:
>    | Method (Type 1 benchmark; higher values are better) | Full Prompt | Min. Object | BLIP-score |
>    |--|--|--|--|
>    | SD v1.4 | 0.3000 | 0.1611 | 0.5934 |
>    | A&E (based on SD v1.4) | 0.3544 | 0.2017 | 0.7049 |
>    | A&E-HRV (Ours, based on SD v1.4) | **0.3702** | **0.2078** | **0.7491** |
>    |SDXL| 0.3565 | 0.1910 | 0.6979 |
>    ||
>    | **Method (Type 2 benchmark; higher values are better)** | **Full Prompt** | **Min. Object** | **BLIP-score** |
>    | SD v1.4 | 0.3420 | 0.1458 | 0.5633 |
>    | A&E (based on SD v1.4)  | 0.3883 | 0.1972 | 0.6373 |
>    | A&E-HRV (Ours, based on SD v1.4) | **0.3971** | **0.2073** | **0.6580**|
>    | SDXL | 0.3928 | 0.1899 | 0.6350 |
>
>       The results indicate that SDXL performs comparably or slightly worse than Attend-and-Excite (A&E; implemented on SD v1.4), which still suffers from catastrophic neglect. Notably, our A&E-HRV, even when implemented on SD v1.4, clearly outperforms SDXL. Although we could not test A&E-HRV on SDXL because A&E implementation for SDXL is not available, we expect improved performance given the architectural similarities between SD v1.4 and SDXL, as well as the success of our HRVs in identifying relevant CA head orderings, as demonstrated by our ordered weakening analysis with SDXL.
>    - Lastly, we believe that SDXL may also require advancements in image editing tasks. However, we were unable to evaluate this due to the lack of well-functioning image editing methods implemented for SDXL. Most existing approaches are implemented for SD v1 models, and the third-party implementations we tested for SDXL did not perform well.
>
>    For multi-concept generation and image editing, we look forward to integrating our algorithm into new methods based on SDXL in future works.
>
> 5. **Some irrelevant concepts after a certain point of ordered weakening (Q1)**:
> Our ordered weakening analysis sequentially weakens the activation of $H$ cross-attention (CA) heads (for all semantic tokens), following the order specified by the HRVs. Since the only condition imposed is this specified weakening order, random influences—such as the appearance of irrelevant concepts—may naturally arise at certain points during the process.
>
> 6. **How the "h" is chosen/computed in HRV updates (Q2)**:
> Figure 2 in Section 3 illustrates the update process for HRV vectors at each head position $h$. This process is repeated across all head positions ($h=1,…,H$) and timesteps ($t=1,…,T$) for a sufficiently large number of random image generations (with $H=128$ and $T=50$ for SD v1.4, and $H=1300$ and $T=50$ for SDXL). In other words, each random image generation involves $H \times T$ updates to the HRV vectors. We have clarified this in the caption of Figure 2 and in Section 3 of our revised version.
>   We thank the reviewer for their comment, which has helped improve the clarity of our explanation.
>
> We believe your comments and suggestions have greatly improved our work, and we would like to thank you once again for your constructive and helpful feedback.

---

> ### Author Response · Authors · 2024-12-01
> **Gentle reminder**
>
> Dear reviewer, thank you once again for your helpful review and follow-up feedback. We believe your detailed comments on our original submission have greatly enhanced the quality of the paper.
>
> We hope that our follow-up response and the revised paper have addressed all your concerns. As the author-reviewer discussion period is about to end, we would greatly appreciate any additional comments or questions you might have.

---

### Official Review · Reviewer_miju · 2024-11-03

**Soundness:** 3
**Presentation:** 3
**Contribution:** 2
**Rating:** 6
**Confidence:** 5

**Summary:**

This work proposes Head Relevance Vectors (HRVs). HRs are an extension of the findings from previous works such as Hertz et al.'s P2P where cross attention maps were used to better understand t2i models and to edit images via prompts. HRV proposes using multiple concept words and concatenating them into a concept embedding matrix K which can then be applied to different heads of the cross-attention and by doing so, disentangle the different heads based on the concept they seem to be focusing on. The authors show this disentanglement of heads based on the concepts learned improved editing of images.

**Strengths:**

The motivation of the paper is clear and build on a well studied problem of understanding the role of cross-attention and what they learn in editing T2I models. The experiments are visually appealing and tell the story of the paper well, especially the weakening of HRVs that shows weakening based on the most and least relevant concepts / heads. The authors show that using HRVs to edit images works better than SDEdit,P2P, etc. They also show improvement over Attend and Excite for the problem of catastrophic forgetting in T2I models.

 While in the weaknesses, I do mention my thoughts on the originality of this work, I believe using previous findings around CAs and targeting different heads and their roles in generating different concepts would be interesting to the community.

**Weaknesses:**

I would argue that the work, while interesting, does not have new insight compared to what previous works such as P2P and Diffusion Self-Guidance have already already shown in regards to the role of cross-attentions. However, this work does take a step towards using those findings to narrow down on head-level manipulation of concept vectors. It goes without saying that T2I models could benefit from more comprehensive evaluation on larger set of generated images / human evaluation. However, I do understand the challenges this poses as well.

**Questions:**

There have been recent works that show the <SOT> and  CAs capture different concepts. I would be interested to see if the authors found anything interesting regarding HRV and these tokens. I am also curious as to how the weakening and strengthening would work on more complex images that share entangled objects and concepts. For instance, what would weakening of "melting" look like for "a plastic car melting". I think this would be an interesting experiment since adjective and verb concepts are entangled with an object in a given image and HRV might to better in these cases than the counterparts.

---

> ### Author Response · Authors · 2024-11-22
> **Response to Reviewer miju**
>
> We are grateful for your constructive and valuable comments. We have revised our paper based on all of the reviewer’s suggestions and uploaded the updated version. The revisions are highlighted in blue for your convenience. Our point-by-point responses to your comments can be found below:
>
> 1. **No new insights compared to previous works**:
> We respectfully disagree with your feedback that our work offers no new insights compared to previous methods such as P2P and Diffusion Self-Guidance. While earlier works have shown how cross-attention (CA) layers can preserve or control structural information (such as position, size, or shape), they have not explored how these layers can be used to control human-specified concepts like color, material, or geometric patterns. To the best of our knowledge, our work is the first to demonstrate that a wide range of human-specified visual concepts can be aligned with CA head position patterns by focusing on CA layers at the head level. Our approach also enables flexible control over concepts, as we have shown through three applications. We believe this provides a valuable new insight for the community, expanding not only our understanding on CA layers but also our capability for effectively utilizing them.
>
> 2. **Our findings about <SOT> and s**:
> Thank you for your interesting suggestion. Since the CLIP text encoder in Stable Diffusion is a causal language model, all CLIP embeddings for the <SOT> token are the same (because it is the first token), regardless of the input prompt. This makes constructing HRVs impossible, as it requires iterative comparisons between different concepts. In contrast, the  token embedding indirectly incorporates semantic token information, which allows it to be used in HRV construction. Therefore, we have performed an extra experiment to address your question. In the table below, we compare HRVs constructed using semantic tokens (our method) and those using  tokens in the ordered weakening analysis for three visual concepts:
>    | | Material | Geometric Patterns | Furniture | Average  |
>    |------------|----------|--------------------|-----------|----------|
>    | HRV with semantic tokens (Ours)| **6.63** | **14.75**| **14.42** | **10.69**|
>    | HRV with \<EOT\>s| 5.07|8.81| 13.95     | 6.94     |
>
>    To compare the two HRVs effectively, we compute the area between the LeRHF (Least Relevant Head positions First) and MoRHF (Most Relevant Head positions First) line plots. Higher values indicate a CA head ordering that better aligns with the relevance of the corresponding concept. In the table, the HRV with  tokens shows an average performance of 6.94, which is much lower than the 10.69 achieved by our HRV constructed with semantic tokens. We believe this difference stems from the fact that semantic tokens directly incorporate semantic information, while  tokens do so indirectly.
>    We hope this addresses your question.
>
> 3. **Ordered weakening analysis on more complex images**:
> Thank you for your interesting suggestion. In our revised version, we have added additional examples using the prompts 'a plastic car melting' and 'a metal chair rusting' in Figure 43 of Appendix G.2. In the MoRHF weakening of 'melting' for the generation of 'a plastic car melting,' the concept of 'melting' is eliminated first, while 'car' persists for a longer period. The entangled property of 'plastic' is affected during the weakening of 'melting,' but it is eliminated slightly later than 'melting' itself. This effect is even more noticeable in the MoRHF weakening of 'rusting' for the generation of 'a metal chair rusting,' where 'metal' and 'chair' remain longer than 'rusting.'
> Thank you for your interesting question. We believe the newly added Figure 43 would be insightful to the readers.
>
> We believe your comments and suggestions have greatly improved our work, and we would like to thank you once again for your constructive and helpful feedback.

---

> ### Comment · Reviewer_miju · 2024-11-26
> **Acknowledged**
>
> Thanks to the authors for responding to my questions and concerns. I have no further questions.

---

> > ### Author Response · Authors · 2024-11-26
> > **Thanks for the response**
> >
> > We thank the reviewer for the review and response again. We believe the HRV and the findings presented in our revised paper may inspire interesting directions for future work.

---

### Meta-Review · Area_Chair_WV39 · 2024-12-20

**Metareview:**

The paper presents a method for constructing Head Relevance Vectors (HRVs) in text-to-image diffusion models to align with relevant visual concepts. These HRVs are vectors that indicate the importance of each cross-attention head for a given visual concept. The study demonstrates the effectiveness of HRVs through ordered weakening analysis and introduces methods for concept strengthening and adjusting to enhance visual generative tasks.
The paper's strengths include: (1) Clear motivation and idea building on existing research. (2) Comprehensive comparisons with other solutions. (3) Demonstrated effectiveness over existing methods like Attend and Excite. The paper's weaknesses include: (1) New insights over P2P and Diffusion Self-Guidance are marginal. (2) Unclearity and missing details on certain parts (e.g., human evaluation). (3) Requires fixing set concepts for HRV construction.
The majority of the reviewers' concerns were considered properly addressed in the rebuttal and all reviewers lean on the positive side. Hence, the AC would like to recommend for acceptance.

**Additional Comments On Reviewer Discussion:**

Most reviewers have acknowledged the rebuttal and considered their concerns to be properly addressed.

---

### Decision · Program_Chairs · 2025-01-22

Accept (Poster)